# On the spectral bias of two-layer linear networks

**Aditya Varre**
EPFL
aditya.varre@epfl.ch

**Maria-Luiza Vladarean**
EPFL
maria-luiza.vladarean@epfl.ch

**Loucas Pillaud-Vivien**
Courant Institute of Mathematics, NYU / Flatiron Institute
lpillaudvivien@flatironinstitute.org

**Nicolas Flammarion**
EPFL
nicolas.flammarion@epfl.ch

## Abstract

This paper studies the behaviour of two-layer fully connected networks with linear activations trained with gradient flow on the square loss. We show how the optimization process carries an *implicit bias* on the *parameters* that depends on the scale of its initialization. The main result of the paper is a variational characterization of the loss minimizers retrieved by the gradient flow for a specific initialization shape. This characterization reveals that, in the small scale initialization regime, the linear neural network's *hidden layer* is biased toward having a low-rank structure. To complement our results, we showcase a hidden mirror flow that tracks the dynamics of the singular values of the weights matrices and describe their time evolution. We support our findings with numerical experiments illustrating the phenomena.

## 1 Introduction

The most forceful driver of advancements in the field of Machine Learning over the past decades has been the success of deep neural networks. Amongst the striking qualities of these models is the fact that, despite being heavily overparametrized, their optimization consistently yields minima with good generalization properties. A beckoning research direction is thus to unravel the process through which neural networks learn internal representations for a given task [Bengio et al., 2013]. Understanding such phenomena is crucial for lowering the interpretability barrier of these models and developing a principled approach to their training and deployment in practice.

Recent experimental evidence identified one of the likely paths towards achieving these goals as the study of the inherent regularization properties (or implicit biases) of training algorithms [Neyshabur et al., 2014, Zhang et al., 2016]. These observations laid the foundation for a new line of work [see, e.g., Vardi, 2022] whose driving question is which minimum, amongst the many, awaits at the tail end of optimization.

One of the determining factors for the implicit bias of gradient methods is the initialization scale, which controls their operational regime as shown by empirical studies [Chizat et al., 2019]. More precisely, gradient descent with a low-scale initialization is capable of learning rich feature representations from the data. Strikingly, despite overparameterization, the hidden-layer neurons align in the direction of the features [Chizat et al., 2019, Atanasov et al., 2022] and learning of representations reflects in the low-rank structure of the hidden layers. Our work aims to precisely explain this phenomenon and quantify the impact of the initialization scale on feature learning.

Unfortunately, studying such phenomena for the types of neural networks used in practice is mathematically challenging at present due to the non-linearity of their activations. Their less expressive *linear* counterparts, however, are more tractable and represent a good proxy due to their non-convex

37th Conference on Neural Information Processing Systems (NeurIPS 2023).

loss landscape and non-linear learning dynamics. Consequently, the study of deep linear networks has received significant amounts of attention over the past years, and spans several important directions, including convergence [Arora et al., 2019a, 2018, Min et al., 2021], learning dynamics [Saxe et al., 2014, Braun et al., 2022] and the implicit bias of optimization algorithms [Azulay et al., 2021]. This work complements these previous approaches by mathematically describing the properties of their parameters at convergence, highlighting the implicit bias phenomenon, and further analyzing the evolution of weight matrices throughout the optimization process.

Specifically, this paper studies overparameterized vector regression problems on two-layer fully-connected linear neural networks. We show the following results when the network is trained with gradient flow (GF).

(i) In Theorem 3.1, we prove that the zero-loss solutions retrieved by the gradient flow are the minimizers of a potential that depends on the initialization scale. Additionally, we provide explicit expressions for the singular values of the hidden layer weights, also as a function of the initialization scale. These characterizations reveal how low-magnitude initialization induces a low-rank structure of the hidden layer.

(ii) In Theorem 3.2, we show that gradient flow on the parameters induces a mirror flow on the singular values. In the specific case of scalar regression, we show that the gradient flow on the weights is equivalent to a mirror flow on the linear predictor. These characterizations give the geometrical structure of the training dynamics of linear neural networks.

(iii) In Proposition 4.1, we design a simple process to analytically describe how stochastic noise in the training algorithm can likewise induce low-rank structures in the weights *regardless of the initialization scale*.

We proceed by presenting related work in Section 1.1, formalizing the problem setup and assumptions in Section 2, stating and discussing our results in Section 3, and finally, we provide supporting numerical evidence in Section 4.

## 1.1 Related Work

The first pillar of our work addresses the implicit bias of GF and its stochastic variant in regression problems. One of the hallmarks of bias in this setting is the impact of initialization scale: large initial weights induce a learning regime in which the parameters travel a short distance to convergence and feature learning fails to happen (*lazy* regime), while small initialization effects a polar opposite behaviour of the system (*rich* regime) Chizat et al. [2019], Woodworth et al. [2020]. Training and generalization in the lazy regime are well-studied [Jacot et al., 2018, Du et al., 2019a, Arora et al., 2019c, Soltanolkotabi et al., 2017], however this scenario fails to capture the observed behaviour of neural networks in practice [Ghorbani et al., 2019]. While the rich regime more faithfully approximates the feature learning abilities of these models, it is comparatively more challenging to analyze and few results are known. Amongst them are those concerning diagonal linear networks, where a preference towards sparse representations is shown [Woodworth et al., 2020], and a restricted setting of the matrix factorization problem, where the implicit bias leads to low-rank representations [Gunasekar et al., 2017, Arora et al., 2019b, Li et al., 2018]. We similarly study the rich representation learning regime and provide initialization scale-dependent statements on implicit bias for two-layer fully connected linear networks.

Most theoretical results on the implicit bias of GF in overparametrized models rely on the identification of a related mirror flow in a reparametrized space [Gunasekar et al., 2018]. Diagonal linear networks are amenable to this technique and therefore well-studied [Woodworth et al., 2020, Pesme et al., 2021]. For linear fully-connected networks, however, the existence of a mirror flow is not always guaranteed [Li et al., 2022]. To partly alleviate this issue, Azulay et al. [2021] introduce a nonlinear time-rescaling technique and show that for scalar least-squares regression on a two-layer fully connected network with zero-balance initialization, the implicit bias selects low $\ell_2$-norm predictors. We prove a similar result under *imbalanced* initialization controlled by a scale parameter, and characterize the weight matrices independently at convergence, thus presenting a higher-resolution view of the problem.

Other works on linear networks include [Min et al., 2021] where convergence is studied in the presence of weight imbalance and implicit bias results are provided in the functional space; and [Yun

et al., 2021] where tensor networks are studied with the goal of unifying the implicit bias results for linear parameterization. In the case of linear networks, Yun et al. [2021] further show an implicit bias towards minimum $\ell_2$ linear predictors for vanishingly small initializations. For classification in the case of linear networks, Ji and Telgarsky [2019] show that the weights grow to infinity and the layers of the deep linear network align during the course of optimization. Timor et al. [2023] show a similar phenomenon happens for two-layer ReLU networks.

The second pillar concerns the learning dynamics of linear neural networks. The two-layer case optimized with GF on the square loss has been studied by Fukumizu [1998], Saxe et al. [2014, 2019], Braun et al. [2022]. The common setup of these works is that of zero-balance initialization and whitened data. First, Saxe et al. [2014, 2019] provide expressions for the temporal evolution of singular values of the predictor by assuming decoupled dynamics and a specific data-dependent initialization of the weights. This latter condition is alleviated by the approach of Fukumizu [1998] and Braun et al. [2022], Tarmoun et al. [2021] who solve a matrix Ricatti equation yielding solutions for the weight dynamics in the case where the network initialization has full rank. Finally, Gidel et al. [2019] loosen the whitened data assumption through a perturbation analysis and provide the time-evolution of singular values of the weight matrices. Our work removes the requirement of zero-balanced initialization and full-rank network initialization, and gives formulas for the weights' evolution as a function of the initialization scale. We further provide mirror flows on the weights' singular values and show that components are learned in a hierarchical manner for the case of whitened data.

Related work is further addressed in the following sections, as part of the discussion of results.

## 2   Preliminaries and problem setup

**Notation.**   Time-dependent variables are written in bold fonts: we drop the $t$ in $A(t)$ and simply denote it as $\mathbf{A}$. The time derivative of such variables is denoted $\frac{d}{dt}A(t)$ as $\dot{\mathbf{A}}$.

**Vector Regression.**   The set-up is that of standard vector regression problems with inputs $(x_1, \ldots, x_n) \in \left(\mathbb{R}^d\right)^n$ and outputs $(y_1, \ldots y_n) \in \left(\mathbb{R}^k\right)^n$ in the so-called overparametrized regime where $d \geq n$. Regarding the output dimension, the reader may keep in mind throughout the article that $k \ll d$, though the analysis holds for any $k, d$ pair. In order to learn the input/output rule, we minimize the square loss over a class of parametric models $\mathcal{H} = \{f_{\boldsymbol{\theta}}(\cdot) : \mathbb{R}^d \to \mathbb{R}^k \mid \boldsymbol{\theta} \in \mathbb{R}^p\}$ which we specify in the next paragraph. The train loss therefore can be written as

$$\mathcal{L}(\boldsymbol{\theta}) = \frac{1}{2n} \sum_{i=1}^{n} \left(y_i - f_{\boldsymbol{\theta}}(x_i)\right)^2. \tag{2.1}$$

**Parameterization with a Linear Network.**   We consider the parametric model of *two-layer linear neural networks* of width $l \in \mathbb{N}^*$: this corresponds to the parametrization $\boldsymbol{\theta} = (\boldsymbol{W}_1, \boldsymbol{W}_2)$, $\boldsymbol{W}_1 \in \mathbb{R}^{d \times l}$, $\boldsymbol{W}_2 \in \mathbb{R}^{l \times k}$ and $f_{\boldsymbol{\theta}}(x) = \boldsymbol{W}_2^\top \boldsymbol{W}_1^\top x$. The model is linear in the input $x$, and in terms of expressivity, it is equivalent to the linear class of predictors given by $f_{\boldsymbol{\beta}}(x) = \boldsymbol{\beta}^\top x$, with $\boldsymbol{\beta} = \boldsymbol{W}_1 \boldsymbol{W}_2$. We henceforth use the symbol $\boldsymbol{\beta}$ to denote the associated linear predictor of the network. An important consequence of this reparametrization is that the prediction function $f_{\boldsymbol{\theta}}$ is positively homogeneous of degree 2 in $\boldsymbol{\theta}$: $\forall \lambda \in \mathbb{R}$, it holds that $f_{\lambda \boldsymbol{\theta}} = \lambda^2 f_{\boldsymbol{\theta}}$, as it is the case for two-layer ReLU networks. This property has important consequences in the loss landscape through which $\boldsymbol{\theta}$ goes.

**Train loss.**   Assume momentarily that $k = 1$ and denote $\phi(x) = \boldsymbol{W}_1^\top x \in \mathbb{R}^l$. It is then clear that the predictor rewrites as $f_{\boldsymbol{\theta}}(x) = \langle \phi(x), \boldsymbol{W}_2 \rangle$. For this reason, we call the hidden layer $\boldsymbol{W}_1$ the *feature layer* and the last layer $\boldsymbol{W}_2$ the *weight matrix*. We study the *overparametrized setting* where $l \gg d$. Letting $X^\top := [x_1, \ldots, x_n]$ and $Y^\top := [y_1, \ldots, y_n]$, the loss becomes

$$\mathcal{L}(\boldsymbol{W}_1, \boldsymbol{W}_2) = \frac{1}{2N} \|X \boldsymbol{W}_1 \boldsymbol{W}_2 - Y\|^2. \tag{2.2}$$

For brevity, we ignore the $N$ in Eq.(2.2) by implicitly rescaling the data as $(X, Y) \leftarrow (X/\sqrt{N}, Y/\sqrt{N})$.

**Interpolators.** Note that when $Y \in \mathrm{span}(X)$ and $X$ is non-degenerate (which occurs with probability one if e.g., $X, Y$ are Gaussian and $d \geq n$), there always exists a solution which attains *zero* loss, i.e., $\boldsymbol{\beta}^* \in \mathbb{R}^{d \times k}$ such that $X\boldsymbol{\beta}^* = Y$. We emphasize the fact that there are two levels of overparametrization here: on one hand, when $d > n$, the set of zero loss linear predictors $\mathcal{I}_{\boldsymbol{\beta}} := \{\boldsymbol{\beta} \in \mathbb{R}^{d \times k} \mid X\boldsymbol{\beta} = Y\}$ is typically an affine set of dimension $(d-n)k$. On the other hand, since we also reparametrize $\boldsymbol{\beta}$ as a linear network of width $l \gg d$, the manifold of interpolators in the reparametrized space of $\boldsymbol{\theta}$, defined by $\mathcal{I}_{\boldsymbol{\theta}} := \{\boldsymbol{\theta} = (\boldsymbol{W}_1, \boldsymbol{W}_2) \mid \boldsymbol{W}_1\boldsymbol{W}_2 \in \mathcal{I}_{\boldsymbol{\beta}}\}$, is of dimension $l(d+k) - nk$. A natural question, therefore, is to which of these interpolators $\boldsymbol{\theta}^* \in \mathcal{I}_{\boldsymbol{\theta}}$ does a given optimization algorithm converge. This concept is referred to as the *implicit bias* of an algorithm. The aim of this work is to study that of gradient flow.

**Gradient Flow.** The dynamics induced in parameter space by running gradient flow on (2.2) is given by

$$\dot{\boldsymbol{\theta}} = -\nabla_{\boldsymbol{\theta}} \mathcal{L}(\boldsymbol{\theta}). \tag{2.3}$$

We wish to describe the implicit regularization properties of this continuous-time process, which is the vanishing stepsize limit of (stochastic) gradient descent. While the latter algorithms incur additional regularization properties from using non-zero stepsizes [Keskar et al., 2017], the study of GF is an important stepping stone to understanding the implicit bias of gradient-based methods in practice. In terms of $\boldsymbol{W}_1, \boldsymbol{W}_2$ the dynamics translates to

$$\dot{\boldsymbol{W}}_1 = X^\top \left(Y - X\boldsymbol{W}_1\boldsymbol{W}_2\right) \boldsymbol{W}_2^\top , \tag{2.4a}$$

$$\dot{\boldsymbol{W}}_2 = \boldsymbol{W}_1^\top X^\top \left(Y - X\boldsymbol{W}_1\boldsymbol{W}_2\right). \tag{2.4b}$$

We emphasize a crucial point: even if the function $\boldsymbol{\beta} \to \|X\boldsymbol{\beta} - Y\|^2$ is convex, its reparametrization in terms of $\boldsymbol{W}_1, \boldsymbol{W}_2$ is not. Non-convexity and non-linearity makes the analysis challenging and a priori it is not even clear whether the time evolution of $\boldsymbol{\beta}$ can be expressed as a closed system.

**Initialization.** One of our primary objects of study is the impact of initialization on the behaviour of GF. We describe here our initialization choice, to which we henceforth refer as $I_\gamma$.

(a) **Orthogonal feature layer:** We initialize the inner layer such that the rows of $\boldsymbol{W}_1$ are orthogonal and scale with parameter $\gamma > 0$. Mathematically, this translates to $\boldsymbol{W}_1(0) = \sqrt{2\gamma}P$ for $P \in \mathbb{R}^{d \times l}$ in the Stiefel manifold $V_d(\mathbb{R}^l) := \{P \in \mathbb{R}^{d \times l}, \text{ such that } PP^\top = I_d\}$. Initializing with an orthogonal matrix is studied by Pennington et al. [2018], Hu et al. [2020], however from an optimization perspective. Note that when $l$ is very large, this setting approximates the real-world scenario of initializing the hidden neurons with $d$ i.i.d. Gaussian vectors in $\mathbb{R}^l$, which are known to be almost orthogonal.

(b) **Zero weight layer:** In order to remove any initialization bias from the linear layer, we initialize it at $\boldsymbol{W}_2(0) = 0$. This can be seen as the limiting case of initializing the weight layer with a very small *relative* scale $\overline{\gamma} \ll \gamma$.

As already mentioned in Section 1.1, existing studies on linear networks assume a "zero-balance initialization", namely that $\boldsymbol{W}_1^\top(0)\boldsymbol{W}_1(0) = \boldsymbol{W}_2(0)\boldsymbol{W}_2^\top(0)$ [Saxe et al., 2014, Arora et al., 2019b, Azulay et al., 2021]. This condition introduces the invariant $\boldsymbol{W}_1^\top\boldsymbol{W}_1 = \boldsymbol{W}_2\boldsymbol{W}_2^\top$ [Du et al., 2018], which holds for all times $t \geq 0$. This balancedness can be seen as a degeneracy assumption on the flow, since it implies that $\boldsymbol{W}_1$ has at most rank $k$ during the entire process, irrespective of the scale of initialization $\gamma$. In contrast, we show that *depending on* $\gamma$ the feature layer $\boldsymbol{W}_1$ is biased (or not) toward a low-rank predictor, thus unveiling a truly rich representation learning regime.

## 3 Main result: implicit bias and dynamics description

### 3.1 Implicit bias on the parameters

Non-convex gradient flows are generally not guaranteed to reach *global* minimizers of the objective and even when they do, such results are difficult to formally prove. Moreover, the existence of many zero-loss solutions with different generalization properties raises the question of which interpolating

network is yielded by training. An elegant answer to such questions is to express the resulting predictor as the *optimum* among all the possible interpolators of some new, a priori unspecified cost. In addition to the descriptive power of such variational formulations, they express a form of capacity control over the estimator which can be further used to describe its generalization abilities [Bartlett et al., 2020]. The following theorem adds to this series of works, by precisely deriving such a characterization for GF in the setting of linear networks.

**Theorem 3.1.** *Let $(\boldsymbol{W}_1, \boldsymbol{W}_2)$ be the process that follows the GF equations* (2.4a)-(2.4b)*, initialized according to condition $I_\gamma$, for some $\gamma > 0$. Then*

*(i) The parameters converge to a global optimum of the loss*

$$\lim_{t \to \infty} (\boldsymbol{W}_1(t), \boldsymbol{W}_2(t)) = (\boldsymbol{W}_1^\infty, \boldsymbol{W}_2^\infty) \in \mathcal{I}_{\boldsymbol{\theta}}.$$

*(ii) The linear predictor $\boldsymbol{\beta}$ converges to the minimum $\ell_2$-norm interpolator*

$$\lim_{t \to \infty} \boldsymbol{\beta}(t) = \operatorname*{argmin}_{X\boldsymbol{\beta}=Y} \ \left\|\boldsymbol{\beta}\right\|_F \stackrel{\text{def}}{=} \boldsymbol{\beta}_*.$$

*(iii) We have the following variational characterization of the limiting parameters*

$$(\boldsymbol{W}_1^\infty, \boldsymbol{W}_2^\infty) \in \operatorname*{argmin}_{X\boldsymbol{W}_1\boldsymbol{W}_2=Y} \frac{1}{2}\left\|\boldsymbol{W}_2\right\|_F^2 + \frac{1}{2}\left\|\boldsymbol{W}_1\right\|_F^2 - \gamma \log\left(\det\left(\boldsymbol{W}_1\boldsymbol{W}_1^\top\right)\right). \quad (3.1)$$

**Interpretation of the theorem.** The theorem is divided into three parts which state that (i) the matrices converge to a zero loss solution, which is a priori non-trivial since the loss in non-convex; (ii) among all the interpolators in $\mathcal{I}_{\boldsymbol{\beta}}$, $\boldsymbol{\beta}$ converges to the minimum $\ell_2$-norm interpolator for all $\gamma > 0$; and (iii) among all the interpolators in $\mathcal{I}_\theta$, $(\boldsymbol{W}_1, \boldsymbol{W}_2)$ converge to the ones that minimize a $\gamma$-dependent potential. To fully capture the richness of this result, we observe that in the limit of $\gamma \to 0$, problem (3.1) informally translates to [Attouch, 1996]

$$\lim_{\gamma \to 0} (\boldsymbol{W}_1^\infty, \boldsymbol{W}_2^\infty) \in \operatorname*{argmin}_{X\boldsymbol{W}_1\boldsymbol{W}_2=Y} \frac{1}{2}\left\|\boldsymbol{W}_2\right\|_F^2 + \frac{1}{2}\left\|\boldsymbol{W}_1\right\|_F^2.$$

This is equivalent, in the space of linear predictors $\boldsymbol{\beta}$ to the minimum nuclear norm solution $\boldsymbol{\beta} \in \operatorname{argmin}_{X\boldsymbol{\beta}=Y} \left\|\boldsymbol{\beta}\right\|_*$ (which is also the minimum $\ell_2$-norm interpolator for the problem we study). We informally derived this interpretation by taking *first the limit $t \to \infty$ and only after $\gamma \to 0$*. The theorem naturally does not hold if the two limits are reversed, since $\gamma = 0$ places the initialization at a saddle point of the loss, which is a stationary point of the flow.

With increasing $\gamma$, we move towards solutions with a large $\log \det (\boldsymbol{W}_1\boldsymbol{W}_1^\top)$, which is a smooth approximation of the rank [Fazel et al., 2003]. Intuitively, this means that solutions with increasing rank are preferred as $\gamma$ grows. This scale-induced implicit bias is reminiscent of the *rich* and *lazy* regimes [Chizat et al., 2019, Woodworth et al., 2020], albeit visible in the space of representations rather than in that of predictors. As such, our result for linear networks is akin to Woodworth et al. [2020]'s, which characterizes the rich and lazy regimes for simpler diagonal linear networks.

**Comparison with works on implicit bias of $\boldsymbol{\beta}$.** Azulay et al. [2021], Min et al. [2021] also study the implicit bias phenomenon in linear networks, however, these results only address the structure of the final predictor $\boldsymbol{\beta}$ and not that of the factorized problem $(\boldsymbol{W}_1, \boldsymbol{W}_2)$. As shown in Theorem 3.1, these works fall short of unveiling all the nuances of the implicit regularization induced by GF in the case of linear networks.

To give a precise example, consider the simplest case of scalar regression ($k = 1$) for which both Theorem 3.1 and [Azulay et al., 2021] show that $\boldsymbol{\beta}$ is biased towards low-$\ell_2$ interpolators. This view is not complete, since there exist many pairs $(\boldsymbol{W}_1, \boldsymbol{W}_2)$ such that $\boldsymbol{W}_1\boldsymbol{W}_2 = \boldsymbol{\beta}$. Theorem 3.1 goes one step further and provides variational characterization of $(\boldsymbol{W}_1, \boldsymbol{W}_2)$ at convergence. Moreover, it shows that when $\gamma \to 0$ all columns of $\boldsymbol{W}_1$ align in the direction of $\boldsymbol{\beta}$, thus creating a rank one hidden layer. This is an example of rich representation learning, where $\boldsymbol{W}_1$ is learning the only feature needed to make a prediction.

**Implicit bias of the singular values.** Proceeding with the description of the spectral bias, we provide a characterization of $W_1$'s limiting *singular values* which highlights their dependence on $\gamma$.

**Corollary 3.1.** *[Singular values at the limit] Using the same quantities as in Theorem 3.1, denote* $(\sigma_1(W_1^\infty), \ldots, \sigma_d(W_1^\infty))$ *and* $(\sigma_1(\boldsymbol{\beta}_*), \ldots, \sigma_k(\boldsymbol{\beta}_*))$ *the singular values of* $W_1^\infty, \boldsymbol{\beta}_*$ *are*

$$\sigma_i\left(W_1^\infty\right) = \left(\sqrt{\sigma_i\left(\boldsymbol{\beta}_*\right)^2 + \gamma^2} + \gamma\right)^{1/2}, \text{ for } 1 \leq i \leq k.$$

$$\sigma_i\left(W_1^\infty\right) = (2\gamma)^{1/2}, \text{ for } k < i \leq d.$$

**Discussion.** Similar expressions for the singular values of $W_2$ can be derived and are deferred to the Appendix B due to lack of space. In the vanishing initialization limit, only the first $k$ singular values are activated and $W_1$ resembles a rank $k$ matrix. Conversely, for large $\gamma$ all the singular values are approximately equal in scale and $W_1$ resembles an isotropic full-rank matrix.

To ease interpretation, we again focus on the case of scalar regression. Corollary 3.1 shows that, for small $\gamma$, only one singular value grows while the others remain small constants. More precisely, when $\gamma \sim o(\|\boldsymbol{\beta}_*\|)$, the training model is approximately rank one, with only one spiked singular value. Conversely, when $\gamma \sim \Omega(\|\boldsymbol{\beta}_*\|)$ the low-rank structure disappears. This result perfectly captures the rich learning regime at low initialization where the hidden layer *learns* the defining feature of the problem, whereas in the lazy regime (large $\gamma$) the singular values of the matrix hardly move and no structure is present in $W_1$.

Our analysis removes the assumptions of balanced/spectral initialization and whitened data of previous works studying the evolution of singular values [Saxe et al., 2014, Gidel et al., 2019], thus allowing us to reveal the dependence on the scale of initialization.

### 3.2 Description of the dynamics

So far we have described the structure of the parameters of the neural network *at convergence*. Here, we show that the dynamics of the singular values of $\boldsymbol{\beta}$ enjoy a very particular property: it satisfies a *mirror flow* [Alvarez et al., 2004] with a mirror potential that can be written explicitly.

**Theorem 3.2.** *[Dynamics of the flow] With the same notations as in Theorem 3.1,*

(a) ***Mirror on singular values:** The singular values of $\boldsymbol{\beta}$, denoted by $\mathbf{D}_{\boldsymbol{\beta}}$, follow the mirror flow*

$$\mathrm{d}\nabla\Psi_\gamma\left(\mathbf{D}_{\boldsymbol{\beta}}\right) = -\nabla_{\mathbf{D}_{\boldsymbol{\beta}}}\mathcal{L}\,\mathrm{d}t,$$

*where the potential writes* $\Psi_\gamma\left(\mathbf{D}_{\boldsymbol{\beta}}\right) := \mathrm{tr}\left(\frac{1}{2}\mathbf{D}_{\boldsymbol{\beta}}\sinh^{-1}\left(\mathbf{D}_{\boldsymbol{\beta}}/\gamma\right) - \sqrt{\mathbf{D}_{\boldsymbol{\beta}}^2 + \gamma^2}\right).$

(b) ***Mirror on $\boldsymbol{\beta}$.** If $k = 1$, the dynamics of $\boldsymbol{\beta}$ can be characterized as a mirror flow*

$$\mathrm{d}\nabla\psi_\gamma\left(\boldsymbol{\beta}\right) = -\left[\gamma + \sqrt{\|\boldsymbol{\beta}\|^2 + \gamma^2}\right]^{1/2}\nabla\mathcal{L}\left(\boldsymbol{\beta}\right)\,\mathrm{d}t, \tag{3.3}$$

*where the potential writes* $\psi_\gamma\left(\boldsymbol{\beta}\right) := \frac{2}{3}\left[\sqrt{\|\boldsymbol{\beta}\|^2 + \gamma^2} + \gamma\right]^{3/2} - 2\gamma\left[\sqrt{\|\boldsymbol{\beta}\|^2 + \gamma^2} + \gamma\right]^{1/2}.$

**Mirror on Singular Values.** For vector regression, GF on the parameters induces a continuous-time mirror descent (which we also refer as mirror flow) with the hyperbolic entropy function [Ghai et al., 2020]. This extends Arora et al. [2019b]'s characterization of the evolution of singular values when the initialization is balanced. Note that our result does not *fully characterize* the evolution of the system, since the characterization of the singular vectors is absent. Still, some interesting comments can be made. In the rich regime in which $\gamma \to 0$, the hyperbolic entropy $\Psi_\gamma \sim (-\ln\gamma)\|\boldsymbol{\beta}\|_*$. Thus, informally, for small $\gamma$ the gradient flow on parameters is approximately equivalent to a mirror flow on the nuclear norm. This is reminiscent of the case of diagonal linear networks where such an equivalence is proven rigorously [Pesme and Flammarion, 2023] and is known to lead to an incremental saddle-to-saddle dynamics [Li et al., 2021, Jacot et al., 2022, Berthier, 2022].

In the case of whitened data, i.e., $X^\top X = I$, we show that the singular vectors of $\boldsymbol{\beta}$ are stationary (see Appendix C.11 for details). Therefore, the mirror flow on the singular values characterizes the entire system. In this case, we can even provide an exact expression for the evolution of the singular values

(Appendix C.12) by solving a matrix Ricatti equation [Bittanti et al., 1991]. In the limit of $\gamma \to 0$, we can show that, beyond the case of balanced initialization [Saxe et al., 2014, Gidel et al., 2019], the singular values are learned in a hierarchical manner. When $\gamma \to 0$ and with appropriately rescaled time, the limiting trajectory for the $i^{th}$ singular value $\sigma_{i,\boldsymbol{\beta}}$ can be seen as the *jump process*

$$\sigma_{i,\boldsymbol{\beta}}\left(\ln\left(\frac{1}{\gamma}\right)t\right) = \sigma_{i,\boldsymbol{\beta}_*}\mathbb{1}\left(t > \frac{1}{2\sigma_{i,\boldsymbol{\beta}_*}}\right),$$

where $\sigma_{i,\boldsymbol{\beta}_*}$ is the $i^{th}$ singular value of $\boldsymbol{\beta}^*$. Each singular value is activated at time $-\ln(\gamma)(2\sigma_{i,\boldsymbol{\beta}_*})^{-1}$. Therefore, we observe an incremental learning process, where the activation begins with the largest singular value and proceeds accordingly.

**Mirror descent for scalar regression.** The result (b) states that the GF on the parameters $(\boldsymbol{W}_1, \boldsymbol{W}_2)$ implies a mirror flow on the predictor $\boldsymbol{\beta}$ with the potential $\psi_\delta$. To be more precise, the evolution is governed by a mirror flow with the time scaled as a function of $\|\boldsymbol{\beta}\|$. This technique of time-warping was proposed in Azulay et al. [2021] for the case of a linear network with a single neuron ($l = 1$) with balanced initializations. In contrast, with a specific initialization shape, we show the existence of a mirror flow for an arbitrary number of neurons and unbalanced initialization of any scale. The existence of a mirror flow is surprising since the reparametrization defining linear networks is not commutative in general [Li et al., 2022]. However, due to the specific initialization we use, this problem can be circumvented by preserving certain commutative properties.

The equivalence with mirror descent enables us to show that $\mathcal{L}(\boldsymbol{\beta}(t)) = O(1/\gamma t)$ (see Appendix C.8), thus providing a convergence rate for the training loss independent of the conditioning of data, in contrast to Min et al. [2021], Du et al. [2019b]. Note that with decreasing initialization scale $\gamma$, the convergence speed diminishes, while according to the results in Theorems 3.1 better implicit bias is achieved. This suggests the existence of a trade-off between optimization and implicit bias already observed in several works [Woodworth et al., 2020], where achieving better quality solutions is linked to slower optimization. In contrast to this behaviour, for the case of balanced initialization [Braun et al., 2022] emphasizes a decoupling between the learning speed and the quality of solutions. Conversely, we stress that in the general setting (e.g., under imbalance) such a decoupling is absent.

### 3.3 Sketch of the proofs

In this section, we give a short description of the proofs of the main results from the previous sections. The common theme of the following intermediate results is to identify natural invariants of the dynamics, which can be leveraged to understand the hidden mirror structure of the flow.

**Lemma 3.1.** *Consider the dynamics of the gradient flow (2.4) initialized at $(\boldsymbol{W}_1(0), \boldsymbol{W}_2(0)) = (\sqrt{2\gamma}P, 0)$. Let $\boldsymbol{Z}_1 := \boldsymbol{W}_1 P^\top, \boldsymbol{Z}_2 := P\boldsymbol{W}_2$ and the residual $\boldsymbol{R} := X^\top(Y - X\boldsymbol{Z}_1\boldsymbol{Z}_2)$, then the evolution of $(\boldsymbol{Z}_1, \boldsymbol{Z}_2)$ is governed by the following ODE*

$$\dot{\boldsymbol{Z}}_1 = \boldsymbol{R}\boldsymbol{Z}_2^\top \quad , \quad \dot{\boldsymbol{Z}}_2 = \boldsymbol{Z}_1^\top \boldsymbol{R}. \tag{3.4}$$

*Furthermore, the dynamics of gradient flow (2.4) is equivalent to (3.4), i.e., $(\boldsymbol{W}_1(t), \boldsymbol{W}_2(t)) = (\boldsymbol{Z}_1(t)P, P^\top \boldsymbol{Z}_2(t))$ at any time $t$.*

Lemma 3.1 derives an equivalent dynamics to equations (2.4). It shows that weights $\boldsymbol{W}_1^\top, \boldsymbol{W}_2$ always stay in the column span of the initialization $P$, thus restricting their evolution to a subspace. Going forward, we derive the invariants of the dynamics (3.4).

**Lemma 3.2.** *For the projected matrices given in (3.4), we have the following invariant,*

$$\boldsymbol{Z}_1^\top \boldsymbol{Z}_1 - \boldsymbol{Z}_2\boldsymbol{Z}_2^\top = 2\gamma\mathbf{I}.$$

This invariant ensures that $\boldsymbol{Z}_1^\top \boldsymbol{Z}_1$ and $\boldsymbol{Z}_2\boldsymbol{Z}_2^\top$ commute which is a crucial ingredient in the proofs of Theorems 3.1, 3.2. Now, we derive the evolution of $\boldsymbol{\alpha} := \boldsymbol{Z}_1^{-\top}\boldsymbol{Z}_2$, which turns out to be the central quantity enabling our result. The lemma below describes certain properties of the evolution of $\boldsymbol{\alpha}$.

**Lemma 3.3.** *Let $\boldsymbol{\alpha} := \boldsymbol{Z}_1^{-\top}\boldsymbol{Z}_2$, we have the following time evolution of parameters:*

$$\dot{\boldsymbol{\alpha}} = \boldsymbol{R} - \boldsymbol{\alpha}\boldsymbol{R}^\top\boldsymbol{\alpha}, \quad and \quad \boldsymbol{\beta} = \left(1 - \boldsymbol{\alpha}\boldsymbol{\alpha}^\top\right)^{-1}\boldsymbol{\alpha}.$$

**An outline of the proof of Theorem 3.1.** With an *ansatz* on the potential that it is decomposable in terms of $\boldsymbol{Z}_1, \boldsymbol{Z}_2$, we derive KKT conditions for the constrained optimization problem

$$\underset{X\boldsymbol{Z}_1\boldsymbol{Z}_2=Y}{\operatorname{argmin}} \ \psi_1(\boldsymbol{Z}_1) + \psi_2(\boldsymbol{Z}_2).$$

Using Lemmas 3.3, we show that $\boldsymbol{\alpha}$ stays in $\operatorname{span}(X)$. We use the isotropic property of the imbalance from Lemma 3.2 to find appropriate functions $\psi_1, \psi_2$ and finally prove Theorem 3.1. The proofs for theorem 3.1, 3.2 and corollary 3.1 can be found in Appendix B.

## 4 Further thoughts and perspectives

The previous section provided a deep-dive into the dynamics of the gradient flow, which we complement here with a few steps in the direction of understanding the dynamics with stochastic gradients. We investigate stochastic gradient descent (SGD) by studying its simpler counterpart, label noise gradient descent (LNGD) Blanc et al. [2020].

### 4.1 The role of noise

It was observed that the noise in stochastic gradient descent has a parameter-dependent shape that induces certain regularization properties [HaoChen et al., 2021, Blanc et al., 2020]. Here, we study the properties of the noise shape induced in the case of parameterization with linear neural networks.

Inspired from the analysis of HaoChen et al. [2021] and the large noise regime described by Pillaud-Vivien et al. [2022] in the context of diagonal neural networks, we design a process driven purely by noise and which carries the same geometric properties as SGD's noise. We consider the scalar ($k = 1$) regression problem with $l = d$ and, through an abuse of notation, denote $\boldsymbol{W}_1 = \boldsymbol{W}$, and $\boldsymbol{W}_2 = \mathbf{a}$. The noise-driven process which we consider is:

$$\mathrm{d}\boldsymbol{W} = (\mathrm{d}\boldsymbol{B}_t)\,\mathbf{a}^\top \quad , \quad \mathrm{d}\mathbf{a} = \boldsymbol{W}^\top \mathrm{d}\boldsymbol{B}_t, \tag{4.1}$$

where $\boldsymbol{B}_t$ is a $d$-dimensional Brownian motion. Details on how this SDE captures the noise of SGD are deferred to Appendix D. We show that, similarly to the rich regime of the gradient flow ($\gamma \to 0$), this noise also carries a rich spectral bias *but for any initialization*. Indeed, we have the following result on the SDE dynamics. The proof can be found in Appendix D.

**Proposition 4.1.** *The dynamics* (4.1) *has the following convergence properties*

(a) **Variance explosion**. *The variance of the norms of* $\boldsymbol{W}$, $\mathbf{a}$ *explode, i.e.,*

$$\lim_{t\to\infty} \mathbb{E}\left[\left\|\boldsymbol{W}(t)\right\|^2\right] \to \infty \quad , \quad \lim_{t\to\infty} \mathbb{E}\left[\left\|\mathbf{a}(t)\right\|^2\right] \to \infty.$$

(b) **Scale divergence**. *For $d \geq 5$, for any $\alpha > 0$, we have that,*

$$\lim_{t\to\infty} \mathbb{E}\left[\left\|\boldsymbol{W}(t)\right\|^\alpha\right] \to \infty \quad , \quad \lim_{t\to\infty} \mathbb{E}\left[\left\|\mathbf{a}(t)\right\|^\alpha + \left\|\bar{\mathbf{a}}(t)\right\|^\alpha\right] \to \infty.$$

*where $\bar{\mathbf{a}} := e^{-t}\int\limits_0^t e^s\mathbf{a}(s)\mathrm{d}s$ is the exponential moving average of $\mathbf{a}$.*

(c) **Alignment - spectral bias**. *Denote the $i^{th}$ row of $\boldsymbol{W}$ as $\boldsymbol{w}_i$. Using $[\boldsymbol{w}_i, \mathbf{a}] \overset{\text{def}}{=} \boldsymbol{w}_i\mathbf{a}^\top - \mathbf{a}\boldsymbol{w}_i^\top$,*

$$\lim_{t\to\infty} \mathbb{E}\left[|[\boldsymbol{w}_i, \mathbf{a}]|\right] \to 0.$$

For any two vectors $u, v$, $[u, v]$ denotes the commutator of the vectors: remark that if $[u, v] = 0$, then $u, v$ are aligned, i.e $u = cv$, for some scalar $c \in \mathbb{R}$. First, notice that for $d = 1$, the SDE in fact corresponds to the geometric Brownian motion with no drift and the dynamics collapses to zero [Oksendal, 2013]. For dimension $d \geq 2$, the proposition states that the system diverges and the weights grow towards infinity. However, despite the fact that the norm grows, the commutator $[\boldsymbol{w}_i, \mathbf{a}]$ goes to zero, indicating that all the rows of $\boldsymbol{w}_i$ align towards $\mathbf{a}$. Overall, similarly to the gradient flow in the rich regime, this induces a low rank structure in $\boldsymbol{W}$. This phenomenon can be further seen through the evolution of singular values, where the top singular value of $\boldsymbol{W}$ grows unboundedly, whereas the remaining singular values decay to 0 as depicted in Figure 3a in the Appendix. This sheds some light on how SGD induces a particular parameter-dependent noise which implicitly biases the solutions towards having a low-rank structure of the hidden layer [Andriushchenko et al., 2022].

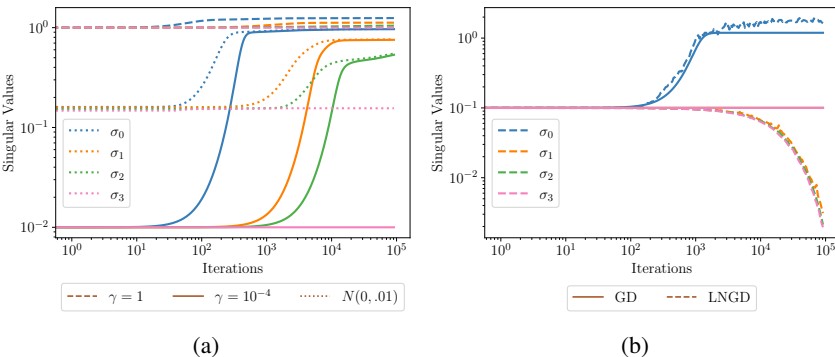

Figure 1: (a) Vector regression with orthogonal initialization and scales $\gamma = 1, 10^{-4}$ and Gaussian initialization with entries from $N(0, 0.01)$ (b) Scalar regression with Gradient Descent (GD) and Label Noise Gradient Descent (LNGD).

**Intricate dynamics in presence of drift.** Proposition 4.1 focuses on the process that is solely driven by noise. However, in general SGD also encompasses a drift term which corresponds to the dynamics studied in Section 3. The continuous-time SDE describing the process is

$$\mathrm{d}\boldsymbol{W} = -\nabla_{\boldsymbol{W}}\mathcal{L}\left(\boldsymbol{W}, \mathbf{a}\right)\mathrm{d}t + \delta\left(\mathrm{d}\boldsymbol{B}_t\right)\mathbf{a}^{\top} \quad , \quad \mathrm{d}\mathbf{a} = -\nabla_{\mathbf{a}}\mathcal{L}\left(\boldsymbol{W}, \mathbf{a}\right)\mathrm{d}t + \delta\boldsymbol{W}^{\top}\mathrm{d}\boldsymbol{B}_t,$$

where $\delta > 0$ indicates the scale of the noise. The presence of drift quickly complicates the analysis, but intuitively, the noise simplifies the model by inducing a rank reduction, whereas the drift terms prevent the weights from growing unbounded. This noise-driven mechanism relaxes the role of initialization. Empirically this is illustrated in Figure 1b. Gradient descent already exhibits a regularization effect as it increases only one singular value while keeping the others constant. However, gradient descent with label noise [Blanc et al., 2020] enhances this regularization effect by decaying the singular values and promoting low-rank representations. As a result, even for larger initialization scales, we observe the presence of low-rank structures in the hidden layer, unlike in gradient descent. The precise characterization of the this phenomenon is left for future research.

**Experiments.** We consider a regression problem on synthetic data, with $n = 5$ samples of Gaussian data in $\mathbb{R}^{10}$ ($d = 10$) and the labels in $\mathbb{R}^3$ ($k = 3$) generated by a ground truth $\boldsymbol{\beta}_* \in \mathbb{R}^{d \times k}$ . We consider a network with width $l = 200$. In Figure 1a, we show the evolution of the top-4 singular values of the hidden layer $\boldsymbol{W}_1$. We use orthogonal initialization for the network with the two scales of initialization $\gamma = 1, 10^{-4}$. Note that, as depicted by Corollary 3.1, for the smaller scale only the first $k = 3$ singular values are significant in comparison to the remaining $d - k$ singular values. This shows that the matrix is approximately rank $k$ and the neurons align along three directions. In contrast, for the larger scale $\gamma = 1$, the final weight matrix has rank $d$. To complement this, we also consider a Gaussian initialization with variance $0.01$ – specifically, we initialize the inner layer with $d = 10$ Gaussian random vectors in $\mathbb{R}^l$. As described when $l \gg d$, the initialization is close to the orthogonal initialization. Hence, in this case, we can see that only $k$ singular values grow and the final model has an approximately rank $k$ hidden layer. In figure 1b, we depict the time evolution of singular values for GD and LNGD on a scalar regression problem with orthogonal data in $\mathbb{R}^5$ ($n, d = 5$) and a network with $l = 200$. Further details on hyper-parameters can be found in the Appendix.

**Extension to non-linear activations.** Huh et al. [2023], Andriushchenko et al. [2022] empirically demonstrate a low-rank phenomenon through extensive experiments on deep networks with non-linear activations. However, a comprehensive theoretical comprehension of this behavior remains elusive, despite some efforts addressing these issues [Boursier et al., 2022]. To show that our analysis extends beyond linear activations, we present a toy experiment for ReLU networks (see Figure 2 and further details in Figure 5). Consider a scalar regression problem in a ReLU teacher-student setup. We generate a training set of size 10 sampled from a random Gaussian distribution in $\mathbb{R}^5$. The training data $(x_i, y_i)_{i=1}^{10} \in \mathbb{R}^5 \times \mathbb{R}$ is generated by a teacher ReLU network with 2 neurons $(w_0, w_1)$, i.e.,

$$y_i = a_0\sigma(w_0^{\top}x_0) + a_1\sigma(w_1^{\top}x_1),$$

where $\sigma$ is the ReLU non-linearity. We train a student network with 20 hidden neurons. Note that there are two relevant directions $w_0, w_1$ for the student network to learn, therefore we expect the hidden layer to represent these two directions (i.e., a rank 2 hidden layer, and a singular value decomposition with two non-zero singular values). This property is empirically verified in Figure 2. We plot the time evolution of singular values and when initialized at low-scale the network converges to an approximately rank-2 matrix. When initialized at a larger scale, the network weight matrix is high rank and the neurons do not learn the teacher directions.

**Perspectives.** Learning representations which can be transferred to downstream tasks is a key attribute for the success of deep learning [Bengio et al., 2013, LeCun et al., 2015]. In this work, we present an archetypal problem where for the same predictor in functional space, there exist multiple representations in parameter space, some of which can exhibit a rich structure. This scenario presents a case for going beyond the characterization of implicit bias in the functional space [Parhi and Nowak, 2022] and further studying the implicit bias in the parameter space. Such characterizations facilitate the identification of crucial ingredients in training algorithms that enable effective feature learning.

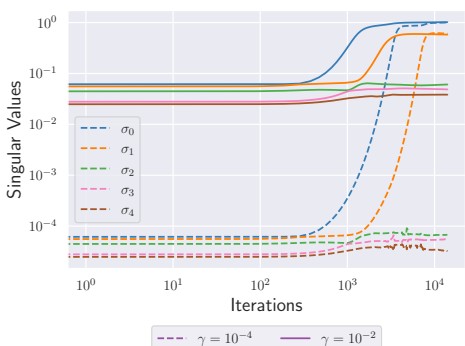

Figure 2: The time evolution of singular values of the hidden layer weights of a 2-layer ReLU network when trained with gradient flow initialized with Gaussian random variables with different scales. We consider a scalar regression problem in a teacher-student setup.

**Limitations and Future Work.** This paper tackles the phenomenon of implicit bias, with the aim of furthering the understanding of how neural networks learn in practice. Unfortunately, practical models are highly nonlinear due to their activations and rely on various heuristics to achieve state-of-the-art performance, thus being difficult to grasp mathematically. This work therefore studies the simplified setting of two-layer linear neural networks. In terms of the assumptions we make, the orthogonality of initialization is only approximately faithful to practical settings where small random weights are used. Nevertheless, we are confident that this requirement can be loosened through a perturbation analysis in the vein of Gidel et al. [2019]. Finally, our dynamical description of the system is yet to be completed in the vector regression case with non-whitened data. A careful set of assumptions is necessary here, and hopefully ones that are weaker than the restricted isometry property used in related works [Li et al., 2018]. Finally, we only partially describe the dynamics in the presence of stochastic noise and giving a full characterization remains a desired objective of future investigations. Further discussion on these aspects is presented in Appendix C.1.

## Acknowledgments and Disclosure of Funding

AV is supported by Swiss data science fellowship.

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

## Organization

The appendix is organized as follows,

- In section A, we present the experiment details.

- In section B, we present the proof of Theorems 3.1, 3.2 and Corollary 3.1.

- section C contains the proofs of the supporting lemmas.

- In the final section D, we discuss our choice on the noise model and present the proof of Proposition 4.1.

- For the results referenced in the main section, the convergence rate of mirror flow can be found at C.8, the time evolution of singular values and their limiting jump process is available at C.12, the stationarity of singular vectors in the orthogonal case at C.11, the discussion on the noise model at D.

**Notations.** For matrices of appropriate dimensions, we use $[A, B]$ to denote $AB - BA$.

## A  Experiment Details

**Experiments.** We discretize the SDE (4.1) with a step-size $\sim 1/\sqrt{t}$. We simulate three parallel runs and track the evolution of singular values and the evolution of alignment using the commutator of the row $\boldsymbol{w}_i, \mathbf{a}$, i.e., $[\boldsymbol{w}_i, \mathbf{a}] \coloneqq (\boldsymbol{w}_i \mathbf{a}^\top - \mathbf{a} \boldsymbol{w}_i^\top)$. We consider the SDE for dimemsion $d = 2$. The evolution is initialized at $\boldsymbol{W}(0) = \mathbf{I}_2$ and $\mathbf{a}(0) = 0$. As seen in Figure 3a, the noise shape inherently induces a low-rank structure where it intensifies a singular value and significantly diminishes the other singular value. As predicted by our proposition (4.1), figure 3b shows that the rows of $\boldsymbol{W}$ align with $\mathbf{a}$, thus giving a rank 1 structure. The experiments were run on a 16-GB RAM Apple M1 mac with OS Ventura 13.3.1.

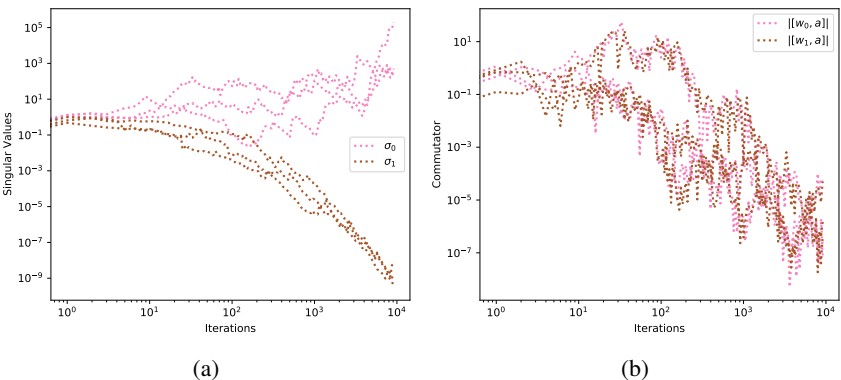

(a)                              (b)

Figure 3: Three parallel runs of the noise dynamics Eq. (4.1) for $d = 2$. (a) The evolution of singular values with $\sigma_0$ increasing and $\sigma_1$ decaying. (b) Measuring the norm of the commutator again as predicted by Proposition 4.1.

## B  Main Proofs

In this section, we present the proofs of the theorem discussed in Section 3.

**Theorem 3.1.** *Let* $(\boldsymbol{W}_1, \boldsymbol{W}_2)$ *be the process that follows the GF equations* (2.4a)-(2.4b), *initialized according to condition* $I_\gamma$, *for some* $\gamma > 0$. *Then*

*(i) The parameters converge to a global optimum of the loss*

$$\lim_{t \to \infty} (\boldsymbol{W}_1(t), \boldsymbol{W}_2(t)) = (\boldsymbol{W}_1^\infty, \boldsymbol{W}_2^\infty) \in \mathcal{I}_{\boldsymbol{\theta}}.$$

*(ii) The linear predictor $\boldsymbol{\beta}$ converges to the minimum $\ell_2$-norm interpolator*

$$\lim_{t \to \infty} \boldsymbol{\beta}(t) = \operatorname*{argmin}_{X\boldsymbol{\beta}=Y} \left\|\boldsymbol{\beta}\right\|_F \overset{\text{def}}{=} \boldsymbol{\beta}_*.$$

*(iii) We have the following variational characterization of the limiting parameters*

$$(\boldsymbol{W}_1^\infty, \boldsymbol{W}_2^\infty) \in \operatorname*{argmin}_{X\boldsymbol{W}_1\boldsymbol{W}_2=Y} \frac{1}{2}\left\|\boldsymbol{W}_2\right\|_F^2 + \frac{1}{2}\left\|\boldsymbol{W}_1\right\|_F^2 - \gamma \log\left(\det\left(\boldsymbol{W}_1\boldsymbol{W}_1^\top\right)\right). \quad (3.1)$$

*Proof.* We initialize such that $\boldsymbol{W}_1 = \sqrt{2\gamma}P, \boldsymbol{W}_2 = 0$. Lemma 3.1 states that the dynamics of gradient flow is restricted to a subspace and can be equivalently described by,

$$\dot{\boldsymbol{Z}}_1 = \boldsymbol{R}\boldsymbol{Z}_2^\top \quad, \quad \dot{\boldsymbol{Z}}_2 = \boldsymbol{Z}_1^\top\boldsymbol{R}. \quad (B.1)$$

where $\boldsymbol{R} := X^\top(Y - X\boldsymbol{Z}_1\boldsymbol{Z}_2)$.

(i) To show the convergence, we track the evolution of $\operatorname{tr}\left(\boldsymbol{R}^\top\boldsymbol{R}\right)$ and use the following descent inequality to show that it converges to 0. With $\lambda_{\min}(X^TX)$ denoting the smallest eigenvalue of the $X^\top X$, the descent inequality (C.13) is as follows,

$$\overbrace{\operatorname{tr}\left(\boldsymbol{R}^\top\boldsymbol{R}\right)}^{\bullet} \le -2\gamma\lambda_{\min}(X^\top X)\operatorname{tr}\left(\boldsymbol{R}^\top\boldsymbol{R}\right).$$

Refer to Lemma C.6 for the detailed proof.

(ii) To show that $\boldsymbol{\beta} \to \boldsymbol{\beta}_*$ in the limit, we show that $\boldsymbol{\beta} \in span\left(X^\top\right)$, i.e., $\boldsymbol{\beta} = X^\top\lambda$, for some $\lambda$. This satisfies the KKT conditions required for the following minimization problem.

$$\boldsymbol{\beta}_\infty \in \operatorname*{argmin}_{X\boldsymbol{\beta}=Y} \frac{1}{2}\left\|\boldsymbol{\beta}\right\|^2 = \boldsymbol{\beta}_*.$$

The complete proof can be found at Lemma C.6.

(iii) For the limit of the projected dynamics $(\boldsymbol{Z}_1^\infty, \boldsymbol{Z}_2^\infty) := \lim_{t \to \infty}(\boldsymbol{Z}_1(t), \boldsymbol{Z}_2(t))$, Lemma C.4 shows the following,

$$(\boldsymbol{Z}_1^\infty, \boldsymbol{Z}_2^\infty) \in \operatorname*{argmin}_{X\boldsymbol{Z}_1\boldsymbol{Z}_2=y} \frac{1}{2}\left\|\boldsymbol{Z}_2\right\|_F^2 + \frac{1}{2}\left\|\boldsymbol{Z}_1\right\|_F^2 - \gamma \log\left(\det\left(\boldsymbol{Z}_1\boldsymbol{Z}_1^\top\right)\right).$$

Using the transformation from Eq. C.7, we have,

$$\boldsymbol{W}_1 = \boldsymbol{Z}_1 P, \quad \boldsymbol{W}_2 = P^\top\boldsymbol{Z}_2.$$

Thus,

$$\left\|\boldsymbol{W}_1\right\|_F = \left\|\boldsymbol{Z}_1\right\|_F, \quad \left\|\boldsymbol{W}_2\right\|_F = \left\|\boldsymbol{Z}_2\right\|_F,$$
$$\boldsymbol{Z}_1\boldsymbol{Z}_1^\top = \boldsymbol{W}_1\boldsymbol{W}_1^\top, \quad \boldsymbol{Z}_1\boldsymbol{Z}_2 = \boldsymbol{W}_1\boldsymbol{W}_2.$$

Therefore,

$$(\boldsymbol{W}_1^\infty, \boldsymbol{W}_2^\infty) \in \operatorname*{argmin}_{X\boldsymbol{W}_1\boldsymbol{W}_2=Y} \frac{1}{2}\left\|\boldsymbol{W}_2\right\|_F^2 + \frac{1}{2}\left\|\boldsymbol{W}_1\right\|_F^2 - \gamma \log\left(\det\left(\boldsymbol{W}_1\boldsymbol{W}_1^\top\right)\right).$$

This hold on the set $\{(\boldsymbol{W}_1, \boldsymbol{W}_2) : \boldsymbol{W}_1 P_\perp^\top = 0, P_\perp \boldsymbol{W}_2 = 0\}$ which is ensured from gradient flow from Lemma 3.1.

$\square$

**Corollary 3.1.** *[Singular values at the limit] Using the same quantities as in Theorem 3.1, denote $(\sigma_1(\boldsymbol{W}_1^\infty), \ldots, \sigma_d(\boldsymbol{W}_1^\infty))$ and $(\sigma_1(\boldsymbol{\beta}_*), \ldots, \sigma_k(\boldsymbol{\beta}_*))$ the singular values of $\boldsymbol{W}_1^\infty, \boldsymbol{\beta}_*$ are*

$$\sigma_i\left(\boldsymbol{W}_1^\infty\right) = \left(\sqrt{\sigma_i\left(\boldsymbol{\beta}_*\right)^2 + \gamma^2} + \gamma\right)^{1/2}, \textit{ for } 1 \le i \le k.$$
$$\sigma_i\left(\boldsymbol{W}_1^\infty\right) = (2\gamma)^{1/2}, \textit{ for } k < i \le d.$$

*Proof.* Using the transformation $\boldsymbol{W}_1 = \boldsymbol{Z}_1 P$ from Lemma 3.1, we obtain,

$$\boldsymbol{W}_1 \boldsymbol{W}_1^\top = \boldsymbol{Z}_1 \boldsymbol{Z}_1^\top. \tag{B.2}$$

Thus, $\boldsymbol{W}_1$ and $\boldsymbol{Z}_1$ share the same singular values, similarly $\boldsymbol{W}_2$ and $\boldsymbol{Z}_2$ share the same singular values. The expressions for the singular values of $\boldsymbol{Z}_1$ (similarly $\boldsymbol{Z}_2$) can be found at Lemma C.10. This along with the fact that $\boldsymbol{\beta} \to \boldsymbol{\beta}_*$ proves the Corollary 3.1 $\qquad\square$

**Theorem 3.2.** *[Dynamics of the flow] With the same notations as in Theorem 3.1,*

(a) **Mirror on singular values:** *The singular values of $\boldsymbol{\beta}$, denoted by $\mathbf{D_\beta}$, follow the mirror flow*

$$\mathrm{d}\nabla\Psi_\gamma\left(\mathbf{D_\beta}\right) = -\nabla_{\mathbf{D_\beta}}\mathcal{L}\,\mathrm{d}t,$$

*where the potential writes $\Psi_\gamma\left(\mathbf{D_\beta}\right) := \mathrm{tr}\left(\frac{1}{2}\mathbf{D_\beta}\sinh^{-1}\left(\mathbf{D_\beta}/\gamma\right) - \sqrt{\mathbf{D_\beta^2} + \gamma^2}\right).$*

(b) **Mirror on $\boldsymbol{\beta}$.** *If $k = 1$, the dynamics of $\boldsymbol{\beta}$ can be characterized as a mirror flow*

$$\mathrm{d}\nabla\psi_\gamma\left(\boldsymbol{\beta}\right) = -\left[\gamma + \sqrt{\|\boldsymbol{\beta}\|^2 + \gamma^2}\right]^{1/2}\nabla\mathcal{L}\left(\boldsymbol{\beta}\right)\,\mathrm{d}t, \tag{3.3}$$

*where the potential writes $\psi_\gamma\left(\boldsymbol{\beta}\right) := \frac{2}{3}\left[\sqrt{\|\boldsymbol{\beta}\|^2 + \gamma^2} + \gamma\right]^{3/2} - 2\gamma\left[\sqrt{\|\boldsymbol{\beta}\|^2 + \gamma^2} + \gamma\right]^{1/2}.$*

*Proof.* The equivalence with mirror flow for scalar regression is shown in Lemma C.7. The continuous time mirror descent for singular values of $\boldsymbol{\beta}$ is derived in Lemma C.13. $\qquad\square$

## C Supporting Lemmas

This section contains all the technical lemmas and definitions used in the proofs in the section before.

**Lemma 3.1.** *Consider the dynamics of the gradient flow (2.4) initialized at $(\boldsymbol{W}_1(0), \boldsymbol{W}_2(0)) = \left(\sqrt{2\gamma}P, 0\right)$. Let $\boldsymbol{Z}_1 := \boldsymbol{W}_1 P^\top, \boldsymbol{Z}_2 := P\boldsymbol{W}_2$ and the residual $\boldsymbol{R} := X^\top(Y - X\boldsymbol{Z}_1\boldsymbol{Z}_2)$, then the evolution of $(\boldsymbol{Z}_1, \boldsymbol{Z}_2)$ is governed by the following ODE*

$$\dot{\boldsymbol{Z}}_1 = \boldsymbol{R}\boldsymbol{Z}_2^\top \quad, \quad \dot{\boldsymbol{Z}}_2 = \boldsymbol{Z}_1^\top\boldsymbol{R}. \tag{C.1}$$

*Furthermore, the dynamics of gradient flow (2.4) is equivalent to (3.4), i.e., $(\boldsymbol{W}_1(t), \boldsymbol{W}_2(t)) = (\boldsymbol{Z}_1(t)P, P^\top\boldsymbol{Z}_2(t))$ at any time $t$.*

*Proof.* We choose $P_\perp \in \mathbb{R}^{(l-d)\times l}$ such that $P_\perp P^\top = 0$. Using the fact that $P, P_\perp$ orthogonal and span the entire $\mathbb{R}^{l\times l}$,

$$P^\top P + P_\perp^\top P_\perp = \mathbf{I}_l.$$

Denoting $(\boldsymbol{Z}_1)_\perp := \boldsymbol{W}_1 P_\perp^\top, (\boldsymbol{Z}_2)_\perp := P_\perp \boldsymbol{W}_2$, we have,

$$\boldsymbol{W}_1 = \boldsymbol{W}_1\left[P^\top P + P_\perp^\top P_\perp\right] = \boldsymbol{W}_1 P^\top P + \boldsymbol{W}_1 P_\perp^\top P_\perp = \boldsymbol{Z}_1 P + (\boldsymbol{Z}_1)_\perp P_\perp. \tag{C.2}$$

$$\boldsymbol{W}_2 = \left[P^\top P + P_\perp^\top P_\perp\right]\boldsymbol{W}_2 = P^\top P\boldsymbol{W}_2 + P_\perp^\top P_\perp \boldsymbol{W}_2 = P^\top \boldsymbol{Z}_2 + P_\perp^\top (\boldsymbol{Z}_2)_\perp. \tag{C.3}$$

Recalling the evolution of gradient flow (2.4) on the loss

$$\dot{\boldsymbol{W}}_1 = X^\top(Y - X\boldsymbol{W}_1\boldsymbol{W}_2)\boldsymbol{W}_2^\top,$$

$$\dot{\boldsymbol{W}}_2 = \boldsymbol{W}_1^\top X^\top(Y - X\boldsymbol{W}_1\boldsymbol{W}_2).$$

Multiplying the gradient flow updates with $P^\top, P$ from the right, left (resp.) for the above equations

$$\dot{\boldsymbol{W}}_1 P^\top = X^\top(Y - X\boldsymbol{W}_1\boldsymbol{W}_2)\boldsymbol{W}_2^\top P^\top, \quad P\dot{\boldsymbol{W}}_2 = P\boldsymbol{W}_1^\top X^\top(Y - X\boldsymbol{W}_1\boldsymbol{W}_2).$$

Similarly multiplying with $P_\perp^\top, P_\perp$, we have,

$$\dot{\boldsymbol{W}}_1 P_\perp^\top = X^\top(Y - X\boldsymbol{W}_1\boldsymbol{W}_2)\boldsymbol{W}_2^\top P_\perp^\top, \quad P_\perp\dot{\boldsymbol{W}}_2 = P_\perp\boldsymbol{W}_1^\top X^\top(Y - X\boldsymbol{W}_1\boldsymbol{W}_2).$$

Using the above, we have,

$$\boldsymbol{W}_1\boldsymbol{W}_2 = \boldsymbol{W}_1 \left[ P^\top P + P_\perp^\top P_\perp \right] \boldsymbol{W}_2,$$
$$= \boldsymbol{W}_1 P^\top P \boldsymbol{W}_2 + \boldsymbol{W}_1 P_\perp^\top P_\perp \boldsymbol{W}_2.$$

$$\boldsymbol{W}_1\boldsymbol{W}_2 = \boldsymbol{Z}_1\boldsymbol{Z}_2 + (\boldsymbol{Z}_1)_\perp (\boldsymbol{Z}_2)_\perp . \tag{C.5}$$

Rewriting the evolution in terms of $\boldsymbol{Z}_1, \boldsymbol{Z}_2, (\boldsymbol{Z}_1)_\perp, (\boldsymbol{Z}_2)_\perp$,

$$\dot{\boldsymbol{Z}}_1 = X^\top \left( Y - X \left[ \boldsymbol{Z}_1\boldsymbol{Z}_2 + (\boldsymbol{Z}_1)_\perp (\boldsymbol{Z}_2)_\perp \right] \right) \boldsymbol{Z}_2^\top, \quad \dot{\boldsymbol{Z}}_2 = \boldsymbol{Z}_1^\top X^\top \left( Y - X \left[ \boldsymbol{Z}_1\boldsymbol{Z}_2 + (\boldsymbol{Z}_1)_\perp (\boldsymbol{Z}_2)_\perp \right] \right).$$
$$\dot{(\boldsymbol{Z}_1)}_\perp = X^\top \left( Y - X \left[ \boldsymbol{Z}_1\boldsymbol{Z}_2 + (\boldsymbol{Z}_1)_\perp (\boldsymbol{Z}_2)_\perp \right] \right) (\boldsymbol{Z}_2)_\perp^\top, \quad \dot{(\boldsymbol{Z}_2)}_\perp = (\boldsymbol{Z}_1)_\perp^\top X^\top \left( Y - X \left[ \boldsymbol{Z}_1\boldsymbol{Z}_2 + (\boldsymbol{Z}_1)_\perp (\boldsymbol{Z}_2)_\perp \right] \right).$$

This is exactly equivalent to the gradient flow under the linear transformation which maps $\boldsymbol{W}_1, \boldsymbol{W}_2$ to $(\boldsymbol{Z}_1, (\boldsymbol{Z}_1)_\perp), (\boldsymbol{Z}_2, (\boldsymbol{Z}_2)_\perp)$. Now taking into consideration the evolution of $(\boldsymbol{Z}_1)_\perp, (\boldsymbol{Z}_2)_\perp$, we have that $(0, 0)$ is a equilibrium point for the dynamics. From our initialization $I_\gamma$,

$$(\boldsymbol{Z}_1)_\perp \big|_{t=0} = \boldsymbol{W}_1 \big|_{t=0} P_\perp^\top = \sqrt{2\gamma} P P_\perp^\top = 0,$$
$$(\boldsymbol{Z}_2)_\perp \big|_{t=0} = P_\perp \boldsymbol{W}_2 \big|_{t=0} = 0.$$

As we initialized at the equilibrium of the dynamics we have $((\boldsymbol{Z}_1)_\perp, (\boldsymbol{Z}_2)_\perp) = (0, 0)$ for any time $t$. From Eq. (C.5), we have,

$$\boldsymbol{W}_1\boldsymbol{W}_2 = \boldsymbol{Z}_1\boldsymbol{Z}_2. \tag{C.6}$$

The gradient flow (2.4) is equivalent to

$$\dot{\boldsymbol{Z}}_1 = X^\top \left( Y - X \boldsymbol{Z}_1 \boldsymbol{Z}_2 \right) \boldsymbol{Z}_2^\top,$$
$$\dot{\boldsymbol{Z}}_2 = \boldsymbol{Z}_1^\top X^\top \left( Y - X \boldsymbol{Z}_1 \boldsymbol{Z}_2 \right).$$

where $\boldsymbol{Z}_1(0) = \sqrt{2\gamma} P P^\top = \mathbf{I}_d, \boldsymbol{Z}_2(0) = 0$. Furthermore, from Eq. C.2, C.3, we have the following,

$$\boldsymbol{W}_1 = \boldsymbol{Z}_1 P \quad , \quad \boldsymbol{W}_2 = P^\top \boldsymbol{Z}_2. \tag{C.7}$$

This finishes the proof of the lemma. □

**Lemma C.1.** *For the projected matrices given in* (C.1)*, we have the following invariant,*

$$\boldsymbol{Z}_1^\top \boldsymbol{Z}_1 - \boldsymbol{Z}_2 \boldsymbol{Z}_2^\top = 2\gamma \mathbf{I}. \tag{C.8}$$

*Proof.* Recalling the dynamics (C.1)

$$\dot{\boldsymbol{Z}}_1 = \boldsymbol{R} \boldsymbol{Z}_2^\top \quad , \quad \dot{\boldsymbol{Z}}_2 = \boldsymbol{Z}_1^\top \boldsymbol{R}.$$

$$\widehat{\dot{\boldsymbol{Z}_1^\top \boldsymbol{Z}_1}} = (\dot{\boldsymbol{Z}}_1)^\top \boldsymbol{Z}_1 + \boldsymbol{Z}_1^\top (\dot{\boldsymbol{Z}}_1), = \boldsymbol{Z}_2 \boldsymbol{R}^\top \boldsymbol{Z}_1 + \boldsymbol{Z}_1^\top \boldsymbol{R} \boldsymbol{Z}_2^\top.$$

Similarly,

$$\widehat{\dot{\boldsymbol{Z}_2 \boldsymbol{Z}_2^\top}} = \boldsymbol{Z}_2 \boldsymbol{R}^\top \boldsymbol{Z}_1 + \boldsymbol{Z}_1^\top \boldsymbol{R} \boldsymbol{Z}_2^\top.$$

Hence, $\widehat{\dot{\boldsymbol{Z}_1^\top \boldsymbol{Z}_1 - \boldsymbol{Z}_2 \boldsymbol{Z}_2^\top}} = 0$. This implies,

$$\boldsymbol{Z}_1^\top \boldsymbol{Z}_1 - \boldsymbol{Z}_2 \boldsymbol{Z}_2^\top = \left[ \boldsymbol{Z}_1^\top \boldsymbol{Z}_1 - \boldsymbol{Z}_2 \boldsymbol{Z}_2^\top \right] \bigg|_{t=0} = 2\gamma P P^\top = 2\gamma \mathbf{I}_d.$$

□

**Lemma C.2.** *Let* $\boldsymbol{\alpha} := \boldsymbol{Z}_1^{-\top} \boldsymbol{Z}_2$*, we have the following time evolution of parameters:*

$$\dot{\boldsymbol{\alpha}} = \boldsymbol{R} - \boldsymbol{\alpha} \boldsymbol{R}^\top \boldsymbol{\alpha}, \quad \text{and} \quad \boldsymbol{\beta} = 2\gamma \left( \mathbf{I} - \boldsymbol{\alpha} \boldsymbol{\alpha}^\top \right)^{-1} \boldsymbol{\alpha}.$$

*Proof.* Taking the time derivative of $\alpha$,

$$\dot{\alpha} = \dot{Z_1^{-1}} Z_2 + Z_1^{-1} \dot{Z_2} = -Z_1^{-1} \dot{Z_1} Z_1^{-1} Z_2 + Z_1^{-1} Z_1 R,$$
$$= -Z_1^{-1} Z_2 R^\top Z_1^{-1} Z_2 + R,$$
$$= -\alpha R^\top \alpha + R.$$

The evolution of $Z_1 Z_1^\top$, $\alpha \alpha^\top$,

$$\dot{\overline{(Z_1 Z_1^\top)}} = Z_1 Z_1^\top \alpha R^\top + R \alpha^\top Z_1 Z_1^\top, \qquad (C.9)$$

$$\dot{\overline{\alpha \alpha^\top}} = (R - \alpha R^\top \alpha) \alpha^\top + \alpha (R - \alpha R^\top \alpha)^\top,$$
$$= R \alpha^\top - \alpha R^\top \alpha \alpha^\top + \alpha R^\top - \alpha \alpha^\top R \alpha^\top,$$
$$= (I - \alpha \alpha^\top) R \alpha^\top + \alpha R^\top (I - \alpha \alpha^\top).$$

Computing the evolution of $(I - \alpha \alpha^\top)^{-1}$,

$$\dot{\overline{(I - \alpha \alpha^\top)^{-1}}} = (I - \alpha \alpha^\top)^{-1} \left[ \dot{\overline{\alpha \alpha^\top}} \right] (I - \alpha \alpha^\top)^{-1}, \qquad (C.10)$$

$$= R \alpha^\top (I - \alpha \alpha^\top)^{-1} + (I - \alpha \alpha^\top)^{-1} \alpha R^\top. \qquad (C.11)$$

Let $C_\alpha := 2\gamma (I - \alpha \alpha^\top)^{-1}$ and $C_Z := Z_1 Z_1^\top$, so we have,

$$\dot{C_\alpha} = C_\alpha \alpha R^\top + R \alpha^\top C_\alpha,$$
$$\dot{C_Z} = C_Z \alpha R^\top + R \alpha^\top C_Z.$$

Since, at initialization, $C_\alpha(0) = C_Z(0)$, we have $C_\alpha = C_Z$, for any time $t$. Therefore, we have,

$$Z_1 Z_1^\top = 2\gamma (I - \alpha \alpha^\top)^{-1}. \qquad (C.12)$$

Using $\beta = Z_1 Z_2 = Z_1 Z_1^\top \alpha$ and the above invariant, we obtain $\beta = 2\gamma (I - \alpha \alpha^\top)^{-1} \alpha$. $\square$

**Lemma C.3.** *The following property holds for $\alpha$:*

$$\alpha_\infty = \lim_{t \to \infty} \alpha(t) \in span(X^T).$$

*Proof.* From the evolution of $\alpha$, we have

$$\dot{\alpha} = R - \alpha R^\top \alpha.$$

Let $U \in \mathbb{R}^{d \times d}$ be the matrix projection each column of $\alpha$ on the column span of $X^\top$, i.e., $span\{x_1, \ldots, x_n\}$, $U_\perp \in \mathbb{R}^{d \times d}$ be the matrix projection on the orthogonal space, i.e., $\text{Ker}(X^\top)$. So $\alpha = U\alpha + U_\perp \alpha$. Note that $R = X^\top (Y - X Z_1 Z_1^\top \alpha)$. So $U_\perp R = 0$, since $U_\perp X^\top = 0$. The evolution of $U_\perp \alpha$ is that

$$\dot{U_\perp \alpha} = -U_\perp \alpha R^\top (U\alpha + U_\perp \alpha).$$

Again, $U_\perp \alpha = 0$ is the equilibrium point and our initialization $\alpha = 0$ ensures that it stays at this equilibrium. This proves the lemma. $\square$

**Lemma C.4.** *Let $(Z_1^\infty, Z_2^\infty) \overset{\text{def}}{=} \lim_{t \to \infty} (Z_1(t), Z_2(t))$ the limit of the gradient flow dynamics. Then,*

$$(Z_1^\infty, Z_2^\infty) \in \underset{X Z_1 Z_2 = Y}{\operatorname{argmin}} \frac{1}{2} \|Z_2\|^2 + \frac{1}{2} \|Z_1\|_F^2 - \gamma \log (\det (Z_1 Z_1^\top)).$$

*Proof.* From Lemma C.3, we have that the $(\boldsymbol{Z}_1^\infty)^{-\top}\boldsymbol{Z}_2^\infty \in span(X)$, so the condition **(P2)** from Proposition C.5 holds. Note that from **(P2)** of Proposition C.5, we have,

$$\nabla\Psi_1\left(\boldsymbol{Z}_1^\infty\right) = (\boldsymbol{Z}_1^\infty)^{-\top}\boldsymbol{Z}_2^\infty\left(\boldsymbol{Z}_2^\infty\right)^\top,$$
$$= (\boldsymbol{Z}_1^\infty)^{-\top}\left((\boldsymbol{Z}_1^\infty)^\top\boldsymbol{Z}_1^\infty - 2\gamma\mathbf{I}\right),$$
$$= \boldsymbol{Z}_1^\infty - 2\gamma\left(\boldsymbol{Z}_1^\infty\right)^{-\top}.$$

which is satisfied by the potential

$$\Psi_1\left(\boldsymbol{Z}_1\right) = \frac{1}{2}\|\boldsymbol{Z}_1\|_F^2 - \gamma\log\left(\det\left(\boldsymbol{Z}_1\boldsymbol{Z}_1^\top\right)\right).$$

When the imbalance is not isotropic, i.e., $\boldsymbol{Z}_1^\top\boldsymbol{Z}_1 - \boldsymbol{Z}_2\boldsymbol{Z}_2^\top = D$, where $D$ is some diagonal matrix ($\neq c\mathbf{I}$, for any constant $c$). It this case,

$$\nabla\Psi_1\left(\boldsymbol{Z}_1^\infty\right) = \boldsymbol{Z}_1^\infty - D\left(\boldsymbol{Z}_1^\infty\right)^{-\top},$$

and there exists no such function $\Psi_1$ and the proof breaks. $\qquad\square$

**Proposition C.5.** *Let* $(\boldsymbol{Z}_1^*,\boldsymbol{Z}_2^*)$ *satisfy the following minimization problem*

$$(\boldsymbol{Z}_1^*,\boldsymbol{Z}_2^*) = \operatorname*{argmin}_{X\boldsymbol{Z}_1\boldsymbol{Z}_2=y}\ \Psi_1\left(\boldsymbol{Z}_1\right) + \Psi_2\left(\boldsymbol{Z}_2\right),$$

*for some non-negative potential functions* $\Psi_1\left(\boldsymbol{Z}_1\right),\Psi_2\left(\boldsymbol{Z}_2\right)$. *Then,* $(\boldsymbol{Z}_1^*,\boldsymbol{Z}_2^*)$ *satisfies*

**(P1)** $(\boldsymbol{Z}_1^*)^{-\top}\nabla\Psi_2(\boldsymbol{Z}_2^*) \in \operatorname{span}(X)$,

**(P2)** $\nabla\Psi_1\left(\boldsymbol{Z}_1^*\right) = (\boldsymbol{Z}_1^*)^{-\top}\nabla\Psi_2(\boldsymbol{Z}_2^*)(\boldsymbol{Z}_2^*)^\top$.

*Proof.* The Lagrangian for the minimization problem above is,

$$\mathcal{L}\left(\boldsymbol{Z}_1,\boldsymbol{Z}_2,\lambda\right) = \Psi_1\left(\boldsymbol{Z}_1\right) + \Psi_2\left(\boldsymbol{Z}_2\right) + \langle\lambda, X\boldsymbol{Z}_1\boldsymbol{Z}_2 - y\rangle.$$

Taking derivatives w.r.t. to $\boldsymbol{Z}_1,\boldsymbol{Z}_2$, we get,

$$\nabla_{\boldsymbol{Z}_1}\mathcal{L}\left(\boldsymbol{Z}_1,\boldsymbol{Z}_2,\lambda\right) = \nabla\Psi_1\left(\boldsymbol{Z}_1\right) + \left(X^\top\lambda\right)\boldsymbol{Z}_2^\top.$$
$$\nabla_{\boldsymbol{Z}_2}\mathcal{L}\left(\boldsymbol{Z}_1,\boldsymbol{Z}_2,\lambda\right) = \nabla\Psi_2\left(\boldsymbol{Z}_2\right) + \boldsymbol{Z}_1^\top(X^\top\lambda).$$

As $\boldsymbol{Z}_1^*,\boldsymbol{Z}_2^*$ should satisfy $\nabla\mathcal{L} = 0$.

$$(\boldsymbol{Z}_1^*)^{-\top}\nabla\Psi_2(\boldsymbol{Z}_2^*) = -X^\top\lambda,$$
$$\nabla\Psi_1\left(\boldsymbol{Z}_1^*\right) = (\boldsymbol{Z}_1^*)^{-\top}\nabla\Psi_2(\boldsymbol{Z}_2^*)(\boldsymbol{Z}_2^*)^\top$$

$\qquad\square$

**Lemma C.6.** *Let* $(\boldsymbol{Z}_1,\boldsymbol{Z}_2)$ *be the process that follows the GF equations* (3.1)*, initialized according to condition* $I_\gamma$*, for some* $\gamma > 0$. *Then*

*(i) The parameters converge to a global optimum of the loss*

$$\lim_{t\to\infty} Y - X\boldsymbol{Z}_1(t)\boldsymbol{Z}_2(t) = 0.$$

*(ii) The linear predictor* $\boldsymbol{\beta}$ *converges to the minimum* $\ell_2$*-norm interpolator*

$$\lim_{t\to\infty}\boldsymbol{\beta}(t) = \operatorname*{argmin}_{X\boldsymbol{\beta}=Y}\ \|\boldsymbol{\beta}\|_2 \overset{\text{def}}{=} \boldsymbol{\beta}_*.$$

*Proof.* The evolution of $\boldsymbol{R}$ writes,

$$\dot{\boldsymbol{R}} = -(X^\top X)\left[\boldsymbol{Z}_1\boldsymbol{Z}_1^\top\boldsymbol{R} + \boldsymbol{R}\boldsymbol{Z}_2^\top\boldsymbol{Z}_2\right],$$
$$\widehat{\operatorname{tr}\left(\boldsymbol{R}^\top\boldsymbol{R}\right)} = -2\operatorname{tr}\left(\boldsymbol{R}^\top(X^\top X)\boldsymbol{Z}_1\boldsymbol{Z}_1^\top\boldsymbol{R}\right) - 2\operatorname{tr}\left(\boldsymbol{R}^\top(X^\top X)\boldsymbol{R}\boldsymbol{Z}_2^\top\boldsymbol{Z}_2\right).$$

Note that for any three PSD matrices $ABC$. Using Lasserre [1995],( [see Lemma 7, Min et al., 2021])

$$\text{tr}\left(ABC\right) \geq \lambda_{\min}(A)\lambda_{\min}(B)\text{tr}\,C.$$

From the invariance, we have,

$$\boldsymbol{Z}_1^\top \boldsymbol{Z}_1 = 2\gamma \mathbf{I}_d + \boldsymbol{Z}_2^\top \boldsymbol{Z}_2,$$
$$\boldsymbol{Z}_1^\top \boldsymbol{Z}_1 \succcurlyeq 2\gamma \mathbf{I}_d.$$

Since $\boldsymbol{Z}_1^\top \boldsymbol{Z}_1$ and $\boldsymbol{Z}_1 \boldsymbol{Z}_1^\top$, share the same eigenvalues, $\boldsymbol{Z}_1 \boldsymbol{Z}_1^\top \succcurlyeq 2\gamma \mathbf{I}_d$. Therefore $\lambda_{min}(\boldsymbol{Z}_1 \boldsymbol{Z}_1^\top) \geq 2\gamma$. Let $\lambda_{\min}(X^\top X)$ be the smallest non-zero eigenvalue of $X^\top X$. Although $(X^\top X)$ can have zero eigenvalues, we can always restrict the evolution to the $span\left(X^\top\right)$, which $\boldsymbol{R}$ belongs to and without loss of generality assume that all the eigenvalues are non-zero. Using this for the first term, we have,

$$\text{tr}\left(\boldsymbol{R}^\top (X^\top X)\boldsymbol{Z}_1 \boldsymbol{Z}_1^\top \boldsymbol{R}\right) = \text{tr}\left((X^\top X)\boldsymbol{Z}_1 \boldsymbol{Z}_1^\top \boldsymbol{R}\boldsymbol{R}^\top\right),$$
$$\geq \lambda_{\min}(X^\top X)\lambda_{\min}(\boldsymbol{Z}_1 \boldsymbol{Z}_1^\top)\text{tr}\left(\boldsymbol{R}\boldsymbol{R}^\top\right) \geq 2\lambda_{\min}(X^\top X)\gamma\text{tr}\left(\boldsymbol{R}\boldsymbol{R}^\top\right).$$

For the second term, $\boldsymbol{Z}_2^\top \boldsymbol{Z}_2 \succcurlyeq 0$ we have,

$$\text{tr}\left(\boldsymbol{R}^\top (X^\top X)\boldsymbol{R}\boldsymbol{Z}_2^\top \boldsymbol{Z}_2\right) = \text{tr}\left((X^\top X)\boldsymbol{R}\boldsymbol{Z}_2^\top \boldsymbol{Z}_2 \boldsymbol{R}^\top\right),$$
$$\geq \lambda_{\min}(X^\top X)\text{tr}\left(\boldsymbol{R}\boldsymbol{Z}_2^\top \boldsymbol{Z}_2 \boldsymbol{R}^\top\right),$$
$$\geq 0.$$

Combining both,

$$\overset{\bullet}{\overbrace{\text{tr}\left(\boldsymbol{R}^\top \boldsymbol{R}\right)}} \leq -2\lambda_{\min}(X^\top X)\gamma\text{tr}\left(\boldsymbol{R}\boldsymbol{R}^\top\right). \tag{C.13}$$

Thus, using Gronwall we can show that $\left\|\boldsymbol{R}\right\|_F^2$ decays exponentially to zero, thus we have $X\boldsymbol{\beta} - Y \to 0$. The last step is due to overparameterization, i.e., $d > n$ and existence of a interpolating solution.

**Implicit bias of $\boldsymbol{\beta}$.** We know that $\boldsymbol{\alpha} \in \text{span}(X^\top)$ and $\boldsymbol{\beta} = \left(\mathbf{I} - \boldsymbol{\alpha}\boldsymbol{\alpha}^\top\right)^{-1}\boldsymbol{\alpha}$. Using, Woodbury matrix identity,

$$\left(\mathbf{I} - \boldsymbol{\alpha}\boldsymbol{\alpha}^\top\right)^{-1} = \mathbf{I} - \boldsymbol{\alpha}\left(\mathbf{I} - \boldsymbol{\alpha}^\top \boldsymbol{\alpha}\right)^{-1}\boldsymbol{\alpha}^\top,$$
$$\left(\mathbf{I} - \boldsymbol{\alpha}\boldsymbol{\alpha}^\top\right)^{-1}\boldsymbol{\alpha} = \boldsymbol{\alpha} - \boldsymbol{\alpha}\left(\mathbf{I} - \boldsymbol{\alpha}^\top \boldsymbol{\alpha}\right)^{-1}\boldsymbol{\alpha}^\top \boldsymbol{\alpha}.$$

Therefore $\boldsymbol{\beta} \in span(X^\top)$. Hence, this satisfies the KKT conditions for

$$\boldsymbol{\beta}_\infty \in \underset{X\boldsymbol{\beta}=Y}{\arg\min}\,\frac{1}{2}\left\|\boldsymbol{\beta}\right\|^2.$$

$\square$

**Lemma C.7** (**Mirror flow(k=1).**)**.** *For $k = 1$, the dynamics of $\boldsymbol{\beta}$ can be characterized as a mirror flow*

$$\mathrm{d}\nabla\psi_\gamma\left(\boldsymbol{\beta}\right) = -\left[\gamma + \sqrt{\left\|\boldsymbol{\beta}\right\|^2 + \gamma^2}\right]^{1/2}\,\nabla\mathcal{L}\left(\boldsymbol{\beta}\right)\,\mathrm{d}t, \tag{C.14}$$

*where the potential writes*

$$\psi_\gamma\left(\boldsymbol{\beta}\right) \coloneqq \frac{2}{3}\left[\sqrt{\left\|\boldsymbol{\beta}\right\|^2 + \gamma^2} + \gamma\right]^{3/2} - 2\gamma\left[\sqrt{\left\|\boldsymbol{\beta}\right\|^2 + \gamma^2} + \gamma\right]^{1/2}.$$

*Proof.* Here we consider the case $k = 1$, let $\boldsymbol{r} \coloneqq X^\top(Y - X\boldsymbol{\beta})$. It is denoted lowercase since $r$ here is a vector. The gradient flow C.1 now in $\boldsymbol{Z}_1, \boldsymbol{\alpha}$ can be now can be written as,

$$\dot{\boldsymbol{\alpha}} = \left(\mathbf{I} - \boldsymbol{\alpha}\boldsymbol{\alpha}^\top\right)\boldsymbol{r} \quad , \quad \dot{\boldsymbol{Z}}_1 = \boldsymbol{Z}_1\boldsymbol{\alpha}\boldsymbol{r}^\top,$$
$$\dot{\boldsymbol{\beta}} = \left(\boldsymbol{Z}_1^\top \dot{\boldsymbol{Z}_1}\boldsymbol{\alpha}\right) = (\boldsymbol{Z}_1^\top \boldsymbol{Z}_1 + \boldsymbol{\alpha}^\top \boldsymbol{Z}_1^\top \boldsymbol{Z}_1\boldsymbol{\alpha})\boldsymbol{r} \quad , \quad \dot{\boldsymbol{r}} = -\left(X^\top X\right)(\boldsymbol{Z}_1^\top \boldsymbol{Z}_1 + \boldsymbol{\alpha}^\top \boldsymbol{Z}_1^\top \boldsymbol{Z}_1\boldsymbol{\alpha})\boldsymbol{r}.$$

$\square$

Recall from Lemma 3.3 that
$$\boldsymbol{\beta} = 2\gamma(\mathbf{I} - \boldsymbol{\alpha}\boldsymbol{\alpha}^\top)^{-1}\boldsymbol{\alpha}.$$

Using Sherman-Morrison,
$$(\mathbf{I} - \boldsymbol{\alpha}\boldsymbol{\alpha}^\top)^{-1} = \mathbf{I} + \frac{\boldsymbol{\alpha}\boldsymbol{\alpha}^\top}{1 - \|\boldsymbol{\alpha}\|^2}.$$

$$\boldsymbol{\beta} = 2\gamma\left[(\mathbf{I} - \boldsymbol{\alpha}\boldsymbol{\alpha}^\top)^{-1}\right]\boldsymbol{\alpha} = 2\gamma\left[\mathbf{I} + \frac{\boldsymbol{\alpha}\boldsymbol{\alpha}^\top}{1 - \|\boldsymbol{\alpha}\|^2}\right]\boldsymbol{\alpha} = 2\gamma\frac{\boldsymbol{\alpha}}{1 - \|\boldsymbol{\alpha}\|^2}.$$

Taking the norms, we get,
$$\|\boldsymbol{\beta}\| = 2\gamma\frac{\|\boldsymbol{\alpha}\|}{1 - \|\boldsymbol{\alpha}\|^2}, \quad \|\boldsymbol{\alpha}\|^2 + \frac{2\gamma}{\|\boldsymbol{\beta}\|}\|\boldsymbol{\alpha}\| - 1 = 0.$$

$$\|\boldsymbol{\alpha}\| = -\frac{\gamma}{\|\boldsymbol{\beta}\|} + \sqrt{\left(\frac{\gamma}{\|\boldsymbol{\beta}\|}\right)^2 + 1}.$$

$$1 - \|\boldsymbol{\alpha}\|^2 = 1 - \left[-\frac{\gamma}{\|\boldsymbol{\beta}\|} + \sqrt{\left(\frac{\gamma}{\|\boldsymbol{\beta}\|}\right)^2 + 1}\right]^2,$$

$$= 2\frac{\gamma}{\|\boldsymbol{\beta}\|}\sqrt{\left(\frac{\gamma}{\|\boldsymbol{\beta}\|}\right)^2 + 1} - 2\left(\frac{\gamma}{\|\boldsymbol{\beta}\|}\right)^2,$$

$$= \frac{2\gamma}{\sqrt{\gamma^2 + \|\boldsymbol{\beta}\|^2} + \gamma}.$$

Now consider the evolution of $\boldsymbol{\alpha}$
$$\dot{\boldsymbol{\alpha}} = \left[\mathbf{I} - \boldsymbol{\alpha}\boldsymbol{\alpha}^\top\right]\boldsymbol{r},$$

$$\left[\mathbf{I} - \boldsymbol{\alpha}\boldsymbol{\alpha}^\top\right]^{-1}\dot{\boldsymbol{\alpha}} = \boldsymbol{r},$$

$$\left[\mathbf{I} + \frac{\boldsymbol{\alpha}\boldsymbol{\alpha}^\top}{1 - \|\boldsymbol{\alpha}\|^2}\right]\dot{\boldsymbol{\alpha}} = \boldsymbol{r}.$$

One cannot write $\left[\mathbf{I} + \frac{\boldsymbol{\alpha}\boldsymbol{\alpha}^\top}{1 - \|\boldsymbol{\alpha}\|^2}\right]$ as a Hessian of any function of $\boldsymbol{\alpha}$. However, multiplying on both sides with $(1 - \|\boldsymbol{\alpha}\|^2)^{-1/2}$, we obtain,

$$\left[\frac{\mathbf{I}}{\left(1 - \|\boldsymbol{\alpha}\|^2\right)^{1/2}} + \frac{\boldsymbol{\alpha}\boldsymbol{\alpha}^\top}{\left(1 - \|\boldsymbol{\alpha}\|^2\right)^{3/2}}\right]\dot{\boldsymbol{\alpha}} = \frac{\boldsymbol{r}}{\left(1 - \|\boldsymbol{\alpha}\|^2\right)^{1/2}},$$

$$\frac{\mathrm{d}}{\mathrm{d}t}\left[\frac{\boldsymbol{\alpha}}{\left(1 - \|\boldsymbol{\alpha}\|^2\right)^{1/2}}\right] = \frac{\boldsymbol{r}}{\left(1 - \|\boldsymbol{\alpha}\|^2\right)^{1/2}}.$$

We tackle the left hand side as follows,
$$\boldsymbol{\beta} = \frac{2\gamma\boldsymbol{\alpha}}{1 - \|\boldsymbol{\alpha}\|^2},$$

$$\frac{\boldsymbol{\alpha}}{\left(1 - \|\boldsymbol{\alpha}\|^2\right)^{1/2}} = \frac{\boldsymbol{\beta}}{2\gamma}\left(1 - \|\boldsymbol{\alpha}\|^2\right)^{1/2} = \frac{\boldsymbol{\beta}}{2\gamma}\left(\frac{2\gamma}{\sqrt{\gamma^2 + \|\boldsymbol{\beta}\|^2} + \gamma}\right)^{1/2},$$

$$= \frac{\boldsymbol{\beta}}{\sqrt{2\gamma}\left(\sqrt{\gamma^2 + \|\boldsymbol{\beta}\|^2} + \gamma\right)^{1/2}}.$$

Substituting the above expression and also $\left(1 - \left\|\boldsymbol{\alpha}\right\|^2\right)^{1/2}$ on the RHS gives us,

$$\frac{\mathrm{d}}{\mathrm{d}t}\left[\frac{\boldsymbol{\beta}}{\left(\sqrt{\gamma^2 + \left\|\boldsymbol{\beta}\right\|^2} + \gamma\right)^{1/2}}\right] = \left(\sqrt{\gamma^2 + \left\|\boldsymbol{\beta}\right\|^2} + \gamma\right)^{1/2}\boldsymbol{r}.$$

Define the mirror potential,

$$\psi_\gamma\left(\boldsymbol{\beta}\right) \stackrel{\text{def}}{=} \frac{2}{3}\left[\sqrt{\left\|\boldsymbol{\beta}\right\|^2 + \gamma^2} + \gamma\right]^{3/2} - 2\gamma\left[\sqrt{\left\|\boldsymbol{\beta}\right\|^2 + \gamma^2} + \gamma\right]^{1/2}. \tag{C.15}$$

Using the fact that,

$$\frac{\mathrm{d}}{\mathrm{d}x}\left[\frac{2}{3}\left[\sqrt{x^2 + p^2} + q\right]^{3/2} - 2q\left[\sqrt{x^2 + p^2} + q\right]^{1/2}\right] = \left[\sqrt{x^2 + p^2} + q\right]^{-1/2}x.$$

We have,

$$\nabla\psi_\gamma\left(\boldsymbol{\beta}\right) = \left(\gamma + \sqrt{\left\|\boldsymbol{\beta}\right\|^2 + \gamma^2}\right)^{-1/2}\boldsymbol{\beta}.$$

Thus, we can write it as a continuous-time mirror descent.

$$\overset{\bullet}{\widehat{\nabla\psi_\gamma\left(\boldsymbol{\beta}\right)}} = -\left(\sqrt{\gamma^2 + \left\|\boldsymbol{\beta}\right\|^2} + \gamma\right)^{1/2}\nabla\mathcal{L}\left(\boldsymbol{\beta}\right).$$

**Lemma C.8** (**Convergence rate of mirror flow**). *For the dynamics of the mirror flow given by Eq. (C.14), we have the following rate of convergence,*

$$\mathcal{L}\left(\boldsymbol{\beta}\left(\mathbf{t}\right)\right) \leq \frac{D_{\psi_\gamma}\left(\boldsymbol{\beta}_*, \boldsymbol{\beta}_0\right)}{\sqrt{2\gamma}\,\mathbf{t}}.$$

*Proof.* Note that $\psi_\gamma$ is convex. Let $D_{\psi_\gamma}\left(.,.\right)$ be the Bregman divergence defined with the potential $\psi_\gamma(.)$. Taking the time derivative of the Bregman divergence, we get,

$$\frac{\mathrm{d}}{\mathrm{d}t}D_{\psi_\gamma}\left(\boldsymbol{\beta}_*, \boldsymbol{\beta}\right) = \left\langle\overset{\bullet}{\widehat{\nabla\psi_\gamma(\boldsymbol{\beta})}}, \boldsymbol{\beta} - \boldsymbol{\beta}_*\right\rangle,$$

$$= -\left(\sqrt{\gamma^2 + \left\|\boldsymbol{\beta}\right\|^2} + \gamma\right)^{1/2}\left\langle\nabla\mathcal{L}\left(\boldsymbol{\beta}\right), \boldsymbol{\beta} - \boldsymbol{\beta}_*\right\rangle.$$

We have,

$$-\left(\sqrt{\gamma^2 + \left\|\boldsymbol{\beta}\right\|^2} + \gamma\right)^{1/2} \leq -\sqrt{2\gamma}$$

Using the convexity of $\mathcal{L}\left(.\right)$ in $\boldsymbol{\beta}$, we have

$$\left\langle\nabla\mathcal{L}\left(\boldsymbol{\beta}\right), \boldsymbol{\beta} - \boldsymbol{\beta}_*\right\rangle \leq \mathcal{L}\left(\boldsymbol{\beta}\right) - \mathcal{L}\left(\boldsymbol{\beta}_*\right) = \mathcal{L}\left(\boldsymbol{\beta}\right).$$

The last step is using the existence of an interpolating solution. Substituting the above two inequalities, we get,

$$\frac{\mathrm{d}}{\mathrm{d}t}D_{\psi_\gamma}\left(\boldsymbol{\beta}_*, \boldsymbol{\beta}\right) \leq -\sqrt{2\gamma}\mathcal{L}\left(\boldsymbol{\beta}\right).$$

Integrating, we get,

$$\int_0^t \mathcal{L}\left(\boldsymbol{\beta}(s)\right)ds \leq \frac{1}{\sqrt{2\gamma}}\left[D_{\psi_\gamma}\left(\boldsymbol{\beta}_*, \boldsymbol{\beta}_0\right) - D_{\psi_\gamma}\left(\boldsymbol{\beta}_*, \boldsymbol{\beta}_t\right)\right] \leq \frac{D_{\psi_\gamma}\left(\boldsymbol{\beta}_*, \boldsymbol{\beta}_0\right)}{\sqrt{2\gamma}}.$$

The last step is using the fact that Bregman divergence is positive for convex functions. We will now show that the loss is decreasing along the trajectory,

$$\dot{\mathcal{L}}(\boldsymbol{\beta}) = \left\langle \nabla \mathcal{L}(\boldsymbol{\beta}), \dot{\boldsymbol{\beta}} \right\rangle = -\frac{1}{\left(\sqrt{\gamma^2 + \|\boldsymbol{\beta}\|^2} + \gamma\right)^{1/2}} \left\langle \dot{\widehat{\nabla \psi_\gamma(\boldsymbol{\beta})}}, \dot{\boldsymbol{\beta}} \right\rangle.$$

$$\dot{\widehat{\nabla \psi_\gamma(\boldsymbol{\beta})}} = \nabla^2 \psi_\gamma(\boldsymbol{\beta}) \dot{\boldsymbol{\beta}}, \quad \left\langle \dot{\widehat{\nabla \psi_\gamma(\boldsymbol{\beta})}}, \dot{\boldsymbol{\beta}} \right\rangle = \left\langle \nabla^2 \psi_\gamma(\boldsymbol{\beta}) \dot{\boldsymbol{\beta}}, \dot{\boldsymbol{\beta}} \right\rangle \geq 0, \quad \text{(Using convexity)}.$$

Thus $\dot{\mathcal{L}}(\boldsymbol{\beta}) \leq 0$, using this,

$$t\mathcal{L}(\boldsymbol{\beta}(t)) \leq \int_0^t \mathcal{L}(\boldsymbol{\beta}(s))\, ds \leq \frac{D_{\psi_\gamma}(\boldsymbol{\beta}_*, \boldsymbol{\beta}_0)}{\sqrt{2\gamma}}.$$

This completes the proof. □

**Definition C.9.** *[Singular value decomposition and notation.] The singular value decomposition of a given matrix $A \in \mathbb{R}^{d \times k}$ be denoted as $U_A D_A V_A^\top$ where $U_A \in \mathbb{R}^{d \times d}$, $D_A \in \mathbb{R}^{d \times k}$, $V_A \in \mathbb{R}^{k \times k}$.*

*Let $k < d$ and consider the products $AA^\top$ and $A^\top A$. We will refer to the diagonal matrices $D_{A^\top A} \in \mathbb{R}^{k \times k}$ and $D_{AA^\top} \in \mathbb{R}^{d \times d}$ as $D_A^2$ and $\tilde{D}_A^2$, in order to ease notation. Note that $\tilde{D}_A^2$ is a block-diagonal matrix containing $D_A^2$ as the first diagonal block and $\mathbf{0}$ as the second diagonal block.*

*Finally, if $k = d$ then $D_A^2 = \tilde{D}_A^2$.*

**Lemma C.10** (**Singular values of $\mathbf{Z}_1$ and $\boldsymbol{\beta}$**)**.** *Using the notations in Definition C.9, the singular values of $\mathbf{Z}_1, \mathbf{Z}_2$ and $\boldsymbol{\beta}$ are related as the following,*

$$\mathbf{D}_{\mathbf{Z}_2}^2 = -\gamma I_k + \sqrt{\gamma^2 I_k + \mathbf{D}_{\boldsymbol{\beta}}^2} \quad \text{and} \quad \tilde{\mathbf{D}}_{\mathbf{Z}_1}^2 = \gamma I_d + \sqrt{\gamma^2 I_d + \tilde{\mathbf{D}}_{\boldsymbol{\beta}}^2},$$

*Proof.* From the invariance (C.8) we further deduce (by appropriately multiplying left and right with $\mathbf{Z}_1, \mathbf{Z}_2$ and their transposes):

$$\boldsymbol{\beta}^\top \boldsymbol{\beta} - (\mathbf{Z}_2^\top \mathbf{Z}_2)^2 = 2\gamma \mathbf{Z}_2^\top \mathbf{Z}_2 \quad \text{and} \quad (\mathbf{Z}_1 \mathbf{Z}_1^\top)^2 - \boldsymbol{\beta}\boldsymbol{\beta}^\top = 2\gamma \mathbf{Z}_1 \mathbf{Z}_1^\top,$$

which implies that $[\boldsymbol{\beta}^\top \boldsymbol{\beta}, \mathbf{Z}_2^\top \mathbf{Z}_2] = \mathbf{0}$ and $[\boldsymbol{\beta}\boldsymbol{\beta}^\top, \mathbf{Z}_1 \mathbf{Z}_1^\top] = \mathbf{0}$. Hence $\boldsymbol{\beta}^\top \boldsymbol{\beta}$ and $\mathbf{Z}_2^\top \mathbf{Z}_2$ commute and can be simultaneously diagonalizable, same is the case with $\boldsymbol{\beta}\boldsymbol{\beta}^\top, \mathbf{Z}_1 \mathbf{Z}_1^\top$. Therefore, the following relation holds (elementwise):

$$\mathbf{D}_{\mathbf{Z}_2}^2 = -\gamma I_k + \sqrt{\gamma^2 I_k + \mathbf{D}_{\boldsymbol{\beta}}^2} \quad \text{and} \quad \tilde{\mathbf{D}}_{\mathbf{Z}_1}^2 = \gamma I_d + \sqrt{\gamma^2 I_d + \tilde{\mathbf{D}}_{\boldsymbol{\beta}}^2}, \qquad \text{(C.16)}$$

where the appropriate dimensionality of the diagonal matrices is evident from the indexing of the identity matrices. □

**Theorem C.11** (**Singular vectors are static under orthogonal data**)**.** *Let $\boldsymbol{\alpha} := \mathbf{Z}_1^{-\top} \mathbf{Z}_2$. Then, it holds that*

$$\mathbf{U}_{\boldsymbol{\beta}} = \mathbf{U}_{\mathbf{Z}_1} = \mathbf{U}_{\boldsymbol{\alpha}},$$
$$\mathbf{V}_{\boldsymbol{\beta}} = \mathbf{V}_{\mathbf{Z}_2} = \mathbf{V}_{\boldsymbol{\alpha}}.$$

*Furthermore, if $X^\top X = I_d$, it additionally holds that*

$$\mathbf{U}_{\boldsymbol{\alpha}} = \mathbf{U}_{\mathbf{R}} = U_{\boldsymbol{\beta}^*},$$
$$\mathbf{V}_{\boldsymbol{\alpha}} = \mathbf{V}_{\mathbf{R}} = V_{\boldsymbol{\beta}^*}.$$

*Proof.* From invariance (C.8), by multiplying left and right with $\mathbf{V}_{\mathbf{Z}_1}^\top$ and $\mathbf{V}_{\mathbf{Z}_1}$, respectively, we get that:

$$\mathbf{D}_{\mathbf{Z}_1}^2 - \mathbf{V}_{\mathbf{Z}_1}^\top \mathbf{Z}_2 \mathbf{Z}_2^\top \mathbf{V}_{\mathbf{Z}_1} = 2\gamma I_d,$$

which implies that $\mathbf{V}_{\mathbf{Z}_1}$ also diagonalizes $\mathbf{Z}_2\mathbf{Z}_2^\top$ (i.e., the left singular vectors of $\mathbf{Z}_1$ are the same as the right singular vectors of $\mathbf{Z}_2$ for all $t > 0$). As a result, $\boldsymbol{\beta} = \mathbf{U}_{\mathbf{Z}_1}\mathbf{D}_{\boldsymbol{\beta}}\mathbf{V}_{\mathbf{Z}_2}^\top$ and thus we have shown that $\mathbf{U}_{\boldsymbol{\beta}} = \mathbf{U}_{\mathbf{Z}_1}$ and $\mathbf{V}_{\boldsymbol{\beta}} = \mathbf{V}_{\mathbf{Z}_2}$.

Next, by definition $\boldsymbol{\alpha} = \mathbf{Z}_1^{-\top}\mathbf{Z}_2$, and $\mathbf{Z}_1$ and $\mathbf{Z}_1^{-T}$ have the same left and right singular vectors (since $\mathbf{Z}_1^{-1} := \mathbf{V}_{\mathbf{Z}_1}\mathbf{D}_{\mathbf{Z}_1}^{-1}\mathbf{U}_{\mathbf{Z}_1}^\top$). This, in conjunction with the above proves $\mathbf{U}_{\mathbf{Z}_1} = \mathbf{U}_{\boldsymbol{\alpha}}$ and $\mathbf{V}_{\mathbf{Z}_2} = \mathbf{V}_{\boldsymbol{\alpha}}$.

To show that $\mathbf{U}_{\boldsymbol{\alpha}} = \mathbf{U}_{\mathbf{R}}$ and $\mathbf{V}_{\boldsymbol{\alpha}} = \mathbf{V}_{\mathbf{R}}$, we need to do a bit more work. We will show that:

$$\boldsymbol{\alpha}^\top\mathbf{R} = \mathbf{R}^\top\boldsymbol{\alpha}, \text{ and} \tag{C.17}$$

$$\boldsymbol{\alpha}\mathbf{R}^\top = \mathbf{R}\boldsymbol{\alpha}^\top. \tag{C.18}$$

Once we prove (C.17) and (C.18), it directly follows that

$$\left(\boldsymbol{\alpha}\boldsymbol{\alpha}^\top\right)\left(\mathbf{R}\mathbf{R}^\top\right) = \left(\mathbf{R}\mathbf{R}^\top\right)\left(\boldsymbol{\alpha}\boldsymbol{\alpha}^\top\right) \quad \text{and} \quad \left(\boldsymbol{\alpha}^\top\boldsymbol{\alpha}\right)\left(\mathbf{R}^\top\mathbf{R}\right) = \left(\mathbf{R}^\top\mathbf{R}\right)\left(\boldsymbol{\alpha}^\top\boldsymbol{\alpha}\right), \tag{C.19}$$

which gives the desired result.

To prove (C.17) we will show that $\mathbf{R}^\top\boldsymbol{\alpha} - \boldsymbol{\alpha}^\top\mathbf{R} = 0, \forall t > 0$. First, we recall some relations that will be needed. Under orthogonal data, it holds that

$$\mathbf{R} = (\boldsymbol{\beta}^* - \boldsymbol{\beta}) \quad \text{and} \quad \dot{\mathbf{R}} = -\dot{\boldsymbol{\beta}}. \tag{C.20}$$

Furthermore, using the reparametrization in terms of $\boldsymbol{\alpha}$ which induces identity (C.12) and the invariance (C.8), the original dynamics of $\boldsymbol{\beta}$ given by $\dot{\boldsymbol{\beta}} = R\mathbf{Z}_2^T\mathbf{Z}_2 + \mathbf{Z}_1\mathbf{Z}_1^\top R$ rewrites as

$$\dot{\boldsymbol{\beta}} = 2\gamma\left[\mathbf{R}\boldsymbol{\alpha}^\top(I_d - \boldsymbol{\alpha}\boldsymbol{\alpha}^\top)^{-1}\boldsymbol{\alpha} + (I_d - \boldsymbol{\alpha}\boldsymbol{\alpha}^\top)^{-1}\mathbf{R}\right]. \tag{C.21}$$

Now, using (C.21), it holds that

$$
\begin{aligned}
\widehat{\dot{\mathbf{R}^\top\boldsymbol{\alpha}}} &= \mathbf{R}^\top\mathbf{R} - (\mathbf{R}^\top\boldsymbol{\alpha})^2 - 2\gamma\mathbf{R}^\top(I_d - \boldsymbol{\alpha}\boldsymbol{\alpha}^\top)^{-1}\boldsymbol{\alpha} - 2\gamma\boldsymbol{\alpha}^\top(I_d - \boldsymbol{\alpha}\boldsymbol{\alpha}^\top)^{-1}\boldsymbol{\alpha}\mathbf{R}^\top\boldsymbol{\alpha} \\
&= \mathbf{R}^\top\mathbf{R} - (\mathbf{R}^\top\boldsymbol{\alpha})^2 - 2\gamma\mathbf{R}^\top\boldsymbol{\alpha}(I_k - \boldsymbol{\alpha}^\top\boldsymbol{\alpha})^{-1} - 2\gamma\boldsymbol{\alpha}^\top\boldsymbol{\alpha}(I_k - \boldsymbol{\alpha}^\top\boldsymbol{\alpha})^{-1}\mathbf{R}^\top\boldsymbol{\alpha} \\
&= \mathbf{R}^\top\mathbf{R} - (\mathbf{R}^\top\boldsymbol{\alpha})^2 - 2\gamma\mathbf{R}^\top\boldsymbol{\alpha}(I_k - \boldsymbol{\alpha}^\top\boldsymbol{\alpha})^{-1} - 2\gamma(I_k - \boldsymbol{\alpha}^\top\boldsymbol{\alpha})^{-1}\mathbf{R}^\top\boldsymbol{\alpha} + \mathbf{R}^\top\boldsymbol{\alpha},
\end{aligned}
$$

were we used the push-through identity $(I + \mathbf{U}\mathbf{V})^{-1}\mathbf{U} = \mathbf{U}(I + \mathbf{V}\mathbf{U})^{-1}$.

Denoting $\mathbf{C} := \mathbf{R}^\top\boldsymbol{\alpha}$, computing the transpose version of the above ODE and using the fact that $\mathbf{C}^2 - (\mathbf{C}^\top)^2 = \frac{1}{2}\left(\mathbf{C} + \mathbf{C}^\top\right)\left(\mathbf{C} - \mathbf{C}^\top\right) + \frac{1}{2}\left(\mathbf{C} - \mathbf{C}^\top\right)\left(\mathbf{C} + \mathbf{C}^\top\right)$, the following holds:

$$
\begin{aligned}
\widehat{\dot{\mathbf{C} - \mathbf{C}^\top}} &= -\mathbf{C}^2 - 2\gamma\mathbf{C}(I_k - \boldsymbol{\alpha}^\top\boldsymbol{\alpha})^{-1} - 2\gamma(I_k - \boldsymbol{\alpha}^\top\boldsymbol{\alpha})^{-1}\mathbf{C} + \mathbf{C} + (\mathbf{C}^\top)^2 \\
&\qquad + 2\gamma(I_k - \boldsymbol{\alpha}^\top\boldsymbol{\alpha})^{-1}\mathbf{C}^\top + 2\gamma\mathbf{C}^\top(I_k - \boldsymbol{\alpha}^\top\boldsymbol{\alpha})^{-1} - \mathbf{C}^\top \\
&= -\left[\mathbf{C}^2 - (\mathbf{C}^\top)^2\right] + (\mathbf{C} - \mathbf{C}^\top) - 2\gamma(I_k - \boldsymbol{\alpha}^\top\boldsymbol{\alpha})^{-1}(\mathbf{C} - \mathbf{C}^\top) - 2\gamma(\mathbf{C} - \mathbf{C}^\top)(I_k - \boldsymbol{\alpha}^\top\boldsymbol{\alpha})^{-1} \\
&= \left[\frac{1}{2}\left(\mathbf{I} - \mathbf{C} - \mathbf{C}^\top\right) - 2\gamma(I_k - \boldsymbol{\alpha}^\top\boldsymbol{\alpha})^{-1}\right](\mathbf{C} - \mathbf{C}^\top) \\
&\qquad\qquad\qquad - (\mathbf{C} - \mathbf{C}^\top)\left[\frac{1}{2}\left(\mathbf{I} - \mathbf{C} - \mathbf{C}^\top\right) - 2\gamma(I_k - \boldsymbol{\alpha}^\top\boldsymbol{\alpha})^{-1}\right].
\end{aligned}
$$

Finally, since $\mathbf{C}(0) - \mathbf{C}^\top(0) = 0$, it is a fixed point of the above equation and it implies that $\mathbf{C} = \mathbf{C}^\top, \forall t$. Thus, identity (C.17) is proven.

We proceed similarly for identity (C.18).

$$\widehat{\dot{\mathbf{R}\boldsymbol{\alpha}^\top}} = -2\gamma\mathbf{R}\boldsymbol{\alpha}^\top(I_d - \boldsymbol{\alpha}\boldsymbol{\alpha}^\top)^{-1} - 2\gamma(I_d - \boldsymbol{\alpha}\boldsymbol{\alpha}^\top)^{-1}\mathbf{R}\boldsymbol{\alpha}^\top + \mathbf{R}\mathbf{R}^\top + \mathbf{R}\boldsymbol{\alpha}^\top - (\mathbf{R}\boldsymbol{\alpha}^\top)^2.$$

Denoting $\mathbf{B} := \mathbf{R}\boldsymbol{\alpha}^\top$ and computing $\dot{\mathbf{B}} - \dot{\mathbf{B}}^\top$ we get:

$$
\begin{aligned}
\widehat{\dot{\mathbf{B} - \mathbf{B}^\top}} &= \left[\frac{1}{2}\left(I_d - \mathbf{B} - \mathbf{B}^\top\right) - 2\gamma(I_d - \boldsymbol{\alpha}\boldsymbol{\alpha}^\top)^{-1}\right](\mathbf{B} - \mathbf{B}^\top) \\
&\qquad\qquad + (\mathbf{B} - \mathbf{B}^\top)\left[\frac{1}{2}\left(I_d - \mathbf{B} - \mathbf{B}^\top\right) - 2\gamma(I_d - \boldsymbol{\alpha}\boldsymbol{\alpha}^\top)^{-1}\right].
\end{aligned}
$$

Since $(\mathbf{B}(0) - \mathbf{B}^\top(0)) = 0$ is a fixed point of the above equation, identity (C.18) is proven.

Finally, we show $\mathbf{U_R} = U_{\boldsymbol{\beta}^*}$ and $\mathbf{V_R} = V_{\boldsymbol{\beta}^*}$ using yet another invariance, namely

$$\mathbf{Z}_1^\top \mathbf{R} \mathbf{Z}_2^\top = \mathbf{Z}_2 \mathbf{R}^\top \mathbf{Z}_1, \forall t > 0. \tag{C.22}$$

We proceed by computing the associated time derivatives and showing that their difference is null:

$$\overset{\bullet}{\overline{\mathbf{Z}_1^\top \mathbf{R} \mathbf{Z}_2^\top}} - \overset{\bullet}{\overline{\mathbf{Z}_2 \mathbf{R}^\top \mathbf{Z}_1}} = \mathbf{Z}_1^\top \overset{\bullet}{\mathbf{R}} \mathbf{Z}_2^\top - \mathbf{Z}_2 \overset{\bullet}{\mathbf{R}} \mathbf{Z}_1 \tag{C.23}$$

$$= -\mathbf{Z}_1^\top \mathbf{R} \mathbf{Z}_2^\top \mathbf{Z}_2 \mathbf{Z}_2^\top - 2\gamma \mathbf{Z}_1^\top \mathbf{R} \mathbf{Z}_2^\top - \mathbf{Z}_2 \mathbf{Z}_2^\top \mathbf{Z}_1^\top \mathbf{R} \mathbf{Z}_2^\top \tag{C.24}$$

$$+ \mathbf{Z}_2 \mathbf{Z}_2^\top \mathbf{Z}_2 \mathbf{R}^\top \mathbf{Z}_1 + 2\gamma \mathbf{Z}_2 \mathbf{R}^\top \mathbf{Z}_1 + \mathbf{Z}_2 \mathbf{R}^\top \mathbf{Z}_1 \mathbf{Z}_2 \mathbf{Z}_2^\top \tag{C.25}$$

$$= \left( \mathbf{Z}_2 \mathbf{R}^\top \mathbf{Z}_1 - \mathbf{Z}_1^\top \mathbf{R} \mathbf{Z}_2^\top \right) \mathbf{Z}_2 \mathbf{Z}_2^\top + \mathbf{Z}_2 \mathbf{Z}_2^\top \left( \mathbf{Z}_2 \mathbf{R}^\top \mathbf{Z}_1 - \mathbf{Z}_1^\top \mathbf{R} \mathbf{Z}_2^\top \right) \tag{C.26}$$

$$- 2\gamma \left( \mathbf{Z}_1^\top \mathbf{R} \mathbf{Z}_2^\top - \mathbf{Z}_2 \mathbf{R}^\top \mathbf{Z}_1 \right), \tag{C.27}$$

where we used (C.20) and the invariance (C.8). Since $\mathbf{Z}_2(0)\mathbf{R}^\top(0)\mathbf{Z}_1(0) - \mathbf{Z}_1(0)^\top\mathbf{R}(0)\mathbf{Z}_2(0)^\top = 0$, it is a fixed point of the above equation and we have shown (C.22).

Finally, it holds that

$$\mathbf{Z}_1^\top R \mathbf{Z}_2^\top = \mathbf{Z}_2 \mathbf{R}^\top \mathbf{Z}_1 \iff \mathbf{Z}_1^\top \boldsymbol{\beta}^* \mathbf{Z}_2^\top - \mathbf{Z}_1^\top \boldsymbol{\beta} \mathbf{Z}_2^\top = \mathbf{Z}_2 \boldsymbol{\beta}^{*\top} \mathbf{Z}_1 - \mathbf{Z}_2 \boldsymbol{\beta}^\top \mathbf{Z}_1$$

$$\iff \mathbf{Z}_1^\top \boldsymbol{\beta}^* \mathbf{Z}_2^\top = \mathbf{Z}_2 \boldsymbol{\beta}^{*\top} \mathbf{Z}_1$$

$$\iff \boldsymbol{\beta}^* \boldsymbol{\alpha}^\top = \boldsymbol{\alpha} \boldsymbol{\beta}^{*\top}, \tag{C.28}$$

where the second equivalence comes from the fact that $\mathbf{Z}_1^\top \mathbf{Z}_1$ and $\mathbf{Z}_2 \mathbf{Z}_2^\top$ are simultaneously diagonalizable and thus commute (from invariance (C.8)), and the second equivalence comes from multiplying left and right with $\mathbf{Z}_1^{-\top}$ and $\mathbf{Z}_1^{-1}$, respectively and the definition of $\boldsymbol{\alpha}$. Furthermore,

$$R^\top \boldsymbol{\alpha} = \boldsymbol{\alpha}^\top R \iff \boldsymbol{\beta}^{*\top} \boldsymbol{\alpha} - \boldsymbol{\beta}^\top \boldsymbol{\alpha} = \boldsymbol{\alpha}^\top \boldsymbol{\beta}^* - \boldsymbol{\alpha}^\top \boldsymbol{\beta}$$

$$\iff \boldsymbol{\beta}^{*\top} \boldsymbol{\alpha} = \boldsymbol{\alpha}^\top \boldsymbol{\beta}^*, \tag{C.29}$$

where the second line comes from the fact that $\boldsymbol{\beta}^\top \boldsymbol{\alpha} = \boldsymbol{\alpha}^\top \mathbf{Z}_1 \mathbf{Z}_1^\top \boldsymbol{\alpha} = \boldsymbol{\alpha}^\top \boldsymbol{\beta}$.

From repeated applications of (C.28) and (C.29) we obtain that

$$\left( \boldsymbol{\alpha}^\top \boldsymbol{\alpha} \right) \left( \boldsymbol{\beta}^{*\top} \boldsymbol{\beta}^* \right) = \left( \boldsymbol{\beta}^{*\top} \boldsymbol{\beta}^* \right) \left( \boldsymbol{\alpha}^\top \boldsymbol{\alpha} \right) \quad \text{and} \quad \left( \boldsymbol{\alpha} \boldsymbol{\alpha}^\top \right) \left( \boldsymbol{\beta}^* \boldsymbol{\beta}^{*\top} \right) = \left( \boldsymbol{\beta}^* \boldsymbol{\beta}^{*\top} \right) \left( \boldsymbol{\alpha} \boldsymbol{\alpha}^\top \right), \tag{C.30}$$

and the final identity follows. $\qquad \square$

**Theorem C.12** (**Time evolution of singular values - orthogonal data**). *Assume that $X^\top X = I_d$. Then, the $i^{th}$ singular value of $\boldsymbol{\beta}$ evolves over time as*

$$\sigma_{\boldsymbol{\beta},i}(t) = \frac{2\gamma \left( \sqrt{1 + \frac{\gamma^2}{\sigma_{\star,i}^2}} \left[ \frac{\gamma + \sqrt{\sigma_{\star,i}^2 + \gamma^2} - \exp\left(-2t\sqrt{\sigma_{\star,i}^2 + \gamma^2}\right)\left(\sqrt{\sigma_{\star,i}^2 + \gamma^2} - \gamma\right)}{\gamma + \sqrt{\sigma_{\star,i}^2 + \gamma^2} + \exp\left(-2t\sqrt{\sigma_{\star,i}^2 + \gamma^2}\right)\left(\sqrt{\sigma_{\star,i}^2 + \gamma^2} - \gamma\right)} \right] - \frac{\gamma}{\sigma_{\star,i}} \right)}{1 - \left( \sqrt{1 + \frac{\gamma^2}{\sigma_{\star,i}^2}} \left[ \frac{\gamma + \sqrt{\sigma_{\star,i}^2 + \gamma^2} - \exp\left(-2t\sqrt{\sigma_{\star,i}^2 + \gamma^2}\right)\left(\sqrt{\sigma_{\star,i}^2 + \gamma^2} - \gamma\right)}{\gamma + \sqrt{\sigma_{\star,i}^2 + \gamma^2} + \exp\left(-2t\sqrt{\sigma_{\star,i}^2 + \gamma^2}\right)\left(\sqrt{\sigma_{\star,i}^2 + \gamma^2} - \gamma\right)} \right] - \frac{\gamma}{\sigma_{\star,i}} \right)^2}.$$

*As a consequence, under vanishing initialization $\gamma \to 0$ and with a rescaling of time as $t \to \ln(1/\gamma)t$, the $i^{th}$ singular value is learned at time $T_i = 1/2\sigma_{\star,i}$:*

$$\lim_{\gamma \to 0} \sigma_{\boldsymbol{\beta},i}(\ln(1/\gamma)t) = \begin{cases} 0, & \text{if } t < 1/2\sigma_{\star,i} \\ \sigma_{\star,i}, & \text{otherwise.} \end{cases}$$

*Proof.* From Lemma 3.3 we have that $\overset{\bullet}{\boldsymbol{\alpha}} = \mathbf{R} - \boldsymbol{\alpha} \mathbf{R} \boldsymbol{\alpha}^\top$. In light of Theorem C.11, identity (C.18) and it holds that:

$$\overset{\bullet}{\mathbf{D}_{\boldsymbol{\alpha}}} = \left( I_d - \tilde{\mathbf{D}}_{\boldsymbol{\alpha}}^2 \right) \mathbf{D}_{\boldsymbol{\beta}^*} - 2\gamma \mathbf{D}_{\boldsymbol{\alpha}}.$$

For the $i^{th}$ singular value of $\boldsymbol{\alpha}$ it holds that:

$$\widehat{\dot{\sigma}_{\boldsymbol{\alpha},i}(t)} = -\sigma_{\star,i}\sigma_{\boldsymbol{\alpha},i}^2(t) - 2\gamma\sigma_{\boldsymbol{\alpha},i}(t) + \sigma_{\star,i}$$

$$= -\sigma_{\star,i}\left[\left(\sigma_{\boldsymbol{\alpha},i}(t) + \frac{\gamma}{\sigma_{\star,i}}\right)^2 - \left(1 + \frac{\gamma^2}{\sigma_{\star,i}^2}\right)\right], \tag{C.31}$$

where $\sigma_{\star,i}$ is the $i^{th}$ singular value of $\boldsymbol{\beta}^*$. Equation (C.31) is a Ricatti ODE and is separable. For ease of notation, let $p := \frac{\gamma}{\sigma_{\star,i}}$, $q := 1 + \frac{\gamma^2}{\sigma_{\star,i}^2}$ and $r := -\sigma_{\star,i}$. Then, we need to solve the IVP:

$$\begin{cases} \widehat{\dot{\sigma}_{\boldsymbol{\alpha},i}(t)} &= r\left[(\sigma_{\boldsymbol{\alpha},i}(t) + p)^2 - q\right] \\ \sigma_{\boldsymbol{\alpha},i}(0) &= 0, \end{cases} \tag{C.32}$$

for which we get $\sigma_{\boldsymbol{\alpha},i}(t) = \sqrt{q}\left[\frac{1-\exp(2r\sqrt{q}(t+c_1))}{1+\exp(2r\sqrt{q}(t+c_1))}\right] - p$ and solving for the initial value gives us $c_1 = \frac{1}{2r\sqrt{q}}\log\left(\frac{\sqrt{q}-p}{\sqrt{q}+p}\right)$. Replacing $p, q, r$ and rearranging we finally obtain:

$$\sigma_{\boldsymbol{\alpha},i}(t) = \sqrt{1 + \frac{\gamma^2}{\sigma_{\star,i}^2}}\left[\frac{\gamma + \sqrt{\sigma_{\star,i}^2 + \gamma^2} - \exp\left(-2t\sqrt{\sigma_{\star,i}^2 + \gamma^2}\right)\left(\sqrt{\sigma_{\star,i}^2 + \gamma^2} - \gamma\right)}{\gamma + \sqrt{\sigma_{\star,i}^2 + \gamma^2} + \exp\left(-2t\sqrt{\sigma_{\star,i}^2 + \gamma^2}\right)\left(\sqrt{\sigma_{\star,i}^2 + \gamma^2} - \gamma\right)}\right] - \frac{\gamma}{\sigma_{\star,i}}. \tag{C.33}$$

In order to obtain the dynamics for the singular values of $\boldsymbol{\beta} = \mathbf{Z}_1\mathbf{Z}_1^\top\boldsymbol{\alpha}$ we recall the relation given in Lemma C.10:

$$\tilde{\mathbf{D}}_{\mathbf{Z}_1}^2 = \gamma I_d + \sqrt{\gamma^2 I_d + \tilde{\mathbf{D}}_{\boldsymbol{\beta}}^2},$$

where the appropriate dimensionality of the diagonal matrices is evident from the indexing of the identity matrices. Therefore, the singular values $\sigma_{\boldsymbol{\beta},i}(t)$ are the solutions to the equation

$$\sigma_{\boldsymbol{\beta},i}(t) = \sigma_{\mathbf{Z}_1,i}^2(t)\sigma_{\boldsymbol{\alpha},i}(t) = \sigma_{\boldsymbol{\alpha},i}(t)\left[\gamma + \sqrt{\gamma^2 + \sigma_{\boldsymbol{\beta},i}^2(t)}\right],$$

and have the following expression:

$$\sigma_{\boldsymbol{\beta},i}(t) = \frac{2\gamma\sigma_{\boldsymbol{\alpha},i}(t)}{1 - \sigma_{\boldsymbol{\alpha},i}^2(t)}$$

$$= \frac{2\gamma\left(\sqrt{1 + \frac{\gamma^2}{\sigma_{\star,i}^2}}\left[\frac{\gamma+\sqrt{\sigma_{\star,i}^2+\gamma^2}-\exp\left(-2t\sqrt{\sigma_{\star,i}^2+\gamma^2}\right)\left(\sqrt{\sigma_{\star,i}^2+\gamma^2}-\gamma\right)}{\gamma+\sqrt{\sigma_{\star,i}^2+\gamma^2}+\exp\left(-2t\sqrt{\sigma_{\star,i}^2+\gamma^2}\right)\left(\sqrt{\sigma_{\star,i}^2+\gamma^2}-\gamma\right)}\right] - \frac{\gamma}{\sigma_{\star,i}}\right)}{1 - \left(\sqrt{1 + \frac{\gamma^2}{\sigma_{\star,i}^2}}\left[\frac{\gamma+\sqrt{\sigma_{\star,i}^2+\gamma^2}-\exp\left(-2t\sqrt{\sigma_{\star,i}^2+\gamma^2}\right)\left(\sqrt{\sigma_{\star,i}^2+\gamma^2}-\gamma\right)}{\gamma+\sqrt{\sigma_{\star,i}^2+\gamma^2}+\exp\left(-2t\sqrt{\sigma_{\star,i}^2+\gamma^2}\right)\left(\sqrt{\sigma_{\star,i}^2+\gamma^2}-\gamma\right)}\right] - \frac{\gamma}{\sigma_{\star,i}}\right)^2}.$$

Note that $\lim_{t\to\infty}\sigma_{\boldsymbol{\alpha},i}(t) = \frac{1}{\sigma_{\star,i}}\left(\sqrt{\sigma_{\star,i}^2 + \gamma^2} - \gamma\right)$ and therefore we can verify that $\boldsymbol{\beta} \xrightarrow{t\to\infty} \boldsymbol{\beta}^*$ by looking at the singular values, since the singular vectors are static (Lemma C.11).

$$\lim_{t\to\infty}\sigma_{\boldsymbol{\beta},i}(t) = \frac{2\gamma\left(\sqrt{\sigma_{\star,i}^2 + \gamma^2} - \gamma\right)}{\frac{2\gamma}{\sigma_{\star,i}}\left(\sqrt{\sigma_{\star,i}^2 + \gamma^2} - \gamma\right)}$$

$$= \sigma_{\star,i}.$$

We can now derive asymptotic transition times at which the singular values are learned. Now, we further process expression (C.33). Let $v := \sqrt{\sigma_{\star,i}^2 + \gamma^2} - \gamma$, $w := \sqrt{\sigma_{\star,i}^2 + \gamma^2} + \gamma$, then

$v + w = 2\sqrt{\sigma_{\star,i}^2 + \gamma^2}$ and $w - v = 2\gamma$. We re-write:

$$\sigma_{\boldsymbol{\alpha},i}(t) = \frac{1}{2\sigma_{\star,i}} \left[ (w+v) \left[ \frac{w - \exp\left(-t(v+w)\right) v}{w + \exp(-t(v+w))v} \right] - (w-v) \right]$$

$$= \frac{1}{2\sigma_{\star,i}} \left[ w \left( \frac{w - \exp\left(-t(v+w)\right) v}{w + \exp(-t(v+w))v} - 1 \right) + v \left( \frac{w - \exp\left(-t(v+w)\right) v}{w + \exp(-t(v+w))v} + 1 \right) \right]$$

$$= \frac{1}{\sigma_{\star,i}} \left[ \frac{-vw \exp\left(-t(v+w)\right)}{w + \exp(-t(v+w))v} + \frac{vw}{w + \exp(-t(v+w))v} \right]$$

$$= \frac{v}{\sigma_{\star,i}} \left[ 1 - \underbrace{\frac{(v+w) \exp\left(-t(v+w)\right)}{w + v\exp(-t(v+w))}}_{=:\, h(t)} \right]. \tag{C.34}$$

Since from before we have that $\sigma_{\boldsymbol{\beta},i}(t) = \dfrac{2\gamma\sigma_{\boldsymbol{\alpha},i}(t)}{1 - \sigma_{\boldsymbol{\alpha},i}^2(t)}$ and we can write $1 - \sigma_{\boldsymbol{\alpha},i}^2(t) = \dfrac{2\gamma v}{\sigma_{\star,i}^2} + \dfrac{v^2 h(t)\left(2 - h(t)\right)}{\sigma_{\star,i}^2}$, the singular values of $\boldsymbol{\beta}$ become:

$$\sigma_{\boldsymbol{\beta},i}(t) = \frac{2\gamma\sigma_{\boldsymbol{\alpha},i}(t)}{1 - \sigma_{\boldsymbol{\alpha},i}^2(t)} = \frac{\frac{2v\gamma}{\sigma_{\star,i}}\left[1 - h(t)\right]}{\frac{2\gamma v}{\sigma_{\star,i}^2} + \frac{v^2 h(t)(2-h(t))}{\sigma_{\star,i}^2}} = \frac{\sigma_{\star,i}\left[1 - h(t)\right]}{1 + \frac{vh(t)(2-h(t))}{2\gamma}}.$$

We wish to study the limit of infinitesimal initialization $\gamma \to 0$. We introduce a constant $c > 0$ and rewrite $\gamma = \exp(-c)$ and have $c = \ln(1/\gamma)$. Rescaling time $t \to ct$ and taking the limit $c \to \infty$ we have

$$\lim_{c\to\infty} \sigma_{\boldsymbol{\beta},i}(ct) = \lim_{c\to\infty} \frac{2\sigma_{\star,i}\left[1 - h(ct)\right]}{2 + vh(ct)\exp(c)\left(2 - h(ct)\right)}$$

$$= \frac{\sigma_{\star,i}}{1 + \lim_{c\to\infty} vh(ct)\exp(c)}$$

$$= \begin{cases} 0, & \text{if } t < 1/2\sigma_{\star,i}, \\ \frac{\sigma_{\star,i}}{1+2\sigma_{\star,i}}, & \text{if } t = 1/2\sigma_{\star,i} \\ \sigma_{\star,i}, & \text{if } t > 1/2\sigma_{\star,i}, \end{cases}$$

since $\displaystyle\lim_{c\to\infty} \exp(c)h(ct) = \lim_{c\to\infty} \frac{(v+w) \exp\left(c[1 - t(v+w)]\right)}{w + v\exp(-ct(v+w))}$

$$= \lim_{c\to\infty} \frac{2\sqrt{\sigma_{\star,i}^2 + \exp(-2c)} \exp\left(c[1 - 2t\sigma_{\star,i}]\right) \exp\left( \frac{-2ct\exp(-2c)}{\sigma_{\star,i} + \sqrt{\sigma_{\star,i}^2 + \exp(-2c)}} \right)}{\sqrt{\sigma_{\star,i}^2 + \exp(-2c)} + \exp(-c) + v\exp(-ct(v+w))}$$

$$= \begin{cases} \infty, & \text{if } t < 1/2\sigma_{\star,i} \\ 2, & \text{if } t = 1/2\sigma_{\star,i} \\ 0, & \text{if } t > 1/2\sigma_{\star,i}. \end{cases} \qquad\qquad \square$$

**Lemma C.13** (**Mirror on singular values**). *With the same notations as in Theorem 3.1, The singular values of $\boldsymbol{\beta}$, denoted by $\mathbf{D}_{\boldsymbol{\beta}}$, follow the mirror flow*

$$\mathrm{d}\nabla\Psi_\gamma\left(\mathbf{D}_{\boldsymbol{\beta}}\right) = -\nabla_{\mathbf{D}_{\boldsymbol{\beta}}}\mathcal{L}(\boldsymbol{\beta})\, \mathrm{d}t,$$

*where the potential is $\Psi_\gamma\left(\mathbf{D}_{\boldsymbol{\beta}}\right) := \mathrm{tr}\left( \frac{1}{2}\mathbf{D}_{\boldsymbol{\beta}} \sinh^{-1}\left(\mathbf{D}_{\boldsymbol{\beta}}/\gamma\right) - \sqrt{\mathbf{D}_{\boldsymbol{\beta}}^2 + \gamma^2} \right).$*

*Proof.* We recall the dynamics induces on $\boldsymbol{\alpha} = \mathbf{Z}_1^{-\top}\mathbf{Z}_2$ given in Lemma 3.3:

$$\dot{\boldsymbol{\alpha}} = \mathbf{R} - \boldsymbol{\alpha}\mathbf{R}^\top\boldsymbol{\alpha}. \tag{C.35}$$

Writing the SVD decomposition of $\boldsymbol{\alpha}$ as $\mathbf{U}_{\boldsymbol{\alpha}}\mathbf{D}_{\boldsymbol{\alpha}}\mathbf{V}_{\boldsymbol{\alpha}}^\top = \mathbf{U}_{\mathbf{Z}_1}\mathbf{D}_{\boldsymbol{\alpha}}\mathbf{V}_{\mathbf{Z}_2}^\top$ we have that:

$$
\begin{aligned}
\mathbf{U}_{\mathbf{Z}_1}^\top \dot{\boldsymbol{\alpha}} \mathbf{V}_{\mathbf{Z}_2} &= \mathbf{U}_{\mathbf{Z}_1}^\top \overbrace{\mathbf{U}_{\mathbf{Z}_1}\mathbf{D}_{\boldsymbol{\alpha}}\mathbf{V}_{\mathbf{Z}_2}^\top}^{\bullet} \mathbf{V}_{\mathbf{Z}_2} \\
&= \mathbf{U}_{\mathbf{Z}_1}^\top \dot{\mathbf{U}}_{\mathbf{Z}_1}\mathbf{D}_{\boldsymbol{\alpha}}\mathbf{V}_{\mathbf{Z}_2}^\top\mathbf{V}_{\mathbf{Z}_2} + \mathbf{U}_{\mathbf{Z}_1}^\top\mathbf{U}_{\mathbf{Z}_1}\dot{\mathbf{D}}_{\boldsymbol{\alpha}}\mathbf{V}_{\mathbf{Z}_2}^\top\mathbf{V}_{\mathbf{Z}_2} + \mathbf{U}_{\mathbf{Z}_1}^\top\mathbf{U}_{\mathbf{Z}_1}\mathbf{D}_{\boldsymbol{\alpha}}\dot{\mathbf{V}}_{\mathbf{Z}_2}^\top\mathbf{V}_{\mathbf{Z}_2} \\
&= \mathbf{U}_{\mathbf{Z}_1}^\top\dot{\mathbf{U}}_{\mathbf{Z}_1}\mathbf{D}_{\boldsymbol{\alpha}} + \dot{\mathbf{D}}_{\boldsymbol{\alpha}} + \mathbf{D}_{\boldsymbol{\alpha}}\dot{\mathbf{V}}_{\mathbf{Z}_2}^\top\mathbf{V}_{\mathbf{Z}_2}.
\end{aligned}
$$

Since $\mathbf{U}_{\mathbf{Z}_1}^\top\dot{\mathbf{U}}_{\mathbf{Z}_1} + \dot{\mathbf{U}}_{\mathbf{Z}_1}^\top\mathbf{U}_{\mathbf{Z}_1} = 0$ and $\mathbf{V}_{\mathbf{Z}_2}^\top\dot{\mathbf{V}}_{\mathbf{Z}_2} + \dot{\mathbf{V}}_{\mathbf{Z}_2}^\top\mathbf{V}_{\mathbf{Z}_2} = 0$, the matrices $\mathbf{U}_{\mathbf{Z}_1}^\top\dot{\mathbf{U}}_{\mathbf{Z}_1}$ and $\mathbf{V}_{\mathbf{Z}_2}^\top\dot{\mathbf{V}}_{\mathbf{Z}_2}$ are skew-symmetric (have 0 diagonal), and therefore the principal diagonals of the products $\mathbf{U}_{\mathbf{Z}_1}^\top\dot{\mathbf{U}}_{\mathbf{Z}_1}\mathbf{D}_{\boldsymbol{\alpha}}$ and $\mathbf{D}_{\boldsymbol{\alpha}}\dot{\mathbf{V}}_{\mathbf{Z}_2}^\top\mathbf{V}_{\mathbf{Z}_2}$ are also 0.

We define the linear operator $\mathrm{diag} : \mathbb{R}^{d \times k} \to \mathbb{R}^{d \times k}$ which maps any $A \in \mathbb{R}^{d \times k}$ to a matrix of the same dimensions whose principal diagonal contains the elements on the principal diagonal of $A$, and zeros otherwise. With this notation, we have that $\mathrm{diag}(\mathbf{U}_{\mathbf{Z}_1}^\top\dot{\boldsymbol{\alpha}}\mathbf{V}_{\mathbf{Z}_2}) = \dot{\mathbf{D}}_{\boldsymbol{\alpha}}$.

We similarly apply the orthogonal matrices to the left and right of the RHS of C.35, letting $\mathbf{R}' := \mathbf{U}_{\mathbf{Z}_1}^\top\mathbf{R}\mathbf{V}_{\mathbf{Z}_2} \in \mathbb{R}^{d \times k}$:

$$
\begin{aligned}
\mathrm{diag}\left(\mathbf{U}_{\mathbf{Z}_1}^\top\left(\mathbf{R} - \boldsymbol{\alpha}\mathbf{R}^\top\boldsymbol{\alpha}\right)\mathbf{V}_{\mathbf{Z}_2}\right) &= \mathrm{diag}\left(\mathbf{R}' - \mathbf{D}_{\boldsymbol{\alpha}}\mathbf{R}'^\top\mathbf{D}_{\boldsymbol{\alpha}}\right) \\
&= \mathrm{diag}\left(\mathbf{R}'\right) - \mathrm{diag}\left(\mathbf{D}_{\boldsymbol{\alpha}}\mathbf{R}'^\top\mathbf{D}_{\boldsymbol{\alpha}}\right) \\
&= \mathrm{diag}\left(\mathbf{R}'\right) - \mathbf{D}_{\boldsymbol{\alpha}}\mathrm{diag}\left(\mathbf{R}'\right)^\top\mathbf{D}_{\boldsymbol{\alpha}} \\
&= \left(I_d - \tilde{\mathbf{D}}_{\boldsymbol{\alpha}}^2\right)\mathrm{diag}\left(\mathbf{R}'\right).
\end{aligned}
$$

Therefore it holds that:

$$\dot{\mathbf{D}}_{\boldsymbol{\alpha}} = \left(I_d - \tilde{\mathbf{D}}_{\boldsymbol{\alpha}}^2\right)\mathrm{diag}\left(\mathbf{R}'\right) \tag{C.36}$$

We now wish to arrive at an expression in $\boldsymbol{\beta}$. From the definition of $\boldsymbol{\alpha}$ it holds that $\boldsymbol{\alpha} = (\mathbf{Z}_1\mathbf{Z}_1^\top)^{-1}\boldsymbol{\beta}$ and, in conjunction with the first part of Theorem C.11 it holds that $\mathbf{D}_{\boldsymbol{\alpha}} = (\tilde{\mathbf{D}}_{\mathbf{Z}_1}^2)^{-1}\mathbf{D}_{\boldsymbol{\beta}}$. Therefore, the LHS of (C.36) becomes:

$$\dot{\mathbf{D}}_{\boldsymbol{\alpha}} = (\tilde{\mathbf{D}}_{\mathbf{Z}_1}^2)^{-1}\dot{\mathbf{D}}_{\boldsymbol{\beta}} + (\tilde{\mathbf{D}}_{\mathbf{Z}_1}^2)^{-1}\dot{\mathbf{D}}_{\boldsymbol{\beta}}. \tag{C.37}$$

For computing $(\tilde{\mathbf{D}}_{\mathbf{Z}_1}^2)^{-1}$ it is perhaps easiest to proceed as we did with $\boldsymbol{\alpha}$.

$$
\begin{aligned}
\mathbf{U}_{\mathbf{Z}_1}^\top\overbrace{(\mathbf{Z}_1\mathbf{Z}_1^\top)^{-1}}^{\bullet}\mathbf{U}_{\mathbf{Z}_1} &= -\mathbf{U}_{\mathbf{Z}_1}^\top(\mathbf{Z}_1\mathbf{Z}_1^\top)^{-1}\overbrace{(\mathbf{Z}_1\mathbf{Z}_1^\top)}^{\bullet}(\mathbf{Z}_1\mathbf{Z}_1^\top)^{-1}\mathbf{U}_{\mathbf{Z}_1} \\
&= -(\tilde{\mathbf{D}}_{\mathbf{Z}_1}^2)^{-1}\left(\mathbf{D}_{\boldsymbol{\beta}}\mathbf{R}'^\top + \mathbf{R}'\mathbf{D}_{\boldsymbol{\beta}}^\top\right)(\tilde{\mathbf{D}}_{\mathbf{Z}_1}^2)^{-1},
\end{aligned}
$$

and therefore

$$
\begin{aligned}
(\tilde{\mathbf{D}}_{\mathbf{Z}_1}^2)^{-1} &= \mathrm{diag}\left(\mathbf{U}_{\mathbf{Z}_1}^\top\overbrace{(\mathbf{Z}_1\mathbf{Z}_1^\top)^{-1}}^{\bullet}\mathbf{U}_{\mathbf{Z}_1}\right) \\
&= -(\tilde{\mathbf{D}}_{\mathbf{Z}_1}^2)^{-1}\left(\mathbf{D}_{\boldsymbol{\beta}}\mathrm{diag}\left(\mathbf{R}'\right)^\top + \mathrm{diag}\left(\mathbf{R}'\right)\mathbf{D}_{\boldsymbol{\beta}}^\top\right)(\tilde{\mathbf{D}}_{\mathbf{Z}_1}^2)^{-1} \\
&= -2(\tilde{\mathbf{D}}_{\mathbf{Z}_1}^2)^{-2}\mathbf{D}_{\boldsymbol{\beta}}\mathrm{diag}\left(\mathbf{R}'\right)^\top.
\end{aligned}
$$

Putting everything back together in (C.37) we have that

$$
\begin{aligned}
\dot{\mathbf{D}}_{\boldsymbol{\alpha}} &= -2(\tilde{\mathbf{D}}_{\mathbf{Z}_1}^2)^{-2}\mathbf{D}_{\boldsymbol{\beta}}\mathrm{diag}\left(\mathbf{R}'\right)^\top\mathbf{D}_{\boldsymbol{\beta}} + (\tilde{\mathbf{D}}_{\mathbf{Z}_1}^2)^{-1}\dot{\mathbf{D}}_{\boldsymbol{\beta}} \tag{C.38} \\
&= -2(\tilde{\mathbf{D}}_{\mathbf{Z}_1}^2)^{-2}\tilde{\mathbf{D}}_{\boldsymbol{\beta}}^2\mathrm{diag}\left(\mathbf{R}'\right) + (\tilde{\mathbf{D}}_{\mathbf{Z}_1}^2)^{-1}\dot{\mathbf{D}}_{\boldsymbol{\beta}}. \tag{C.39}
\end{aligned}
$$

Finally, for dealing with the RHS of (C.36), we recall equation (C.12) which is implies that $2\gamma(\tilde{\mathbf{D}}_{\mathbf{Z}_1}^2)^{-1} = I_d - \tilde{\mathbf{D}}_{\boldsymbol{\alpha}}^2$. Putting together (C.39) and (C.36), we get that:

$$-2(\tilde{\mathbf{D}}_{\mathbf{Z}_1}^2)^{-2}\tilde{\mathbf{D}}_{\boldsymbol{\beta}}^2 \operatorname{diag}(\mathbf{R}') + (\tilde{\mathbf{D}}_{\mathbf{Z}_1}^2)^{-1}\dot{\mathbf{D}}_{\boldsymbol{\beta}} = 2\gamma(\tilde{\mathbf{D}}_{\mathbf{Z}_1}^2)^{-1}\operatorname{diag}(\mathbf{R}'), \qquad (C.40)$$

Therefore, we have the following string of implications:

$$(C.40) \iff \frac{1}{2}\left[\gamma I + \tilde{\mathbf{D}}_{\boldsymbol{\beta}}^2(\tilde{\mathbf{D}}_{\mathbf{Z}_1}^2)^{-1}\right]^{-1}\dot{\mathbf{D}}_{\boldsymbol{\beta}} = \operatorname{diag}(\mathbf{R}')$$

$$\iff \frac{1}{2}\tilde{\mathbf{D}}_{\mathbf{Z}_1}^2\left[\gamma(\tilde{\mathbf{D}}_{\mathbf{Z}_1}^2) + \tilde{\mathbf{D}}_{\boldsymbol{\beta}}^2\right]^{-1}\dot{\mathbf{D}}_{\boldsymbol{\beta}} = \operatorname{diag}(\mathbf{R}')$$

$$\iff \frac{1}{2}(\gamma I_d + \sqrt{\gamma^2 I_d + \tilde{\mathbf{D}}_{\boldsymbol{\beta}}^2})\left[\gamma^2 I_d + \tilde{\mathbf{D}}_{\boldsymbol{\beta}}^2 + \gamma\sqrt{\gamma^2 I_d + \tilde{\mathbf{D}}_{\boldsymbol{\beta}}^2}\right]^{-1}\dot{\mathbf{D}}_{\boldsymbol{\beta}} = \operatorname{diag}(\mathbf{R}')$$

$$\iff \frac{1}{2}\left[\sqrt{\gamma^2 I_d + \tilde{\mathbf{D}}_{\boldsymbol{\beta}}^2}\right]^{-1}\dot{\mathbf{D}}_{\boldsymbol{\beta}} = \operatorname{diag}(\mathbf{R}')$$

$$\implies \frac{d\frac{1}{2}\sinh^{-1}(\frac{1}{\gamma}\mathbf{D}_{\boldsymbol{\beta}})}{dt} = \operatorname{diag}(\mathbf{R}').$$

As a final step, we remark that $\operatorname{diag}(\mathbf{R}') = -\nabla_{\mathbf{D}_{\boldsymbol{\beta}}}\mathcal{L}(\boldsymbol{\beta})$ and that $\nabla_{\mathbf{D}_{\boldsymbol{\beta}}}\operatorname{tr}\left(\frac{1}{2}\mathbf{D}_{\boldsymbol{\beta}}\sinh^{-1}(\mathbf{D}_{\boldsymbol{\beta}}/\gamma) - \sqrt{\mathbf{D}_{\boldsymbol{\beta}}^2 + \gamma^2}\right) = \sinh^{-1}(\frac{1}{\gamma}\mathbf{D}_{\boldsymbol{\beta}})$. $\qquad\square$

### C.1 Extensions and further experiments

In this subsection, we describe how the insights from our analysis can be extend to relaxed assumptions on initialization, discrete gradient descent and non-linear activations.

**Perturbations from assumption on initialization.** Our analysis is contingent upon two assumptions regarding the initialization shape. The first assumption concerns the orthogonality of the initial feature layer $\boldsymbol{W}_1(0)$. In Figure 1a, we explored a scenario where the feature layer is initialized with a random Gaussian matrix, yet the evolution of singular values closely aligns with our theoretical analysis for orthogonal initialization. In Figure 4, we demonstrate the impact of zero initialization for the weight layer $\boldsymbol{W}_2$. We maintain the same experimental setup as in Figure 1a, but employ a random Gaussian initialization for $\boldsymbol{W}_2$ with variance scales of $10^{-2}$ and $10^{-3}$, while initializing $\boldsymbol{W}_1$ with a variance of $10^{-3}$. In this context, the evolution of singular values continues to adhere to the predicted trend from our analysis.

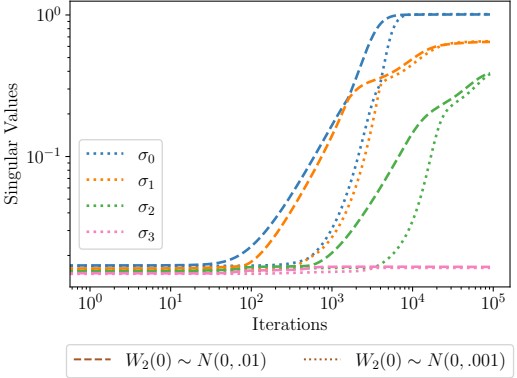

Figure 4: The time evolution of singular values of the hidden layer weights of a 2-layer linear network when trained with gradient flow initialized with Gaussian random variables and non-zero $\boldsymbol{W}_2$.

**Discrete step size.** Here we present a simpler problem to show how we can go beyond continuous time analysis. Consider the problem with $l = d, k = 1, \boldsymbol{W}_1, \boldsymbol{W}_2 = \boldsymbol{W}, \mathbf{a}$ and $\boldsymbol{W}$ is initialized with

*I*. The evolution of $W, a$ with a learning rate $\eta$ can be written as

$$\boldsymbol{W}_{t+1} = \boldsymbol{W}_t + \eta \boldsymbol{R}_t \mathbf{a}_t^\top,$$
$$\mathbf{a}_{t+1} = \mathbf{a}_t + \eta \boldsymbol{W}_t^\top \mathbf{a}_t,$$
$$\boldsymbol{R}_{t+1} = \boldsymbol{R}_t - \eta \left( X^\top X \right) \left( \boldsymbol{W}_t \boldsymbol{W}_t^\top + \|\mathbf{a}_t\|^2 \right) R_t - \eta^2 \left( \mathbf{a}_t^\top \boldsymbol{W}_t \boldsymbol{R}_t \right) \left( X^\top X \right) \boldsymbol{R}_t.$$

If we further assume that $X^\top X = I$, then it can be shown that $\boldsymbol{R}_t, \mathbf{a}_t$ only grow in norm and do not change in direction. So the update of $\boldsymbol{W}_t$ is always aligned with the rank-1 matrix $\boldsymbol{R}_0 \mathbf{a}_0^\top$. Hence for small initialization, the final $\boldsymbol{W}_\infty$ is approximately a rank-1 matrix. This presents a way forward for the discrete step-size case and orthogonal data. With further analysis, we think this can be generalized to any data matrix satisfying the RIP conditions. This is not included in the main paper due to restrictive assumptions on the data but we will include these comments in the appendix.

**Non-linear activations.** We consider the same teacher-student setup as in the Figure 2. To completely characterize the dynamics and the final weight parameters is a challenging problem. The characterization of the dynamics at the small scale of initialization is also absent (for any general data matrix), Boursier et al. [2022] solves this in the case of orthogonal data. To study even this simple case of two neurons, one has to study the dynamics in two phases where one jumps from zero initialization (a saddle point) to another saddle and then further converge to zero train loss as seen in the Figure 5a. It is difficult to precisely characterize this intermediate saddle. With some careful additional work, we believe that our analysis can capture Phase 1 (where you jump to the first saddle) of the dynamics where you approximately learn rank 1 matrix, see Figure 5b, for general data matrices extending the current understanding. We hope this briefly sketches a way forward for ReLU networks.

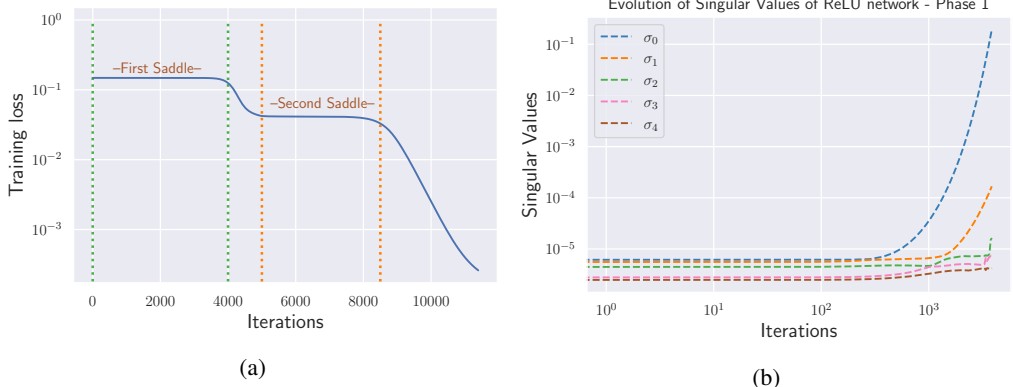

(a)

(b)

Figure 5: (a) The training curve of the teacher-student network which follows a saddle-to-saddle dynamics. (b) The time evolution of singular values of the hidden layer weights of a 2-layer ReLU network when trained with gradient flow. The plot represents Phase 1 of the training where you first learn a (approximately) rank-1 hidden layer.

## D  Noise Dynamics

**Noise Model.** Here we consider the scalar case ($k = 1$), abusing the notation $\boldsymbol{W} = \boldsymbol{W}_1, \mathbf{a} = \boldsymbol{W}_2$. The gradients of the loss $\mathcal{L}(\boldsymbol{W}, \mathbf{a})$ are

$$\nabla_{\boldsymbol{W}} \mathcal{L} = -X^\top (Y - X\boldsymbol{W}\mathbf{a}) \mathbf{a}^\top, \quad \nabla_{\mathbf{a}} \mathcal{L} = -\boldsymbol{W}^\top X^\top (Y - X\boldsymbol{W}\mathbf{a}).$$

When the labels or outputs are doped with a noise of magnitude $\delta > 0$, i.e., adding $\varepsilon \sim \delta \mathrm{N}(0, \mathbf{I}_n)$. Now the gradients computed after doping with this label noise are

$$\nabla_{\boldsymbol{W}} \tilde{\mathcal{L}} = -X^\top (Y + \varepsilon - X\boldsymbol{W}\mathbf{a}) \mathbf{a}^\top, \quad \nabla_{\mathbf{a}} \tilde{\mathcal{L}} = -\boldsymbol{W}^\top X^\top (Y + \varepsilon - X\boldsymbol{W}\mathbf{a}).$$
$$\nabla_{\boldsymbol{W}} \tilde{\mathcal{L}} = \nabla_{\boldsymbol{W}} \mathcal{L} - X^\top \varepsilon \mathbf{a}^\top, \quad \nabla_{\mathbf{a}} \tilde{\mathcal{L}} = \nabla_{\mathbf{a}} \mathcal{L} - \boldsymbol{W}^\top X^\top \varepsilon.$$

Thus label noise gradient descent with step size $\eta$ and with added label noise $\varepsilon_t$ at each iteration writes

$$\boldsymbol{W}_{t+1} = \boldsymbol{W}_t - \eta \left( \nabla_{\boldsymbol{W}} \mathcal{L} \left( \boldsymbol{W}_t, \mathbf{a}_t \right) - X^\top \varepsilon_t \mathbf{a}_t^\top \right).$$

The continuous time version of this SDE writes,

$$\mathrm{d}\boldsymbol{W} = -\eta \nabla_{\boldsymbol{W}} \mathcal{L} \left( \boldsymbol{W}, \mathbf{a} \right) \mathrm{d}t + \eta \delta X^\top \mathrm{d}\boldsymbol{B}_t \mathbf{a}^\top.$$

Now, we consider the large noise regime, where the dominating term in the SDE is the diffusion term. Therefore we consider the SDE,

$$\mathrm{d}\boldsymbol{W} = \eta \delta X^\top \mathrm{d}\boldsymbol{B}_t \mathbf{a}^\top,$$

where $\boldsymbol{B}_t$ is a $n$-dimensional Brownian motion. Note that we can get rid of the term $\eta \delta$ by re-scaling time by a constant factor. Similar steps for the evolution of $\mathbf{a}$ gives the SDE,

$$\mathrm{d}\boldsymbol{W} = X^\top \mathrm{d}\boldsymbol{B}_t \mathbf{a}^\top, \quad \mathrm{d}\mathbf{a} = \boldsymbol{W}^\top X^\top \mathrm{d}\boldsymbol{B}_t.$$

Now consider the compact SVD decomposition of $X$, i.e., $X = UDV^\top$, where $U, D \in \mathbb{R}^{n \times n}, V \in \mathbb{R}^{d \times n}$ and $UU^\top = U^\top U = V^\top V = \mathbf{I}_n$.

$$\mathrm{d}\boldsymbol{W} = VDU^\top \mathrm{d}\boldsymbol{B}_t \mathbf{a}^\top, \quad \mathrm{d}\mathbf{a} = \boldsymbol{W}^\top VDU^\top \mathrm{d}\boldsymbol{B}_t,$$

$$\mathrm{d}V^\top \boldsymbol{W} = D \left( U^\top \mathrm{d}\boldsymbol{B}_t \right) \mathbf{a}^\top, \quad \mathrm{d}\mathbf{a} = \left( V^\top \boldsymbol{W} \right)^\top D \left( U^\top \mathrm{d}\boldsymbol{B}_t \right).$$

Using Levy's characterization $\boldsymbol{Y}_t = U^\top \boldsymbol{B}_t$ is a Brownian motion, since $U$ is an orthogonal matrix. Let $\tilde{\boldsymbol{W}} = V^\top \boldsymbol{W}$, then

$$\mathrm{d}\tilde{\boldsymbol{W}} = D\mathrm{d}\boldsymbol{Y}_t \mathbf{a}^\top, \quad \mathrm{d}\mathbf{a} = \tilde{\boldsymbol{W}}^\top D\mathrm{d}\boldsymbol{Y}_t.$$

Here we consider $D = \mathbf{I}$ and change the notation to get the SDE 4.1 below. Our results can be extended to any diagonal matrix $D$.

$$\mathrm{d}\boldsymbol{W} = (\mathrm{d}\boldsymbol{B}_t)\, \mathbf{a}^\top \quad , \quad \mathrm{d}\mathbf{a} = \boldsymbol{W}^\top \mathrm{d}\boldsymbol{B}_t.$$

**Proposition 4.1.** *The dynamics* (4.1) *has the following convergence properties*

(a) **Variance explosion.** *The variance of the norms of $\boldsymbol{W}$, $\mathbf{a}$ explode, i.e.,*

$$\lim_{t \to \infty} \mathbb{E}\left[ \left\| \boldsymbol{W}(t) \right\|^2 \right] \to \infty \quad , \quad \lim_{t \to \infty} \mathbb{E}\left[ \left\| \mathbf{a}(t) \right\|^2 \right] \to \infty.$$

(b) **Scale divergence.** *For $d \geq 5$, for any $\alpha > 0$, we have that,*

$$\lim_{t \to \infty} \mathbb{E}\left[ \left\| \boldsymbol{W}(t) \right\|^\alpha \right] \to \infty \quad , \quad \lim_{t \to \infty} \mathbb{E}\left[ \left\| \mathbf{a}(t) \right\|^\alpha + \left\| \bar{\mathbf{a}}(t) \right\|^\alpha \right] \to \infty.$$

*where $\bar{\mathbf{a}} := e^{-t} \int\limits_0^t e^s \mathbf{a}(s)\mathrm{d}s$ is the exponential moving average of $\mathbf{a}$.*

(c) **Alignment - spectral bias.** *Denote the $i^{th}$ row of $\boldsymbol{W}$ as $\boldsymbol{w}_i$. Using $[\boldsymbol{w}_i, \mathbf{a}] \overset{\mathrm{def}}{=} \boldsymbol{w}_i \mathbf{a}^\top - \mathbf{a}\boldsymbol{w}_i^\top$,*

$$\lim_{t \to \infty} \mathbb{E}\left[ \left| [\boldsymbol{w}_i, \mathbf{a}] \right| \right] \to 0.$$

*Proof.* Consider the noise model Eq. (4.1),

$$\mathrm{d}\boldsymbol{W} = (\mathrm{d}\boldsymbol{B}_t)\, \mathbf{a}^\top \quad , \quad \mathrm{d}\mathbf{a} = \boldsymbol{W}^\top \mathrm{d}\boldsymbol{B}_t, \tag{D.1}$$

**Variance.** Let $\boldsymbol{w}_i$ be the $i^{th}$ column of $\boldsymbol{W}$ and $\boldsymbol{a}_i$ be the $i^{th}$ coordinate of $\mathbf{a}$. For any $i \in [n]$, the diffusion of the quantities can be separately written as

$$\mathrm{d}\boldsymbol{w}_i = \boldsymbol{a}_i \mathrm{d}\boldsymbol{B}_t, \quad \mathrm{d}\boldsymbol{a}_i = \langle \boldsymbol{w}_i, \mathrm{d}\boldsymbol{B}_t \rangle.$$

Using the Itô chain rule,

$$\mathrm{d}\left\| \boldsymbol{w}_i \right\|^2 = 2 \langle \boldsymbol{w}_i, \mathrm{d}\boldsymbol{w}_i \rangle + \langle \mathrm{d}\boldsymbol{w}_i, \mathrm{d}\boldsymbol{w}_i \rangle,$$

$$= d\left\| \boldsymbol{a}_i \right\|^2 \mathrm{d}t + 2\boldsymbol{a}_i \langle \boldsymbol{w}_i, \mathrm{d}\boldsymbol{B}_t \rangle.$$

Similarly,

$$\mathrm{d}a_i^2 = 2a_i\mathrm{d}a_i + \mathrm{d}a_i\mathrm{d}a_i = 2a_i\langle \boldsymbol{w}_i, \mathrm{d}\boldsymbol{B}_t\rangle + \|\boldsymbol{w}_i\|^2\mathrm{d}t.$$

Note that $\langle \boldsymbol{w}_i, \mathrm{d}\boldsymbol{B}_t\rangle \sim \|\boldsymbol{w}_i\|\mathrm{d}\tilde{\boldsymbol{B}}_t$ for some one-dimensional Brownian motion $(\tilde{\boldsymbol{B}}_t)_{t\geq 0}$.

$$\mathrm{d}a_i^2 = \|\boldsymbol{w}_i\|^2\mathrm{d}t + 2a_i\|\boldsymbol{w}_i\|\mathrm{d}\tilde{\boldsymbol{B}}_t, \quad \mathrm{d}\|\boldsymbol{w}_i\|^2 = d\|\boldsymbol{a}_i\|^2\mathrm{d}t + 2a_i\|\boldsymbol{w}_i\|\mathrm{d}\tilde{\boldsymbol{B}}_t.$$

Let $\boldsymbol{v} := \boldsymbol{a}_i, \boldsymbol{u} := \|\boldsymbol{w}_i\|$, using this notation we get,

$$\mathrm{d}\boldsymbol{u}^2 = d\boldsymbol{v}^2\mathrm{d}t + 2\boldsymbol{u}\boldsymbol{v}\mathrm{d}\tilde{\boldsymbol{B}}_t, \quad \mathrm{d}\boldsymbol{v}^2 = \boldsymbol{u}^2\mathrm{d}t + 2\boldsymbol{u}\boldsymbol{v}\mathrm{d}\tilde{\boldsymbol{B}}_t. \tag{D.2}$$

Let $\boldsymbol{u}_0 := \mathbb{E}\left[\boldsymbol{u}^2\right], \boldsymbol{v}_0 := \mathbb{E}\left[\boldsymbol{v}^2\right]$. Using the Dynkins formula, taking the expectation, we get,

$$\mathrm{d}\mathbb{E}\left[\boldsymbol{u}^2\right] = d\mathbb{E}\left[\boldsymbol{v}^2\right]\mathrm{d}t, \quad \mathrm{d}\mathbb{E}\left[\boldsymbol{v}^2\right] = \mathbb{E}\left[\boldsymbol{u}^2\right],$$
$$\mathrm{d}\boldsymbol{u}_0 = d\boldsymbol{v}_0\mathrm{d}t, \quad \mathrm{d}\boldsymbol{v}_0 = \boldsymbol{u}_0\mathrm{d}t.$$

This system can be transformed into,

$$\mathrm{d}\left(\boldsymbol{u}_0 + \sqrt{d}\boldsymbol{v}_0\right) = \sqrt{d}\left(\boldsymbol{u}_0 + \sqrt{n}\boldsymbol{v}_0\right)\mathrm{d}t.$$

Solving the above ODE we get,

$$\left(\boldsymbol{u}_0 + \sqrt{d}\boldsymbol{v}_0\right) = c_0 e^{\sqrt{d}t}, \quad \text{where } c_0 = \boldsymbol{u}_0(0) + \sqrt{d}\boldsymbol{v}_0(0).$$

Similarly,

$$\mathrm{d}\left(\boldsymbol{u}_0 - \sqrt{d}\boldsymbol{v}_0\right) = -\sqrt{d}\left(\boldsymbol{u}_0 - \sqrt{d}\boldsymbol{v}_0\right)\mathrm{d}t,$$
$$\left(\boldsymbol{u}_0 - \sqrt{d}\boldsymbol{v}_0\right) = c_1 e^{-\sqrt{d}t}, \quad \text{where } c_1 = \boldsymbol{u}_0(0) - \sqrt{d}\boldsymbol{v}_0(0).$$

$$\boldsymbol{u}_0 = \frac{1}{2}\left[c_0 e^{\sqrt{d}t} + c_1 e^{-\sqrt{d}t}\right], \quad \boldsymbol{v}_0 = \frac{1}{2\sqrt{d}}\left[c_0 e^{\sqrt{d}t} - c_1 e^{-\sqrt{d}t}\right].$$

Taking the limit proves the first part of the result.

**Scale.** From Eq. D.2, we have,

$$\mathrm{d}\left(\boldsymbol{u}^2 + \sqrt{n}\boldsymbol{v}^2\right) = \sqrt{n}(\boldsymbol{u}^2 + \sqrt{d}\boldsymbol{v}^2) + 2\left(\sqrt{d}+1\right)\boldsymbol{u}\boldsymbol{v}\mathrm{d}\tilde{\boldsymbol{B}}_t.$$

Again using the Itô chain rule, we get,

$$\mathrm{d}\left(\boldsymbol{u}^2 + \sqrt{d}\boldsymbol{v}^2\right)^{\boldsymbol{\alpha}} = \boldsymbol{\alpha}\left(\boldsymbol{u}^2 + \sqrt{d}\boldsymbol{v}^2\right)^{\boldsymbol{\alpha}-1}\mathrm{d}\left(\boldsymbol{u}^2 + \sqrt{d}\boldsymbol{v}^2\right)$$
$$+ \frac{1}{2}\boldsymbol{\alpha}(\boldsymbol{\alpha}-1)\left(\boldsymbol{u}^2 + \sqrt{d}\boldsymbol{v}^2\right)^{\boldsymbol{\alpha}-2}\mathrm{d}\left(\boldsymbol{u}^2 + \sqrt{n}\boldsymbol{v}^2\right)\mathrm{d}\left(\boldsymbol{u}^2 + \sqrt{n}\boldsymbol{v}^2\right),$$
$$= \boldsymbol{\alpha}\left(\boldsymbol{u}^2 + \sqrt{d}\boldsymbol{v}^2\right)^{\boldsymbol{\alpha}-1}\left[\sqrt{d}(\boldsymbol{u}^2 + \sqrt{d}\boldsymbol{v}^2)\mathrm{d}t + 2\left(\sqrt{d}+1\right)\boldsymbol{u}\boldsymbol{v}\mathrm{d}\tilde{\boldsymbol{B}}_t\right]$$
$$+ \frac{1}{2}\boldsymbol{\alpha}(\boldsymbol{\alpha}-1)\left(\boldsymbol{u}^2 + \sqrt{d}\boldsymbol{v}^2\right)^{\boldsymbol{\alpha}-2}\left(4\left(\sqrt{d}+1\right)^2\boldsymbol{u}^2\boldsymbol{v}^2\right)\mathrm{d}t,$$

The drift term is

$$\boldsymbol{\alpha}\sqrt{n}\left(\boldsymbol{u}^2 + \sqrt{d}\boldsymbol{v}^2\right)^{\boldsymbol{\alpha}} + 2\boldsymbol{\alpha}(\boldsymbol{\alpha}-1)\left(\sqrt{d}+1\right)^2\left(\boldsymbol{u}^2 + \sqrt{n}\boldsymbol{v}^2\right)^{\boldsymbol{\alpha}-2}\boldsymbol{u}^2\boldsymbol{v}^2$$
$$= \boldsymbol{\alpha}\left(\boldsymbol{u}^2 + \sqrt{n}\boldsymbol{v}^2\right)^{\boldsymbol{\alpha}}\left[\sqrt{d} + 2(\boldsymbol{\alpha}-1)\left(\sqrt{d}+1\right)^2\frac{\boldsymbol{u}^2\boldsymbol{v}^2}{\left(\boldsymbol{u}^2 + \sqrt{d}\boldsymbol{v}^2\right)^2}\right].$$

Again using the Dynkins formula for the evolution of expectation, we have,

$$\mathrm{d}\mathbb{E}\left[\left(\boldsymbol{u}^2 + \sqrt{d}\boldsymbol{v}^2\right)^{\boldsymbol{\alpha}}\right] = \mathbb{E}\left[\boldsymbol{\alpha}\left(\boldsymbol{u}^2 + \sqrt{d}\boldsymbol{v}^2\right)^{\boldsymbol{\alpha}}\left[\sqrt{d} + 2(\boldsymbol{\alpha}-1)\left(\sqrt{d}+1\right)^2\frac{\boldsymbol{u}^2\boldsymbol{v}^2}{\left(\boldsymbol{u}^2 + \sqrt{d}\boldsymbol{v}^2\right)^2}\right]\right]\mathrm{d}t.$$
$$\tag{D.3}$$

For any function,

$$g(y) = \frac{y}{\left(y + \sqrt{d}\right)^2},$$

attains its maximum value at $y = \sqrt{d}$, i.e., $g(\sqrt{n}) = 1/(4\sqrt{d})$. Note that $\alpha < 1$ and we have

$$2(\alpha - 1)\left(\sqrt{n}+1\right)^2 \frac{\boldsymbol{u}^2\boldsymbol{v}^2}{\left(\boldsymbol{u}^2 + \sqrt{n}\boldsymbol{v}^2\right)^2} \geq 2(\alpha - 1)\left(\sqrt{n}+1\right)^2 g(\sqrt{n}) = (\alpha - 1)\frac{\left(\sqrt{n}+1\right)^2}{2\sqrt{n}}.$$

The drift can be lower bounded as the following,

$$\alpha\sqrt{n}\left(\boldsymbol{u}^2 + \sqrt{d}\boldsymbol{v}^2\right)^{\alpha} + 2\alpha(\alpha - 1)\left(\sqrt{d}+1\right)^2\left(\boldsymbol{u}^2 + \sqrt{n}\boldsymbol{v}^2\right)^{\alpha-2}\boldsymbol{u}^2\boldsymbol{v}^2$$

$$\geq \alpha\left(\boldsymbol{u}^2 + \sqrt{d}\boldsymbol{v}^2\right)^{\alpha}\left[\sqrt{d} + (\alpha - 1)\frac{\left(\sqrt{d}+1\right)^2}{2\sqrt{d}}\right].$$

For $d \geq 5$ and $0 < \alpha < 1$, we have $c_0 > 0$,

$$\left[\sqrt{d} + (\alpha - 1)\frac{\left(\sqrt{d}+1\right)^2}{2\sqrt{d}}\right] > c_0.$$

Using the above expression in Eq. D.3,

$$d\mathbb{E}\left[\left(\boldsymbol{u}^2 + \sqrt{d}\boldsymbol{v}^2\right)^{\alpha}\right] \geq \alpha c_0 \mathbb{E}\left[\left(\boldsymbol{u}^2 + \sqrt{d}\boldsymbol{v}^2\right)^{\alpha}\right]dt.$$

Taking the limit,

$$\lim_{t \to \infty} \mathbb{E}\left[\left(\boldsymbol{u}^2 + \sqrt{d}\boldsymbol{v}^2\right)^{\alpha}\right] \to \infty.$$

From the SDE (D.2), we obtain the following process with only diffusion,

$$d\left(\boldsymbol{u}^2 - \boldsymbol{v}^2\right) = (d\boldsymbol{v}^2 - \boldsymbol{u}^2)dt,$$

$$d\left(\boldsymbol{u}^2 - \boldsymbol{v}^2\right) + \left(\boldsymbol{u}^2 - \boldsymbol{v}^2\right)dt = (n-1)\boldsymbol{v}^2,$$

$$e^t d\left(\boldsymbol{u}^2 - \boldsymbol{v}^2\right) + e^t\left(\boldsymbol{u}^2 - \boldsymbol{v}^2\right)dt = (n-1)e^t\boldsymbol{v}^2 dt,$$

$$de^t\left(\boldsymbol{u}^2 - \boldsymbol{v}^2\right) = (n-1)e^t\boldsymbol{v}^2 dt,$$

$$\left(\boldsymbol{u}^2(t_2) - \boldsymbol{v}^2(t_2)\right) = e^{-(t_2-t_1)}\left(\boldsymbol{u}^2(t_1) - \boldsymbol{v}^2(t_1)\right) + (n-1)\int_{t_1}^{t_2} e^{-(t_2-t)}\boldsymbol{v}^2(t)dt,$$

$$\left(\boldsymbol{u}^2(t_2) + \sqrt{n}\boldsymbol{v}^2(t_2)\right) = e^{-(t_2-t_1)}\left(\boldsymbol{u}^2(t_1) - \boldsymbol{v}^2(t_1)\right) + \sqrt{n}\boldsymbol{v}^2(t_2) + (n-1)e^{-t_2}\int_{t_1}^{t_2} e^t\boldsymbol{v}^2(t)dt,$$

Denote $C(t) := \boldsymbol{u}^2(t) + \sqrt{d}\boldsymbol{v}^2(t)$, using the fact that $\boldsymbol{u}^2(t_1) - \boldsymbol{v}^2(t_1) \leq C(t_1)$,

$$C(t_2) \leq e^{-(t_2-t_1)}C(t_1) + \sqrt{d}\boldsymbol{v}^2(t_2) + (d-1)e^{-t_2}\int_{t_1}^{t_2} e^t\boldsymbol{v}^2(t),$$

With $t_1 = 0$, using the notation $\tilde{\boldsymbol{v}}^2(t) := e^{-t}\int_0^t e^s\boldsymbol{v}^2(s)ds$, to denote the exponential moving average.

For any time $t > 0$, we have,

$$C(t) \leq e^{-(t)}C(0) + d\left(\boldsymbol{v}^2(t) + \tilde{\boldsymbol{v}}^2(t)\right).$$

Raising to the power of $\alpha$,

$$C(t)^{\alpha} \leq \left[e^{-t}C(0) + d\left(\boldsymbol{v}^2(t) + \tilde{\boldsymbol{v}}^2(t)\right)\right]^{\alpha}.$$

For $a, b > 0$ and $0 < p < 1$, we have $(a + b)^p \leq a^p + b^p$. Using the inequality,

$$C(t)^\alpha \leq e^{-\alpha t} C(0)^\alpha + d^\alpha \left(v^2(t)\right)^\alpha + d^\alpha \left(\tilde{v}^2(t)\right)^\alpha.$$

Taking the expectation,

$$\mathbb{E}\left[C(t)^\alpha\right] \leq e^{-\alpha t}\mathbb{E}\left[C(0)^\alpha\right] + d^\alpha \mathbb{E}\left[\left(v^2(t)\right)^\alpha\right] + d^\alpha \mathbb{E}\left[\left(\tilde{v}^2(t)\right)^\alpha\right].$$

Now, we proceed by taking the limit,

$$\lim_{t\to\infty} \mathbb{E}\left[C(t)^\alpha\right] \leq e^{-\alpha t} \lim_{t\to\infty} \mathbb{E}\left[C(0)^\alpha\right] + d^\alpha \lim_{t\to\infty} \left[\mathbb{E}\left[\left(v^2(t)\right)^\alpha\right] + \mathbb{E}\left[\left(\tilde{v}^2(t)\right)^\alpha\right]\right].$$

Thus, we obtain,

$$\lim_{t\to\infty} \left(\mathbb{E}\left[\left(v^2(t)\right)^\alpha\right] + \mathbb{E}\left[\left(\tilde{v}^2(t)\right)^\alpha\right]\right) \to \infty,$$

where $\tilde{v}^2(t) := e^{-t} \int_0^t e^s v^2(s)\mathrm{d}s$ is the exponential moving average.

A similar computation for $u$ will yield,

$$\lim_{t\to\infty} \mathbb{E}\left[\left(u^2(t)\right)^\alpha\right] \to \infty.$$

Therefore for the limit of $a_i, \|w_i\|$, we have,

$$\lim_{t\to\infty} \mathbb{E}\left[\|w_i(t)\|^{2\alpha}\right] \to \infty, \quad \lim_{t\to\infty} \left(\mathbb{E}\left[\left(a_i^2(t)\right)^\alpha\right] + \mathbb{E}\left[\left(\bar{a}_i^2(t)\right)^\alpha\right]\right) \to \infty.$$

Now we proceed to combine the above results and obtain the result on $\mathbf{a}, \boldsymbol{W}$. Note that for $0 < \alpha < 1$, $x^\alpha$ is concave. Further using the Jensen's inequality, we have,

$$\left(\|\boldsymbol{W}\|^2\right)^\alpha = \left(\sum_{i=1}^d \|w_i\|^2\right)^\alpha \geq d^{1-\alpha} \sum_{i=1}^d \|w_i(t)\|^{2\alpha}.$$

Similar expression for $\mathbf{a}$ and taking the limit, we obtain, the result

$$\lim_{t\to\infty} \mathbb{E}\left[\|\boldsymbol{W}(t)\|^\alpha\right] \to \infty \quad, \quad \lim_{t\to\infty} \mathbb{E}\left[\|\mathbf{a}(t)\|^\alpha + \|\bar{\mathbf{a}}(t)\|^\alpha\right] \to \infty.$$

where $\bar{\mathbf{a}} := e^{-t} \int_0^t e^s \mathbf{a}(s)\mathrm{d}s$ is the exponential moving average of $\mathbf{a}$.

**Alignment.** Let $z_1, z_2 \ldots z_d$ be the rows of the matrix $\boldsymbol{W}$. Now the evolution of the rows can be written as

$$\dot{z}_i = \mathbf{a}(\mathrm{d}\boldsymbol{B}_t^i),$$

$$\mathrm{d}\mathbf{a} = \sum_{j=1}^d z_j \mathrm{d}\boldsymbol{B}_t^j.$$

For any two matrices, with same dimensions define $[u, v] \stackrel{\text{def}}{=} uv^\top - vu^\top$.

$$\mathrm{d}[z_i, \mathbf{a}] = \mathrm{d}\left(z_i \mathbf{a}^\top\right) - \mathrm{d}\left(\mathbf{a} z_i^\top\right),$$

Using the Itô chain rule,

$$d\left(z_i \mathbf{a}^\top\right) = dz_i \mathbf{a}^\top + z_i db^\top + dz_i d\mathbf{a},$$

$$= \mathbf{a}\mathbf{a}^\top (d\mathbf{B}_t^i) + z_i \left(\sum_{j=1}^{d} z_j^\top d\mathbf{B}_t^j\right) + \left(\mathbf{a}(d\mathbf{B}_t^i)\right)\left(\sum_{j=1}^{d} z_j^\top d\mathbf{B}_t^j\right),$$

$$= \mathbf{a}\mathbf{a}^\top (d\mathbf{B}_t^i) + \sum_{j=1}^{d} z_i z_j^\top d\mathbf{B}_t^j + \mathbf{a} z_i^\top dt,$$

$$d\left(\mathbf{a} z_i^\top\right) = \mathbf{a}\mathbf{a}^\top \left(d\mathbf{B}_t^i\right) + z_i \mathbf{a}^\top dt + \sum_{j=1}^{d} z_j z_i^\top d\mathbf{B}_t^j,$$

$$d[z_i, \mathbf{a}] = -[z_i, \mathbf{a}] dt + \sum_{i=1}^{d} [z_i, z_j] d\mathbf{B}_t^j,$$

From the above evolution, we have that,

$$d\mathbb{E}\left[[z_i, \mathbf{a}]\right] = -\mathbb{E}\left[[z_i, \mathbf{a}]\right] dt.$$

Hence, we have,

$$\mathbb{E}\left[[z_i, \mathbf{a}]\right] = [z_i(0), \mathbf{a}(0)] e^{-t}.$$

Let $e_i \stackrel{\text{def}}{=} [z_i, \mathbf{a}][k, l]$, be any $(kl)^{th}$ entry of the matrix. Similarly, let $c_{ij} \stackrel{\text{def}}{=} [z_i, z_j][k, l]$.

$$de_i = -e_i dt + \sum_j c_{ij} d\mathbf{B}_t^j,$$

$$de_i^2 = -2e_i \left(e_i dt + \sum_j c_{ij} d\mathbf{B}_t^j\right) + \sum_j c_{ij}^2 dt,$$

$$de_i^2 = -2e_i^2 dt + \sum_j c_{ij}^2 dt - 2e_i \sum_j c_{ij} d\mathbf{B}_t^j.$$

Again using the Ito formula and computing $(e_i^2)^\alpha$ for some $\alpha \in (0, 1)$, we get,

$$d\left(e_i^2\right)^\alpha = \alpha \left(e_i^2\right)^{\alpha-1} de_i^2 + \frac{1}{2}\alpha(\alpha-1)\left(e_i^2\right)^{\alpha-2} de_i^2 de_i^2,$$

$$d\left(e_i^2\right)^\alpha = \alpha \left(e_i^2\right)^{\alpha-1}\left[-2e_i^2 dt + \sum_j c_{ij}^2 dt - 2e_i \sum_j c_{ij} d\mathbf{B}_t^j\right] + \frac{1}{2}\alpha(\alpha-1)\left(e_i^2\right)^{\alpha-2} 4e_i^2 \left[\sum_j c_{ij}^2\right],$$

$$= -2\alpha \left(e_i^2\right)^\alpha dt + \alpha \left(e_i^2\right)^{\alpha-1}\sum_j c_{ij}^2 dt - 2\alpha \left(e_i^2\right)^{1-\alpha} e_i \sum_j c_{ij} d\mathbf{B}_t^j + \frac{1}{2}\alpha(\alpha-1)\left(e_i^2\right)^{\alpha-2} 4e_i^2 \left[\sum_j c_{ij}^2\right] dt,$$

$$= -2\alpha \left(e_i^2\right)^\alpha dt + \alpha(2\alpha-1)\left(e_i^2\right)^{\alpha-1}\sum_j c_{ij}^2 dt - 2\alpha \left(e_i^2\right)^{\alpha-1} e_i \sum_j c_{ij} d\mathbf{B}_t^j,$$

Taking $\alpha = 0.5$,

$$d|e_i| = -|e_i| dt - |e_i|^{-1} e_i \sum_j c_{ij} d\mathbf{B}_t^j.$$

Taking expectation, we get,

$$d\mathbb{E}\left[|e_i|\right] = -\mathbb{E}\left[|e_i|\right] dt.$$

Hence,

$$\mathbb{E}\left[|[z_i, \mathbf{a}]|\right] = |[z_i(0), \mathbf{a}(0)]| e^{-t}.$$

$\square$

