# OpenReview forum: "On the spectral bias of two-layer linear networks"
_NeurIPS.cc/2023/Conference — NeurIPS 2023 poster_

### Official Review · Reviewer_EWXT · 2023-07-06

**Soundness:** 3 good
**Presentation:** 3 good
**Contribution:** 3 good
**Rating:** 7
**Confidence:** 3

**Summary:**

The paper explores the convergence of gradient descent for two-layers linear neural networks.

**Strengths:**

The paper substantially advances the knowledge on the topic.

**Weaknesses:**

Missing reference: E. Boursier et al, Gradient flow dynamics of shallow ReLU networks for square loss and orthogonal inputs.
Please mention this reference as they already highlight the low-rank structure of the optimal solution in the nonlinear case.

**Questions:**

No questions

**Limitations:**

See above

---

> ### Author Rebuttal · Authors · 2023-08-09
>
> We thank the reviewer for the valuable feedback.
>
> > Missing reference: E. Boursier et. al., Gradient flow dynamics of shallow ReLU networks for square loss and orthogonal inputs. Please mention this reference as they already highlight the low-rank structure of the optimal solution in the nonlinear case.
>
> We thank the reviewer for pointing out the omission and will update our paper with reference [1] and its relation to our work. More precisely, we will include the following points:
>  - Our analysis can be viewed as a theoretical characterization of Phase 1 of [1], in which the neuron alignment happens. The main difference with the ReLU networks considered in [1] is that in the latter, the neurons are required to align along two distinct directions -- one for data with positive outputs and one for negative outputs. In our case (for linear activation), aligning in a single direction is enough (for k=1, i.e., scalar regression).
>   - On a technical note, our analysis broadens the setting of Phase 1 from [1] (in the context of linear networks) in the following way:  [1] works with a very small initialization scale, while our analysis holds for any scale of initialization and, in addition, we do not require the data to be orthogonal.
>
>  [1] E. Boursier et al, Gradient flow dynamics of shallow ReLU networks for square loss and orthogonal inputs

---

### Official Review · Reviewer_itua · 2023-07-07

**Soundness:** 3 good
**Presentation:** 3 good
**Contribution:** 3 good
**Rating:** 6
**Confidence:** 3

**Summary:**

The manuscript studies the evolution of weights in a two layer *linear* NN under gradient flow subject to certain initialization. Results are provided that show how different initialization scales result in qualitatively different behavior of the learned weights.


**Strengths:**

The main points of the manuscript are relatively easy to follow and the variational characterization of the weights (Thm 3.1 part (iii)) is interesting, as is Corollary 3.2. The manuscript also seems to do a relatively good job positioning itself withing the broader landscape.


**Weaknesses:**

The main weaknesses of the manuscript (in the broader context, not in its self defined scope) are: (1) the assumptions and (2) the restriction to the linear activation function and gradient flow.

Regarding the former, I agree that the orthogonality condition on $W_1$ could likely be removed and is not essential. However, it seems that at least point (ii) in Thm 3.1 may crucially depend on the 0 initialization of $W_2.$ Certainly this is the case for analysis of the random features regime as Landweber iteration. What is not so clear in the manuscript is how sensitive (iii) in Thm 3.2 (which I think is the most interesting result) is to finding a solution of minimal norm. More generally, I think the manuscript would benefit from a bit more analysis/discussion of the sensitivity of the results to the assumptions.

I understand the restriction to linear activation functions is, at the moment, necessary to have results. Nevertheless, it is undoubtedly a weakness unless a more concrete connection to how it might enable analysis of the more general case is presented (which would also help articulate the potential impact). Such a connection could substantially strengthen the paper since it would help paint a path forward. Similarly, a brief discussion of the difficultly (or lack thereof) of analyzing gradient descent (rather than gradient flow) would be beneficial. (A quick/rough calculation suggests that there may be substantial difficulties based on additional terms that arise in the per step iteration, but maybe this is missing something.)

A more minor point:

- Characterizing the initialization of $W_2$ at 0 as a limiting case seems odd since there is potentially a substantiate difference between having no component in the null space of $XW_1$ and having something non-trivial. While in the "only learn $W_2$" setting the part in the null space would remain small that is less clear here. If $W_2$ is not initialized to 0 would an interpolating solution with near minimal norm be found or could the joint optimization over $W_1$ and $W_2$ inflate such a component.

**Questions:**

- Where are the characterizations of singular values of $W_2$? Appendix B.1 seems to be about the noise model?

- To make a comment in the weakness's section more concrete: how sensitive, if at all, is (iii) of Thm. 3.2 to the fact that the an interpolating solution with minimal norm is found?

**Limitations:**

- The restriction to a linear model is clearly quite limiting, though it is also clear that direct analysis of the more general setting is quite challenging. Whether or not analysis of the linear setting helps make progress when the activation function is something else is hard to predict.

---

> ### Author Rebuttal · Authors · 2023-08-09
>
> We thank the reviewer for the valuable feedback and interesting questions which help us in clarifying and improving the paper.
>
>  > However, it seems that at least point (ii) in Thm 3.1 may crucially depend on the 0 initialization of $W_2$.
> Certainly this is the case for analysis of the random features regime as Landweber iteration. What is not so clear in the manuscript is how sensitive (iii) in Thm 3.2 (which I think is the most interesting result) is to finding a solution of minimal norm. More generally, I think the manuscript would benefit from a bit more analysis/discussion of the sensitivity of the results to the assumptions.
>
> Thank you for this comment. We agree that to get precise formulations for Theorem 3.1 (ii) and (iii) the initialization of  $W_2$ to zero is required. Regarding how sensitive these results are to the condition $W_2(0) = 0$, both the bias towards low rank and the convergence to a minimum-norm interpolator are similarly sensitive to perturbations of W_2(0) away from 0. A brief sketch to support this claim is given below:
>
> Assume $l=d$ for simplicity. From our analysis, we have that $\alpha_{\infty} - \alpha_{0} \in \mathrm{span}(X)$. Using this property, we get the following expression for $\beta_{\infty}$:
> $$ \beta_{\infty}  \in \left(W_1^{\infty}(W_1^{\infty})^{\top}\right) \mathrm{span}(X) +  O(\|W_2(0)\|). $$
> Looking at the updates of $W_1$, we get that $W_1^{\infty} = W_1(0) + \mathrm{span}(X)$. So, $W_1^{\infty}(W_1^{\infty})^{\top} = W_1(0) W_1(0)^{\top} + \mathrm{span}(X)  $. We initialize such that $W_1(0) W_1(0)^{\top} = 2 \gamma  I$, hence $\beta_{\infty} \in \mathrm{span}(X) + O(\|W_2(0)\|)$. Using the invariance of the dynamics from Lemma~3.5,
> we can say that $W_1^{\top}W_1 = W_1(0)^{\top}W_1(0) + W_2W_2^{\top} - O(\|W_2(0)\|^2)  $. We will extend  the above pertubation analysis both empirically and analytically and discuss the implications in the revised version of the manuscript.
>
>
> > A brief discussion of the difficultly (or lack thereof) of analyzing gradient descent (rather than gradient flow) would be beneficial.
> We agree with the reviewer that a general extension to GD is difficult. Note that, with some restrictive assumptions on data, it is however possible to study the dynamics and characterize the limiting behaviour of the weights, in a manner that goes beyond tracking discretization errors. We provide here a sketch.
>
>
>
> Consider a simpler problem with $ l=d, k=1, W_1,W_2 = W,a$ and $W$ is initialized with $I$. The evolution of $W,a$ with a learning rate $\eta$ can be written as
> \begin{align*}
>     W_{t+1} &= W_t + \eta R_t a_t^{\top}, \\\\
>     a_{t+1} &= a_t + \eta W_t^{\top} a_{t}, \\\\
>     R_{t+1} &= R_t - \eta \left(X^{\top}X\right) ( W_t W_t^{\top} + \|a_{t}\|^2 ) R_{t} - \eta^2 \left( a_t^{\top} W_t r_t \right) \left(X^{\top}X\right) R_t.
>  \end{align*}
>  If we further assume that $X^{\top}X = I$, then it can be shown that $R_{t},a_{t}$ only grow in norm and do not change in direction. So the update of $W_{t}$ is always aligned with the rank-1 matrix $R_{0}a_{0}^{\top}$. Hence for small initialization, the final $W_{\infty}$ is
> approximately a rank-1 matrix. This analysis presents a way forward for the discrete step-size case and orthogonal data. With further work, we think this analysis can be generalized to any data matrix satisfying the RIP conditions. This result was not included in the main paper due to the restrictive assumptions on the data but we will include these comments in the appendix.
>
>   > Characterizing the initialization of $W_2$ at 0 as a limiting case seems odd since there is potentially a substantiate difference between having no component in the null space of $XW_1$ and having something non-trivial.
>  If $W_2$ is not initialized to 0 would an interpolating solution with near minimal norm be found or could the joint optimization over $W_1$ and $W_2$ inflate such a component.
>
> The perturbation analysis in response to the first question of the reviewer may shed some light on why initialization of $W_2$ at 0 can be seen as a limiting case. An alternative explanation involves the trajectories of gradient flow. Let $(W_1(t),W_2(t))\_{t \\geq 0}$ and  $(W_{1}^{\\epsilon}(t),W_{2}^{\\epsilon}(t))\_{t \\geq 0}$ be the trajectories of gradient flow when initialized at $(\\sqrt{2\\gamma}P, 0)$ and $(\\sqrt{2\\gamma}P, \\epsilon U )$, respectively, for orthogonal matrices $P,U$ (of appropriate dimensions). Let $\\theta(t) = (W_1(t),W_2(t))$ and $ \\theta^{\\epsilon} = \\left( W_{1}^{\\epsilon} (t),W_{2}^{\\epsilon} (t) \\right) $. For time $ t < \infty $, the difference in trajectories is tracked by:
> \begin{align*}
> \frac{d}{dt} \|\theta(t) - \theta_t^{\epsilon}(t) \|^2 &= (\theta(t) - \theta_t^{\epsilon}(t))^{\top}\left( \dot{\theta(t)} - \dot{\theta_t^{\epsilon}(t)} \right), \\\\  &= (\theta(t) - \theta^{\epsilon}(t))^{\top} \left( \nabla L{(\theta^{\epsilon}(t))} - \nabla L{(\theta(t))} \right), \\\\
> &\leq \left( \sup_{x \in \mathcal{B}}{\|\nabla^2 L\|} \right) \|\theta(t) - \theta_t^{\epsilon}(t) \|^2.
> \end{align*}
> The inequality rests on the assumption that the trajectories stay in a bounded region $\mathcal{B}$. Using the above evolution and by Grönwall's inequality, we have $\|\theta(t) - \theta_t^{\epsilon}(t) \| \lesssim \mathcal{O}\left(\varepsilon\right), \; \forall t < \infty$.
> This bound establishes the closeness of the trajectories. Regarding the second part of the question, we do not think the joint optimization would inflate the component in the kernel space of $X$ but it will not diminish such a component either.
>
>  > Where are the characterizations of singular values of $W_2$ ?
>
> The expressions for the singular values of $W_2$ can be found in Lemma C.10 in the appendix.

---

> > ### Comment · Reviewer_itua · 2023-08-11
> > **Review update**
> >
> > I would like to thank the authors for their thoughtful replies to my review (and all the other reviews). The response has certainly clarified some points, though ultimately I don't think they change my overall impression of the manuscript—there are some interesting results but certainly (significant) limitations as well.

---

### Official Review · Reviewer_zebK · 2023-07-08

**Soundness:** 3 good
**Presentation:** 3 good
**Contribution:** 2 fair
**Rating:** 5
**Confidence:** 4

**Summary:**

This paper explores the behavior of two-layer fully connected networks with linear activations trained using gradient flow on the square loss. The authors reveal that the optimization process is influenced by the initial parameter values and exhibits an implicit bias. Notably, in the case of small-scale initializations, the hidden layer of the network tends to have a low-rank structure. The paper introduces a variational characterization of the loss minimizers obtained under specific initialization conditions and presents a hidden mirror flow that tracks the evolution of singular values in the weight matrices. Numerical experiments are provided to support the findings and demonstrate the observed phenomena.

**Strengths:**

$\bf Originality:$ First, it introduces a novel approach by studying overparameterized vector regression problems in two-layer fully connected linear neural networks. The focus on linear activations and gradient flow training provides a unique perspective. Additionally, the variational characterization of loss minimizers and the introduction of a hidden mirror flow to track singular values contribute to the originality of the paper.

$\bf Quality:$ The paper exhibits a high level of quality in terms of its theoretical analysis and methodology. Theorems 3.1 and 3.3 provide rigorous mathematical proofs, establishing the relationship between gradient flow training and the potential of the neural network's parameters and singular values. The explicit expressions derived for the singular values offer concrete insights.

$\bf Clarity:$  The paper is well-written and effectively communicates its ideas. The abstract provides a concise overview of the paper's content, and the contributions are clearly stated. Theorems and propositions are presented with clarity, and the mathematical derivations are logically organized and explained.

$\bf Significance$:  The findings regarding the implicit bias induced by the initialization scale and its impact on the low-rank structure of the hidden layer contribute to our understanding of optimization processes in linear neural networks. The variational characterization and hidden mirror flow provide valuable insights into the geometrical structure of training dynamics. The proposition on stochastic noise-induced low-rank structures expands the understanding of how noise affects network weights. These contributions have implications for network design, initialization strategies, and the optimization landscape of linear neural networks.


**Weaknesses:**

$\bf (1)$ Restrictive setting: In this work, the theoretical analysis is conducted on a two-layer linear network, which is too restrictive. Many existing works have more general settings, see, e.g., Ref [1].

[1] S. Arora, N. Cohen,W. Hu, and Y. Luo. Implicit regularization in deep matrix factorization. Advances in Neural Information Processing Systems, 32, 2019b.

$\bf (2)$ Limited discussion on practical implications: While the paper provides valuable theoretical insights into the behavior of linear neural networks, it could benefit from a deeper exploration of the practical implications and applications of the findings. How can the observed low-rank structure or the understanding of gradient flow dynamics be utilized in practical scenarios? Discussing potential applications or implications for network design, initialization strategies, or optimization algorithms would enhance the practical relevance of the paper.

$\bf (3)$ Experimental scope and validation: The paper's experimental section is based on a toy example, which may limit the generalizability and applicability of the findings. The authors should consider conducting more extensive experiments to strengthen the empirical support for their theoretical results.

**Questions:**

$\bf (1)$ Can the insights obtained from this specific setting be applied to deeper networks or networks with non-linear activations?

$\bf (2)$ How does the analysis technique employed in this work compare to existing techniques in the literature? Does the paper present a novel approach or build upon existing analysis methods? It would be beneficial for the authors to clearly highlight the unique aspects of their analysis and how it differs from or improves upon existing approaches in terms of theoretical insights or methodological advancements.

---

> ### Author Rebuttal · Authors · 2023-08-09
>
>
> We thank the reviewer for the valuable feedback and interesting questions which help us in clarifying and improving the paper.
>
> > Restrictive setting: In this work, the theoretical analysis is conducted on a two-layer linear network, which is too restrictive. Many existing works have more general settings, see, e.g., Ref [1].
>
> While [1] indeed studies deeper linear networks, it is still subject to stringent restrictions. In particular, the extension to deep networks is **solely possibly** due to the assumption of fully-balanced weight layers:
> $$
> W^\top_{j+1}(0)W_{j+1}(0) = W^\top_{j}(0)W_{j}(0) \; \text{for all layers } j,
> $$
> Also, the implicit bias towards nuclear norm is shown only  with a very small scale of initialization. In comparison, our initialization is more permissive as it is unbalanced and captures the high dimensional behaviour of the standard random initialization. Yet we still provide **explicit formulas of the weights' implicit bias** (instead of the product matrix $W = \Pi_j  W_j$) as a function of the initialization scale, **for any initialization scale**.
>
>
> In addition, the implicit bias towards low-rank in general settings studied in [1] is only backed-up empirically (see Section 3.1 of the paper) and the theoretical arguments of the paper do not address this phenomenon head-on, but only touch upon aspects of it (see section "Theoretical illustration"). On the other hand, we provide a rigorous characterization of singular values throughout the entire optimization trajectory (for orthogonal data). Also, the derivations in [1] solely showcase the dynamics' dependence on the network's depth and not on the initialization scale, as in the present manuscript.
>
>
> > Limited discussion on practical implications: While the paper provides valuable theoretical insights into the behavior of linear neural networks, it could benefit from a deeper exploration of the practical implications and applications of the findings. How can the observed low-rank structure or the understanding of gradient flow dynamics be utilized in practical scenarios? Discussing potential applications or implications for network design, initialization strategies, or optimization algorithms would enhance the practical relevance of the paper.
>
> Thank you for bringing up the practical implications of our work. We will now address these concerns.
>
> - Learning a low-rank network indicates the model may be represented by a much smaller network. This has implications in model compression [1], knowledge distillation [2], the lottery ticket hypothesis [3], and feature quantization [4].
> - A rigorous characterization of the trajectory of gradient methods can give insight into whether early-stopping is beneficial, and what implications it has for the learned model.
> - In terms of  initialization strategies, we complement existing works showing the optimization benefits of initializing the layers with orthogonal matrices by demonstrating their implicit regularization guarantees. Regarding the training algorithm, we show how stochastic training can still exhibit learning in the rich regime even when the initialization scale is not low. This indicates the benefits of stochastic training.
>
> [1] Denton et. al., Exploiting linear structure within convolutional networks for efficient evaluation. NeurIPS 2014.
>
> [2] Hinton et al. Distilling the knowledge in a neural network, 2015.
>
> [3] Frankle, J. and Carbin, M. The lottery ticket hypothesis:  sparse, trainable neural networks, 2018.
>
> [4] Ramesh et. al.. Hierarchical text-conditional image generation with CLIP latents, 2022.
>
>
> > Experimental scope and validation: The paper's experimental section is based on a toy example, which may limit the generalizability and applicability of the findings. The authors should consider conducting more extensive experiments to strengthen the empirical support for their theoretical results.
>
> Please refer to the additional experiments and the general response showing that the described phenomenon extends empirically to the case of non-linear ReLU activations.
>
> >    How does the analysis technique employed in this work compare to existing techniques in the literature? Does the paper present a novel approach or build upon existing analysis methods?
>
> The novelty of the analysis technique can be summarized by the following points (which will be added at the start of the appendix):
>  -  Existing techniques target scenarios with either fully-balanced initialization (see above for the mathematical expression) or initializations with a very small scale. Our choice of initialization removes these requirements.
>  - To give a variational formulation of the weights at convergence, the existing analyses (for the diagonal linear networks) crucially depend on the existence of a mirror descent equivalent of the gradient flow. Our work takes a different approach, where we first assume an \textit{ansatz} on the potential. We further derive the required conditions for the potential to be minimized and show that these conditions hold along the gradient flow trajectory by studying the evolution of a key quantity ($\alpha$).
> - The mirror descent tracking the dynamics for scalar regression ($k=1$) is new. Earlier analyses only hold for the case of $l=1$ (the width) [2]. This result was enabled by studying the evolution of $\alpha$ and writing $\beta$ as a function of $\alpha$ (see Lemma C.2 for details).
> - Regarding stochastic dynamics, we present a **new** analysis aimed at understanding the role of stochastic noise in the algorithm. Leveraging a continuous-time model, we shed light on how the noise further biases towards low-rank structures.
>
> [1] Arora et. al. Implicit regularization in deep matrix factorization. NeurIPS 2019.
>
> [2] Azulay et. al. On the implicit bias of initialization shape: Beyond infinitesimal mirror descent. ICML 2021.

---

### Official Review · Reviewer_mL5m · 2023-07-24

**Soundness:** 4 excellent
**Presentation:** 4 excellent
**Contribution:** 3 good
**Rating:** 5
**Confidence:** 3

**Summary:**

The paper provides an analysis of an implicit bias in the hidden layer for two-layer linear networks under a small-scale initialization regime.

**Strengths:**

This paper shows the hidden layer is biased in a small-scale initialization under over-parameterized two-linear networks.

**Weaknesses:**

The novelty and results significance are not enough given there are a lot of classic papers in this area such as https://arxiv.org/pdf/1806.08734.pdf.

**Questions:**

Do you have a clue to extend your results to more complex neural networks?
Do you think the phenomenon you observe within the two-layer linear network can be extended to a wider range of networks/models?

**Limitations:**

The analysis only applies to two-layer linear networks while there are a lot of large-scaled neural networks in practice.

---

> ### Author Rebuttal · Authors · 2023-08-09
>
> We thank the reviewer for the valuable feedback.
> > The novelty and results significance are not enough given there are a lot of classic papers in this area such as https://arxiv.org/pdf/1806.08734.pdf.
>
> We respectfully disagree with the reviewer's opinion that "the novelty and significance are not enough given there are a lot of classic papers in this area". More specifically, we would like to emphasize the following:
>
> - The study of the implicit bias of training (even linear!) networks has been very intense in the last five years, and it is probably one of the most promising routes to explain the success of deep learning at present. Its relationship with *capacity of predictor space and norms* makes it amenable to provide generalization bounds. Finally, theoretically characterizing it is a hard task, thus every step forward in this direction is important.
>
> - Despite the close similarity of the title, the paper mentioned by the reviewer [1] is **completely orthogonal to ours**. More specifically, [1] studies the spectral bias of the **functions** learned by ReLU networks (and not of the weights!), through the lens of Fourier analysis. In particular, the paper presents the Fourier decomposition of ReLU networks and provides (solely) empirical evidence that lower frequency functions are learned first. We emphasize that the bias in the spectrum of the weight matrices $W_{1, 2}$ studied in our manuscript is of a **completely different nature** than the bias in the Fourier spectrum considered in [1].
>
> - Adding to the previous point, we present a rigorous and tangible argument for the learning process' incremental nature, by looking at the directly-available weight matrices -- as opposed to the more abstract representation provided by the Fourier spectrum in [1]. Our paper thus provides a view which is complementary to that of [1].
>
> - Please also refer to the general answer on possibilities to extend our analysis beyond linear architectures. The novelty of our results is discussed in detail in the comments after Theorem 3.1,3.2 and after Proposition 4.1.
>
> [1] Rahaman, Nasim, et al. "On the spectral bias of neural networks." International Conference on Machine Learning. PMLR, 2019.

---

> > ### Comment · Reviewer_mL5m · 2023-08-17
> >
> > Thank you for your reply. I have decided to raise my score to 5.

---

### Official Review · Reviewer_ynT5 · 2023-07-28

**Soundness:** 3 good
**Presentation:** 4 excellent
**Contribution:** 3 good
**Rating:** 7
**Confidence:** 3

**Summary:**

The paper studies a fairly simple model of deep learning, with a view to understanding the effect of initialisation scale on the spectral structure of the trained network. The model is a two-layer network with out nonlinearity, trained via gradient flow and square loss.

The authors are able to make explicit statements about the flow followed by the singular values, as well as more implicit statements about potentials that the flow minimises after time $t \to \infty$.

This could be useful for building an understanding of the spectral bias of learning algorithms in more realistic situations.

**Strengths:**

My overall impression of this paper is very good. The authors bite off a relevant albeit idealised problem, and derive solid and technically advanced insights about it. The paper is aware of its own limitations, seems to be aware of related work (although I'm not an expert here) and the paper is very well written. I imagine that the paper could inspire future advances towards a more complete picture of the spectral bias of learning.

By the way, it doesn't really bother me that the authors focus on orthogonal init. In some sense the choice of Gaussian init in practice is somewhat arbitrary, and orthogonal init works well too. See, for example, here: https://github.com/pytorch/pytorch/issues/48144

**Weaknesses:**

## Minor issues:
- please label the axes of your plot!
- please explain more what "mirror flow" means
- typo in various places "dependant" --> "dependent"
- typo in line 90:  "towards minimum l2 linear predictors for as vanishingly small initializations"
- remind reader what $P$ is in Lemma 3.4
- different font for $P$ used in line 284

**Questions:**

Some questions about extensions:
- How hard is it to extend this analysis to deeper networks, and why?
- How hard is it to include a nonlinearity, and why?
- What's the intuitive reason for using orthogonal init?

On that vein, I think it might help followup work if you added a section that explains what the main difficulty is in extending these results to a more realistic scenario along various axes.

- have you thought about a rescaled gradient flow, where the flow for one layer is adjusted relative to the other by a constant? This would correspond to the idea that in the discretised setting, both layers would do not need to have the same learning rate
- does the work suggest any insights about how to do explicit regularisation better?

---

> ### Author Rebuttal · Authors · 2023-08-09
>
> We thank the reviewer for the valuable feedback and interesting questions which help us in clarifying and improving the paper.
>
> > What's the intuitive reason for using orthogonal init?
>
> The orthogonal initialization captures the high-dimensional asymptotics of initialization with i.i.d. Gaussian random vectors, as such random vectors are almost orthogonal in high dimensions. Specifically, as the width $l \ggg d $, $W_1 W_1^{\top} \sim I_{d \times d} $.  Furthermore, technically speaking, this choice eases the theoretical analysis because the invariant (the quantity in Lemma 3.5) becomes isotropic and it gains some commutativity properties (the layers share singular vectors), a fact which is discussed in the comments following Theorem 3.2 and in the Appendix in Theorem C.11.
>
> > Have you thought about a rescaled gradient flow, where the flow for one layer is adjusted relative to the other by a constant? This would correspond to the idea that in the discretised setting, both layers would do not need to have the same learning rate
>
> Thank you for this interesting question. Our analysis does support the rescaled gradient flow. Consider the following dynamics:
> \begin{align*} \dot{W_1} = \epsilon R W_2^{\top} , ~~ \dot{W_2} = W_1^{\top} R, \end{align*}
> which induces the projected dynamics (analogue to eq. (3.4) in  the paper)
> \begin{align*} \dot{Z_1} = \epsilon R Z_2^{\top} , ~~ \dot{Z_2} = Z_1^{\top} R,  \end{align*}
> for which the rescaled invariance equations become $Z_1^{\top}Z_1 - \epsilon Z_2Z_2^{\top} = 2 \gamma I$ (analogue to Lemma 3.5 in our work). Building upon this invariance, we can derive expressions for the singular values at convergence in a much similar way as done already in the manuscript. In the limit of $\epsilon \to 0$, the limit $W_1^{\infty}$ is of approximately full-rank, while for large $\epsilon$,  $W_1^{\infty}$ has a bias towards low-rank. This behavior makes sense since for $\epsilon \to 0$, the feature layer $W_1$ is trained very slowly and feature learning does not happen, while for large $\epsilon$ the feature layer is trained faster and learns the appropriate features.
>
> > Does the work suggest any insights about how to do explicit regularisation better?
>
> Regularization with $\ell_2$-norm of the parameters should induce the low rank behaviour (if a global minimum of the regularized objective is reached). However, with explicit regularization the dynamics will not converge towards a solution that interpolates the training loss.
>
>  > How hard is it to extend this analysis to deeper networks, and why?  How hard is it to include a nonlinearity, and why?
>
> We kindly point the reviewer to the general comment where we discuss the technical difficulties of extending the analysis to deeper and non-linear networks.

---

> > ### Comment · Reviewer_ynT5 · 2023-08-16
> >
> > Thank you for both your rebuttal and the top-level comment above, which I have read. I have decided to maintain my score.

---

### Author Rebuttal · Authors · 2023-08-09

We thank the reviewers for the detailed feedback and positive evaluations. We address general concerns of the reviewers below.


## Networks with more depth. [Reviewers ynT5, zebK, mL5m ]
 A first thought to tackle this problem is to simply extend our result by initializing the first layer as an orthogonal one and all other layers as zero. With a similar analysis as the one for two-layer network, we would obtain similar results.  However, such extensions would not contribute to any further understanding or insights as layers other than the first one will be balanced and existing works focusing on balanced initialization already capture this effect.

For general unbalanced initializations, a rigorous characterization of the implicit bias is still an open problem (even in the two-layer case). In the current literature, the only architecture where a rigorous characterization is proven lies in the case of 2-layer diagonal networks. The characterization here depends on the existence of a hidden mirror flow. For general linear networks, such an equivalence does not hold. In our results, we overcome this difficulty by an ansatz on the potential function and show that gradient flow ensures some properties of the relevant quantities ($\alpha$) (more precisely that $\alpha$ stays in $\mathrm{span}(\{X\})$).

The hard part to extend these results to deeper networks is to come up with an ansatz for the potential and identify key quantities which can be tracked as we track $\alpha$.

---------------------------------------------------------------------------------------------

## Networks beyond linear activations. [Reviewers ynT5, zebK, mL5m]
We believe the low-rank phenomenon to be universal and to extend to (homogeneous) non-linear activation. Rigorous experiments with deeper networks demonstrating a low rank phenomenon can be seen in [1,2] but theoretically unravelling this behaviour is still an important open problem. We believe our analysis can serve as a stepping stone to analyze non-linear settings.


To showcase our result's expected extension, we add a toy experiment for ReLU networks (see attached pdf). More precisely, we consider a scalar regression problem modelled through a ReLU network in a teacher-student setup. We generate a training set of size $10$ sampled from a random Gaussian distribution in $\mathbb{R}^{5}$. The training data $(x_i,y_i)_{i=1}^{10} \in \mathbb{R}^{5} \times \mathbb{R}$ is generated by a teacher ReLU network with $2$ neurons $(w_0,w_1)$, i.e.,\begin{align*} y_i =  a_0 \sigma(w_0^{\top}x_i) + a_1 \sigma(w_1^{\top}x_i) , \end{align*} where $\sigma$ is the ReLU non-linearity. We train a student network with 20 hidden neurons. Note that there are two relevant directions $w_0,w_1$ for the student network to learn, therefore we expect the hidden layer to represent these two directions (i.e., a rank 2 hidden layer, and a singular value decomposition with two non-zero singular values). This property is empirically verified in Figure 1 of the attached pdf where we plot the time evolution of singular values. When initialized at low-scale the network converges to an approximately rank-2 matrix. When initialize at large scale, the network is high rank and the neurons do not learn the teacher directions. Thus this new experiment demonstrates the presence of low rank structure in non linear networks and its dependence on the initialization scale.

However, providing a theoretical characterization of the dynamics or of the final weight parameters is very challenging. To study even this simple case of two neurons with a very low initialization scale, one has to study the dynamics in two phases where the iterate jumps from zero initialization (a saddle point) to another saddle and then further converge to zero train loss as seen in Figure 2a. It is difficult to precisely characterize this intermediate saddle. With some careful additional work, we believe that our analysis can capture Phase 1, in which the dynamics progresses towards the first saddle and a rank-1 matrix is approximately learned. This stage is depicted in Figure 2b. We believe the strategy discussed above can help extend the current understanding of ReLU networks to general (i.e., not orthogonal) data matrices, demonstrating how our analysis of the linear case fits in the broader context of non-linear networks.

[1] M. Huh, H. Mobahi, R. Zhang, B. Cheung, P. Agrawal, and P. Isola. The low-rank simplicity bias in deep networks. TMLR, 2021.

[2] M. Andriushchenko, A. Varre, L. Pillaud-Vivien, and N. Flammarion. SGD with large step sizes learns sparse features. In ICML, 2023.

-----

We again thank the reviewers for their valuable comments and remain open to further discussions on any other points they may raise.

---

### Decision · Program_Chairs · 2023-09-21

**Decision:**

Accept (poster)

**Comment:**

This paper investigates the behavior of two-layer fully connected networks with linear activations when subjected to gradient flow training using the square loss. The authors examine how the optimization process introduces an implicit bias on the network parameters, which is contingent upon the scale of their initialization.The central contribution of the study lies in presenting a variational characterization of the loss minimizers achieved by the gradient flow under specific initialization conditions. This characterization unveils that, within the context of small-scale initializations, the hidden layer of the linear neural network exhibits a bias toward adopting a low-rank structure.

All reviewers thought the paper had interesting ideas and recommended acceptance. They raised some technical concerns most of which were addressed. Therefore I recommend acceptance.